# Meru co-ordinates spindle orientation with cell polarity and cell cycle progression

Melissa M McLellan [ID][1,3], Birgit L Aerne [ID][1,3], Jennifer J Banerjee Dhoul[1], Maxine V Holder [ID][1], Tania Auchynnikava[2] & Nicolas Tapon [ID][1✉]

## Abstract

**Correct mitotic spindle alignment is essential for tissue architecture and plays an important role in cell fate specification through asymmetric cell division. Spindle tethering factors such as *Drosophila* Mud (NuMA in mammals) are recruited to the cell cortex and capture astral microtubules, pulling the spindle in the correct orientation. However, how spindle tethering complexes read the cell polarity axis and how spindle attachment is coupled to mitotic progression remains poorly understood. We explore these questions in *Drosophila* sensory organ precursors (SOPs), which divide asymmetrically to give rise to epidermal mechanosensory bristles. We show that the scaffold protein Meru, which is enriched at the posterior cortex by the Frizzled/Dishevelled planar cell polarity complex, in turn recruits Mud, linking the spindle tethering and polarity machineries. Furthermore, Cyclin A/Cdk1 associates with Meru at the posterior cortex, promoting the formation of the Mud/Meru/Dsh complex via Meru and Dsh phosphorylation. Thus, Meru couples spindle orientation with cell polarity and provides a cell cycle-dependent cue for spindle tethering.**

**Keywords** Spindle Orientation; Asymmetric Cell Division; Cell Polarity; Drosophila; Development
**Subject Categories** Cell Adhesion, Polarity & Cytoskeleton; Cell Cycle; Development

## Introduction

Accurate mitotic spindle orientation is key to the development and adult homoeostasis of multicellular organisms. In symmetrically dividing epithelial cells, the mitotic spindle generally aligns parallel to the tissue plane, allowing the daughter cells to insert themselves seamlessly into the epithelium, thereby maintaining tissue architecture and integrity (Bergstralh et al, 2017; Ragkousi and Gibson, 2014). Correct spindle orientation is also essential for the generation of cell diversity through the process of asymmetric cell division (ACD). During ACD, cell fate determinants (CFDs) are

unequally inherited by the daughter cells, promoting progenitor self-renewal or differentiation (Sunchu and Cabernard, 2020). CFD asymmetric segregation depends upon the polarisation of the cell cortex, which directs spindle alignment and ensures unequal CFD inheritance. Thus, cell polarity and spindle orientation must be tightly linked, and aberrant alignment can result in developmental defects and neoplasia (Bergstralh et al, 2017; Ragkousi and Gibson, 2014; Sunchu and Cabernard, 2020).

Mushroom body defective (Mud; Nuclear Mitotic Apparatus Protein/NuMA in mammals; LIN-5 in *C. elegans*) is a conserved spindle tethering factor that is recruited to the cell cortex by cell polarity proteins and exerts forces on astral microtubules by binding the Dynein/Dynactin motor complex (di Pietro et al, 2016). A core complex comprising the Mud binding partner Pins (Partner of inscrutable)/LGN and Gαi has been implicated in Mud cortical recruitment (Bergstralh et al, 2017; Morin and Bellaïche, 2011; Siller and Doe, 2009). However, how Pins itself is recruited to the correct subcellular location and whether it is even required for Mud localisation is clearly context-dependent. For instance, *Drosophila* Mud localises to the cortex in a Pins-dependent manner in the ovarian follicular epithelium and via a Pins-independent mechanism in the wing and thoracic epithelia (Bergstralh et al, 2016; Bosveld et al, 2016; Nakajima et al, 2013, 2019; Neville et al, 2023). Thus, how the spatial alignment of the mitotic spindle adapts to the polarity of different symmetrically or asymmetrically dividing cell types remains unclear. Recent work has also proposed that, as well as being cortically recruited, the Mud/NuMA spindle tethering complex must be activated as the cells enter mitosis (Neville et al, 2023). However, our understanding of how the spindle tethering machinery is coupled to cell cycle progression remains limited. Here, we use the asymmetric division of *Drosophila* sensory organ precursor cells (SOPs) as a model to study these questions.

SOPs (also known as pI), which give rise to the adult sensory bristles of the fly epidermis, are a well-studied example of ACD (Hartenstein and Posakony, 1989; Schweisguth, 2015). The best-studied SOPs are located on the dorsal thorax (notum) and produce the mechanosensory microchaetes that cover this tissue (Hartenstein and Posakony, 1989; Schweisguth, 2015). SOPs undergo consecutive rounds of asymmetric division to produce five distinct cell types: neuron, sheath, shaft, socket and a glial cell that undergoes apoptosis (Hartenstein and Posakony, 1989). The SOP lineage cells then recruit a neighbouring epidermal cell (the F-Cell)

[1]Apoptosis and Proliferation Control Laboratory, The Francis Crick Institute, 1 Midland Road, London NW1 1AT, UK. [2]Proteomics Science Technology Platform, The Francis Crick Institute, 1 Midland Road, London NW1 1AT, UK. [3]These authors contributed equally: Melissa M McLellan, Birgit L Aerne. ✉E-mail: nic.tapon@crick.ac.uk

and together these assemble into a functional sensory hair (Mangione et al, 2023). In the notum, SOPs are specified at ~12 h APF (hours After Puparium Formation) from the epithelial sheet through the highly conserved Notch pathway (Corson et al, 2017; Gómez-Skarmeta et al, 2003; Simpson, 2007).

Until the SOPs are specified, they share the same polarity as the surrounding epithelial cells, characterised by two polarity axes: (1) planar cell polarity (PCP), whereby the cells align in the tissue plane and (2) apical-basal (A–B) polarity which defines the cellular apical and basal domains separated by the adherens junctions (AJs) (Buckley and St Johnston, 2022; Goodrich and Strutt, 2011). The core PCP pathway is established by three transmembrane proteins that form opposing domains at the AJs through mutual antagonism (Goodrich and Strutt, 2011; Yang and Mlodzik, 2015). In the notum, heterodimers of the seven-pass transmembrane protein Flamingo (Fmi, also known as Starry night/Stan; CELSR1/2/3 in vertebrates) with the four-pass transmembrane protein Van Gogh (Vang, also known as Strabismus; Vang1/2 in vertebrates) are located on the anterior side of each cell, while heterodimers of Fmi with the seven-pass protein Frizzled (Fz; Fz1 in vertebrates) are on the posterior side (Bellaïche et al, 2001; Schweisguth, 2015; Ségalen et al, 2010). Vang then recruits its downstream effector Prickle (Pk) (Bastock et al, 2003; Jenny et al, 2003; Taylor et al, 1998), while Fz recruits Dishevelled (Dsh; DVL1-3 in vertebrates) and Diego (Dgo) (Axelrod et al, 1998; Feiguin et al, 2001; Jenny et al, 2005). In A–B polarity, the apical domain is defined by the Par complex, comprised of Bazooka (Baz; Par3 in vertebrates), atypical Protein Kinase C (aPKC; PKCι/PKCζ in vertebrates) and Par6 which regulates the placement of the AJs (Bilder et al, 2000; Buckley and St Johnston, 2022; Tepass, 2012). The basolateral domain is defined by the septate junction components Scribble (Scrib; SCRIB in vertebrates), Discs large (Dlg; Dlg1-5 in vertebrates) and Lethal giant larvae (Lgl; LLGL1/2 in vertebrates) (Albertson et al, 2004; Bilder et al, 2000; Woods and Bryant, 1991).

Upon SOP specification, the proneural transcription factors of the Achaete-Scute complex turn on the expression of the N-terminal RASSF (Ras-association domain family) protein Meru (Banerjee et al, 2017; Buffin and Gho, 2010; Reeves and Posakony, 2005). Meru is enriched to the posterior cortex by Dsh, and in turn recruits Baz, leading to its planar asymmetry, specifically in the SOP (Banerjee et al, 2017). Baz then polarises the rest of the Par complex to the posterior cortex, leading to the exclusion of the CFDs Numb and Neuralized (both Notch pathway modulators) from the posterior pole prior to ACD (Bellaïche et al, 2001; Bellaïche et al, 2001; Besson et al, 2015; Le Borgne and Schweisguth, 2003; Roegiers et al, 2001a, 2001b; Smith et al, 2007; Wirtz-Peitz et al, 2008). Thus, Meru provides the initial planar bias that triggers the polarisation of the Par complex and the CFDs along the antero-posterior axis (Banerjee et al, 2017). To ensure accurate segregation of CFDs, the mitotic spindle must also align along the antero-posterior axis. This is achieved by redundant spindle tethering complexes on the anterior and posterior poles (Gomes et al, 2009; Schweisguth, 2015) (Fig. 1A). On the anterior side, Gαi/Dlg/Pins recruit Mud basally, while on the posterior side, Mud is recruited in a Fz/Dsh-dependent manner to the apical cortex (Bellaïche et al, 2001; Bellaïche et al, 2001; David et al, 2005; Johnston et al, 2013; Schaefer et al, 2000; Ségalen et al, 2010). This allows the capture of one centrosome each by the anterior and posterior poles, but also imparts a characteristic A–B tilt to the spindle (David et al, 2005;

Schweisguth, 2015; Ségalen et al, 2010) (Fig. 1A). Thus, SOP division is a powerful system to study how spindle orientation adapts to cellular context, since two distinct Mud localisation mechanisms co-exist in this cell type.

An important open question remains how Mud is recruited to the posterior cortex as the SOPs enter mitosis. Two potential mechanisms have been proposed: (1) Mud binds directly to Dsh (Ségalen et al, 2010) and (2) the mitotic cyclin, Cyclin A (CycA), is recruited to the posterior cortex in a Dsh-dependent manner and in turn recruits Mud (Darnat et al, 2022). Here, we show that, in parallel to its role in posterior Baz recruitment, Meru acts as a bridging factor between Dsh and Mud. Examination of the Meru sequence revealed a CycA docking site that is required for Meru cortical localisation, Mud localisation and spindle alignment. We also identified multiple Cdk1 (Cyclin-dependent kinase 1) phosphorylation consensus sites that promote the Meru/Mud interaction. Meru therefore links PCP with spindle tethering, and its ability to recruit Mud is coupled to the cell cycle through CycA to ensure timely alignment of the spindle during mitosis.

## Results

### Meru localises to the apical–posterior pole prior to and during SOP mitosis

To address the potential role of Meru in SOP spindle alignment, we generated a *UAS-mKate2(mK2)-meru* transgenic line to track its localisation during SOP divisions relative to known polarity and spindle factors. We had previously shown that endogenously tagged GFP-Meru localises to the posterior cortex in late G2, where it remains enriched throughout mitosis and colocalises with Dsh (Banerjee et al, 2017). No bristle defects were observed in animals that had mK2-Meru driven in SOPs by the *neur^P72-GAL4* driver (*neurG4*) (Bellaïche et al, 2001), suggesting overexpression does not lead to strong SOP division defects (Fig. 1B–C'). mK2-Meru localised to the posterior–apical cortex at the level of E-cadherin (Ecad) as expected (Banerjee et al, 2017), (Figs. 1D and EV1A,A'). Upon mitotic rounding, Partner of Numb (Pon), the adaptor of the CFD Numb, is rapidly relocalised to the basolateral anterior cortex during prophase, followed by its unequal inheritance into the anterior pIIb daughter cells (Bellaïche et al, 2001; Roegiers et al, 2001b). Co-expression of mK2-Meru and Pon showed that mK2-Meru was present at the posterior domain before the asymmetric localisation of Pon was detectable, indicating SOP polarisation occurs prior to CFD segregation (Fig. 1E,E'). Thus, Meru is localised at the posterior–apical cortex during most of SOP mitosis, consistent with a possible role in spindle positioning.

### Meru promotes Mud recruitment to the posterior cortex

Dsh and Mud have been shown to colocalise at the posterior–apical cortex during SOP mitosis (Ségalen et al, 2010). We reported that Meru and Dsh associate in cell culture, colocalise in fixed tissue, and that Dsh is required for Meru posterior localisation (Banerjee et al, 2017). We therefore sought to test whether there was an interaction between Meru and Mud by tracking the localisation of mK2-Meru and a GFP-tagged Mud construct expressed under the *mud* promoter (GFP-Mud) with live-imaging (Ségalen et al, 2010). Mud and Meru localisations were highly correlated (as indicated by a Pearson's coefficient *r* near +1),

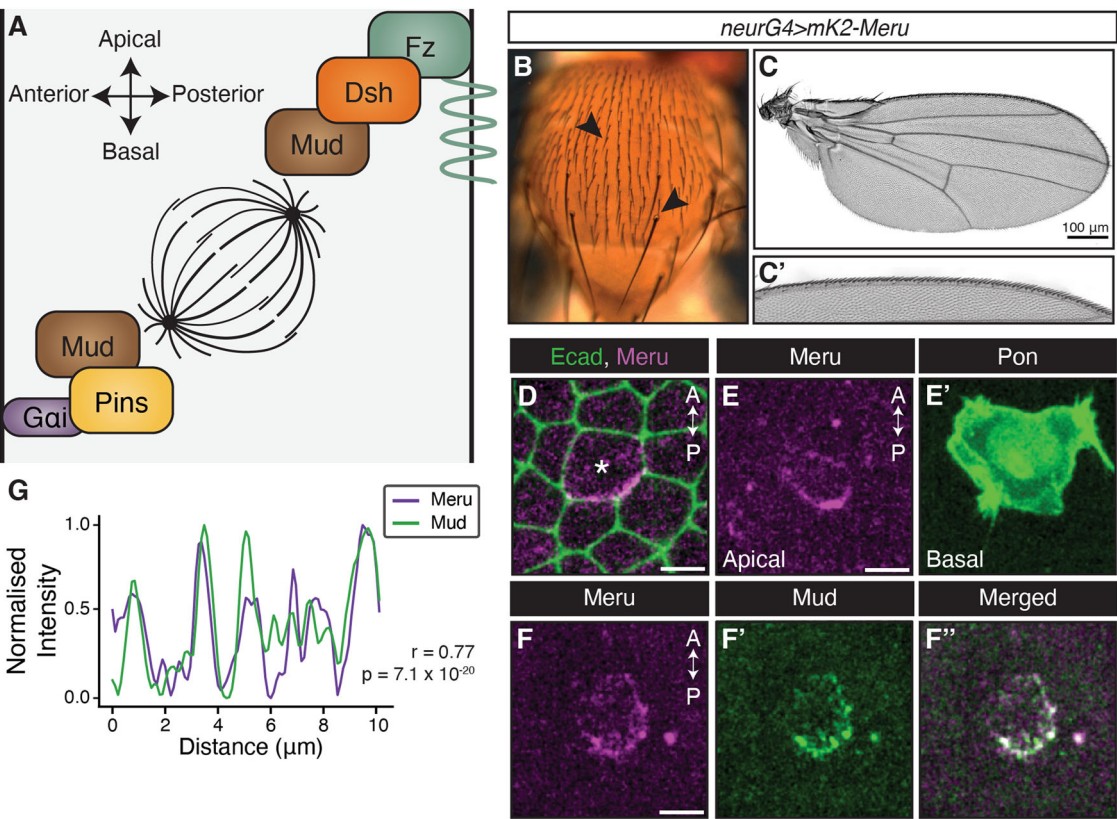

**Figure 1. Meru localises to the apical–posterior domain of the SOP.**

(A) The current model of SOP spindle alignment where Mud is recruited to the anterior cortex by Pins/Gαi and to the posterior cortex through Fz/Dsh. Note that the extent of spindle tilt in the diagram is amplified compared with reality for illustrative purposes. (B, C) Brightfield images of the adult notum (B), wing (C), and anterior wing margin (C') in animals expressing *UAS-mK2-meru* under the *neurG4* driver. No visible defects (arrows indicating micro/macrochaetes) in the notum and wing sensory organs were observed. (D–F) Maximum intensity projections of pupal nota at 16 h APF of *neurG4 > UAS-mK2-meru* (magenta, D–F), Ecad-GFP (composite of Fig. EV1A–A'), Pon-GFP and Mud-GFP (green; D, E' and F', respectively) at 18 min prior to first indication of cytokinesis. SOP marked by a star. Scale bars = 5 μm. (G) Single brightest slice in (F, F') plotted as line graph by measuring the grey value across the posterior domain as defined by Meru localisation normalised to the highest value in each channel. High Pearson's coefficient (r) indicates a positive correlation. P value calculated using a two-tailed test. Source data are available online for this figure.

particularly at the onset of mitosis (Fig. 1F,G). Despite a considerable drop in Mud intensity levels, the two proteins remained positively correlated even after metaphase (Figs. EV1B-C").

The strong Meru/Mud colocalisation is consistent with a physical interaction between these proteins. To test this possibility, we performed co-immunoprecipitations (co-IPs) from S2 cell lysates expressing full-length Meru together with Mud fragments spanning much of the protein (Ségalen et al, 2010) (Fig. 2A,B). We found that Meru binds to the C-terminal part of Mud (1825–2456), a region which is largely disordered, but contains a short domain that mediates Pins binding, known as Pins-Binding Domain (PBD —aa 1825–1961) and a transmembrane-like (TML) domain. The PBD has been shown to recruit Mud cortically in SOPs and neuroblasts (Ségalen et al, 2010; Siller et al, 2006). As Mud (1452–1961) which contains the PBD binds Meru, but Mud (1452–1824) which lacks the PBD does not, the PBD represents one interaction surface for the Meru/Mud interaction (Fig. 2B). Moreover, two truncations lacking the PBD, Mud (1951–2456) and Mud (2089–2499) can both co-immunoprecipitate Meru, though not as efficiently as Mud (1825–2456), which has both the PBD and C-terminus (Fig. 2B). This suggests that Meru/Mud association is mediated both by the Mud PBD and C-terminus (aa 1951–2456).

A previous report showed an interaction between Mud and Dsh in HEK293T cells (Ségalen et al, 2010). We were unable to detect such an interaction in S2 cells between full-length Dsh and any of the Mud fragments (Fig. 2C). However, Meru is not endogenously expressed in S2 cells, whereas the orthologs of Meru (RASSF9/10) are expressed in HEK293T cells (Hauri et al, 2013). Together with our finding that Meru interacts with both Dsh and Mud in cell culture, this prompted us to test whether Meru could bridge the previously reported Dsh/Mud interaction (Ségalen et al, 2010). Indeed, co-expression of Meru elicited a robust association between Dsh and Mud (1452–1961) (Fig. 2D). Thus, Meru colocalises with Mud and Dsh at the posterior cortex during SOP cell divisions and mediates the assembly of a Mud/Meru/Dsh complex in cell culture.

## Meru promotes cortical Mud localisation and proper spindle orientation in vivo

In wild-type SOPs and epithelial cells, Mud localises to three regions: at the cortex (with polarity proteins), at the centrosomes, and at tricellular junctions (Bosveld et al, 2016; Izumi et al, 2006; Ségalen et al, 2010). Specifically, Mud localisation to the posterior cortex in SOPs is dependent on Dsh (Ségalen et al, 2010). As Meru

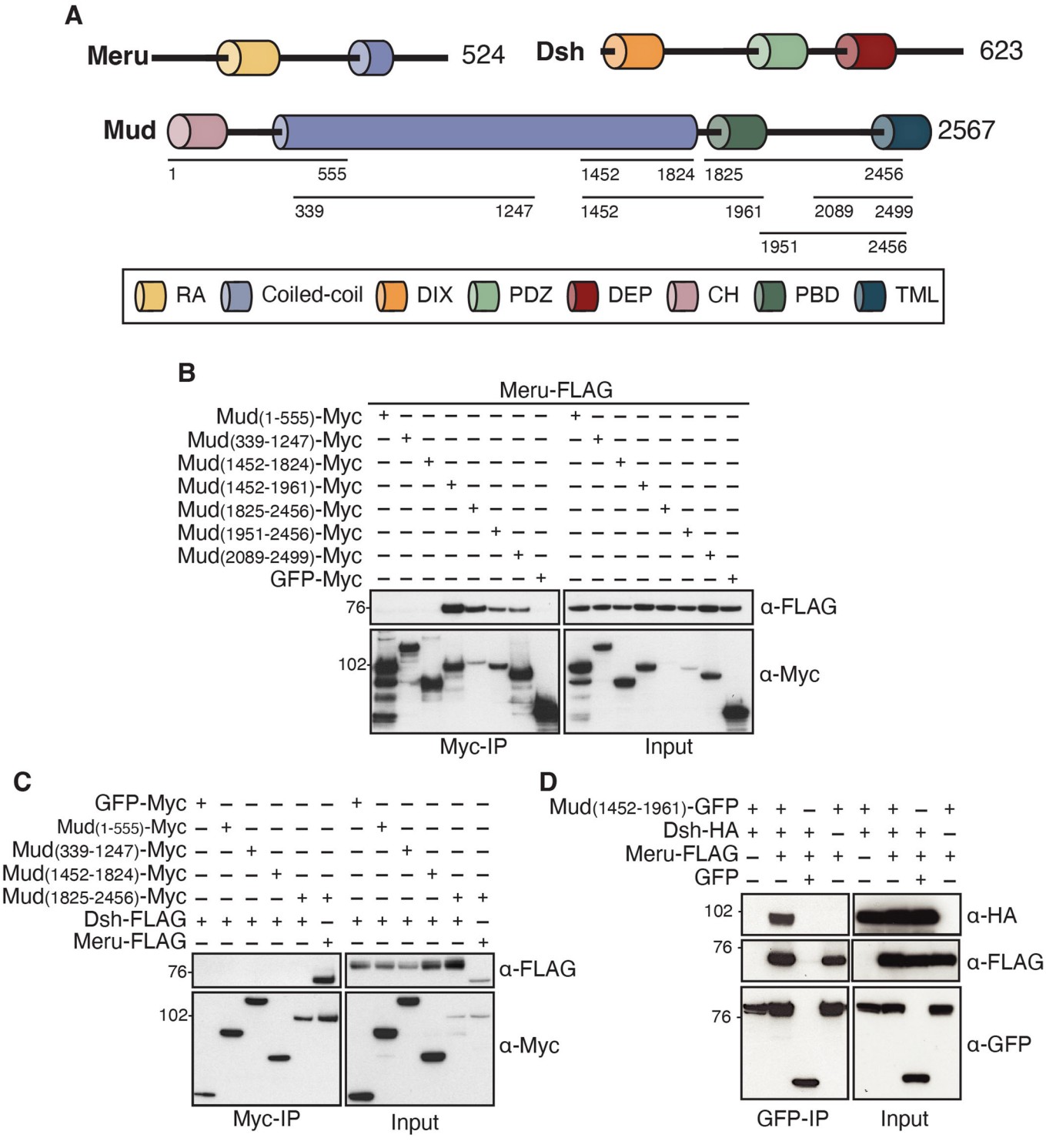

**Figure 2. Meru is required for Dsh/Mud complex formation.**

(A) Schematic of constructs that span Meru, Dsh and Mud used in co-IPs to characterise Meru binding in S2 cells. RA Ras-association domain, DIX DIshevelled and aXin domain, PDZ = PSD96, Dlg, ZO-1 domain, DEP Dishevelled, Egl-10 and Pleckstrin domain, CH Calponin Homology domain, PBD Pins-Binding Domain, TML Trans-Membrane-Like domain. (B–D) Western blots of co-IP experiment using S2 cell lysates from transfected S2 cells, immunoprecipitated using and probed with the indicated antibodies. (B) Full-length Meru immunoprecipitates with Mud fragments containing the PBD and C-terminus. (C) Mud does not co-IP with full-length Dsh. The Meru/Mud interaction is used as a positive control (right-most lane). (D) Meru promotes Dsh/Mud complex formation. Source data are available online for this figure.

is required for the Dsh/Mud interaction in cell culture, we tested whether posterior cortical Mud recruitment is dependent on Meru in vivo. We quantified Mud-GFP at both the anterior and posterior cortex in *meru[1]* mutants (a 1.6 kb deletion that removes half the coding region, including the RA domain (Banerjee et al, 2017)) compared to wild-type flies (Fig. 3A–C). We noted that, in wild-type SOPs, Mud crescent intensity is consistently higher on the anterior side than the posterior (Fig. 3C). We observed a significant decrease in Mud intensity at the posterior cortex of *meru[1]* mutant SOPs relative to the wild-type (Fig. 3C). However, anterior cortical Mud levels were also decreased, potentially indicating a role for Meru in Mud stability (Fig. 3C). Thus, Meru promotes Mud cortical recruitment, consistent with a role in spindle alignment.

We previously showed that *meru[1]* mutant SOPs display cell division axis defects (Banerjee et al, 2017). However, these

measurements were performed in the presence of overexpressed *pon*, which is known to induce changes in SOP polarity (Perdigoto et al, 2008). We therefore directly measured spindle alignment by labelling the spindle using a GFP-tagged version of the microtubule-binding protein Jupiter (Jup) and quantifying the deviation from the A–P axis in wild-type and *meru[1]* mutant flies (Fig. 3D–F'; Movies EV1 and EV2). As the mitotic spindle remains dynamic until metaphase, when it settles for its final division angle (Bellaïche et al, 2001; Bergstralh et al, 2016; Ségalen et al, 2010), our measurements were carried out post-metaphase. Consistent with previous work (Ségalen et al, 2010), 71% of wild-type cells divided within 30° of the A–P axis (Fig. 3D,F), and the spindle alignment in the A–B axis averaged at 6.8°, displaying the characteristic z-tilt of SOP mitotic spindles (David et al, 2005; Ségalen et al, 2010) (Fig. 3F'). Strikingly, *meru[1]* flies had a nearly random spindle

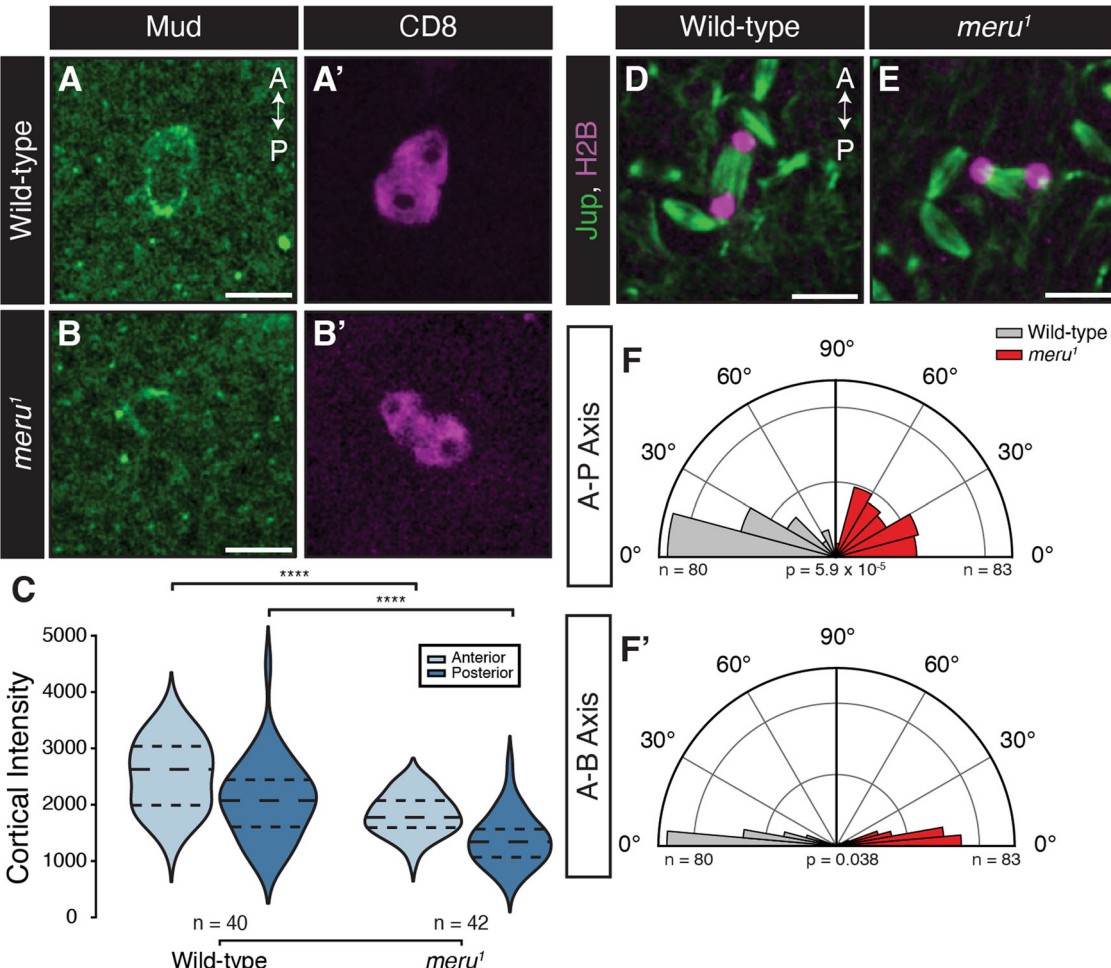

**Figure 3. Meru loss leads to Mud mislocalisation and spindle alignment defects.**

(A, B) Pupal notal confocal live-imaging of *Mud-GFP* (green, max projection of 6 slices) and *UAS-cd8-mRFP* driven by *neurG4* (magenta, max projection of 4 slices) in a *meru* wild-type (A) and *meru[1]* background (B) at 16 h APF at the first indication of cytokinesis. (C) Graph showing the intensity of cortical Mud at the anterior and posterior crescent of each genotype. Measurements were taken at anaphase. Both the anterior and posterior cortical Mud levels were significantly lower in the *meru[1]* flies relative to the wild-type ($P = 1.4 \times 10^{-6}$ and $P = 1.4 \times 10^{-6}$, respectively). ****$P < 0.0001$ using a Mann–Whitney $U$ test. Large and small dashed line represent the median and Q1/Q3, respectively. (D, E) Confocal live-imaging of *neur-H2B-RFP* (magenta) and *Jupiter-GFP* (green) in a *meru* wild-type (D) and *meru[1]* background at approximate cytokinesis (E). (F) Graphs showing the spindle orientation of each genotype relative to the dorsal midline (F) and epithelial plane (F'). *P* values indicated using a Kolmogorov–Smirnov test. Scale bars = 10 μm. Number of SOPs imaged indicated on the panels from three nota in (C) or six wild-type or 4 *meru[1]* nota, respectively, in (F, F'). Source data are available online for this figure.

alignment in the A–P axis, with only 45% of cells dividing within 30° of the midline (Fig. 3E,F). The average A–B angle showed a slight increase to 9.0°, with a significantly wider distribution—nearly 20% of cells dividing over 20°, compared to 2% in controls (Fig. 3F'). Ségalen et al observed that the spindle is more planar in *fz* and *dsh* mutants (Ségalen et al, 2010) rather than more tilted in the A–B axis as it is in *meru* mutants (Fig. 3F') or upon *cycA* depletion (Darnat et al, 2022). The likely explanation is that the authors analysed tissues that are uniformly mutant for *fz* and *dsh*. Because the anterior and posterior PCP complexes stabilise each other across cell junctions as well as antagonise each other intracellularly (Goodrich and Strutt, 2011; Yang and Mlodzik, 2015), global loss of *dsh* or *fz* perturbs the organisation of both the anterior and posterior cortices in SOPs which no longer receive PCP input from their neighbours on the anterior side. In these conditions, the cell divisions become planar since spindle tilt is lost (Ségalen et al, 2010). As depletion of *meru* and *cycA* do not affect PCP in neighbouring cells, the basal pulling force from Pins/Mud remains and is no longer counterbalanced by the apical–posterior side and spindle tilt increases (Fig. 3F' and (Darnat et al, 2022)). Consistent with its role in Mud positioning, Meru is therefore required for A–P and planar spindle alignment.

## Meru associates with CycA

CycA has recently been reported to be enriched at the apical–posterior cortex at the end of G2 and early prophase in SOPs (Darnat et al, 2022). We therefore wished to investigate whether CycA could provide a link between cell cycle progression and spindle orientation by functioning in concert with Meru. We first tracked Meru/CycA colocalisation in vivo. In late G2 phase, when both proteins have been reported to be present at the posterior cortex during SOP mitosis (Banerjee et al, 2017; Darnat et al, 2022), we observed that the localisations of mK2-Meru and endogenously tagged CycA were highly correlated (Fig. 4A,B—Pearson's coefficient = 0.71). The colocalisation persisted until metaphase (Fig. 4A), when CycA enters the nucleus, which is followed by its degradation (Lehner and O'Farrell, 1989). We then tested a potential Meru/CycA association by co-IP in S2 cells. Interestingly, CycA robustly immunoprecipitated with Meru, but not Dsh or Mud (Fig. 4C). To determine whether the Meru/CycA interaction is specific, we examined binding to other Cyclins associated with cell cycle progression (A, B, E and D). We observed an interaction with CycB, which is expressed during early M-phase (Lehner and O'Farrell, 1990). However, this CycB association was far weaker than the interaction with CycA (Fig. 4D). Thus, Meru binds to CycA, with which it colocalises at the posterior cortex in SOPs.

## CycA docking regulates Meru localisation and association with Mud and Dsh

CycA binding is highly regulated through short linear motifs (or SLiMs), the most well-known of which is the RxL motif. This consensus sequence consists of amino acids R/K-x-L-φ or R/K-x-L-x–φ (φ = hydrophobic amino acid) (Tatum and Endicott, 2020) (Fig. EV2A). When we investigated the Meru sequence, we found six possible RxL motifs, two of which (RxL 258 and 259) were very good matches and located immediately adjacent to each other (Figs. 5A and EV2A). We generated mutations at all six RxL motifs,

substituting the R/K and L to A. We initially found that Meru/CycA binding was only affected in the RxL 258/9 double mutant (Fig. EV2B). Subsequent mutation of either the 258 or 259 site alone showed that this was sufficient to disrupt the Meru/CycA interaction (Fig. 5B). In all subsequent experiments, we used the double 258/9 mutants (*meru*<sup>RxL</sup>).

To determine if CycA binding modulates Meru function, we tested if the RxL 258/9 motif was required for association with other Meru partners. We observed that the Meru RxL motif is necessary for Mud and Dsh binding (Fig. 5C,D), but dispensable for binding to Baz (Fig. 5E). To distinguish between the RxL mutation directly interfering with Mud binding versus Meru/CycA binding being required for Meru to associate with Mud, we depleted CycA by RNAi (Fig. EV2C). CycA expression was required for the Meru/Mud association, consistent with the RxL mutation interfering with Meru/Mud binding indirectly by compromising CycA docking to Meru.

We then generated *UAS-meru*<sup>RxL</sup> transgenic flies to test the role of Meru/CycA binding in vivo. Strikingly, mK2-Meru<sup>RxL</sup> failed to be recruited to the cortex, in agreement with the RxL mutation compromising Dsh binding, and we noted that, after nuclear envelope reassembly, some mK2-Meru<sup>RxL</sup> accumulated in the nucleus (Figs. 5F and EV3A,A'). Co-expression with Ecad and Dlg showed that mK2-Meru<sup>RxL</sup> was distributed in both the apical and basal cytoplasm (Fig. EV3B,C'). When we expressed mK2-Meru<sup>RxL</sup> in a *meru*<sup>1</sup> mutant background, we also observed a decreased CycA posterior crescent compared with Meru<sup>WT</sup>-expressing animals (Figs. 5F',G and EV3D,D'), in agreement with the loss of binding demonstrated in Fig. 5B. In summary, CycA docking to Meru via the RxL motif is required for Dsh and Mud binding in cell culture, as well as Meru and CycA cortical localisation in vivo.

## CycA docking is required for Meru-dependent spindle orientation

We sought to address if the role of Meru in SOP spindle alignment was regulated by CycA. We first tested the effect of blocking Meru/CycA association on the formation of the ternary Dsh/Meru/Mud complex in S2 cells. In contrast to wild-type Meru, the Meru<sup>RxL</sup> mutant was unable to bridge the Dsh/Mud interaction (Fig. 6A), suggesting that CycA binding to Meru is required for ternary complex formation. Second, we tested if loss of CycA binding impacted Meru's role in spindle orientation in vivo. We performed a rescue experiment by expressing *UAS-mK2-meru* constructs in a *meru*<sup>1</sup> mutant background and quantified cortical Mud localisation as in wild-type SOPs above (Fig. 6B–D). In the rescued conditions, both the anterior and posterior cortices had nearly identical levels of Mud when mK2-Meru<sup>WT</sup> was expressed (Fig. 6B,B',D). It is interesting to note that, in the absence of Meru overexpression, Mud is more enriched at the anterior than the posterior cortex (Fig. 3C). The equal cortical distribution of Mud upon mK2-Meru<sup>WT</sup> overexpression is therefore consistent with Meru promoting Mud posterior recruitment. In contrast, mK2-Meru<sup>RxL</sup>-expressing SOPs had reduced Mud levels at the posterior cortex (Fig. 6C,C',D), suggesting that CycA binding of Meru is necessary for posterior Mud localisation. We also note that, while Meru<sup>WT</sup> is enriched at the posterior cortex (Fig. 6B'), we consistently detect some Meru at the anterior cortex (see Fig. 4A for another example). This could be a consequence of overexpression or of the way in

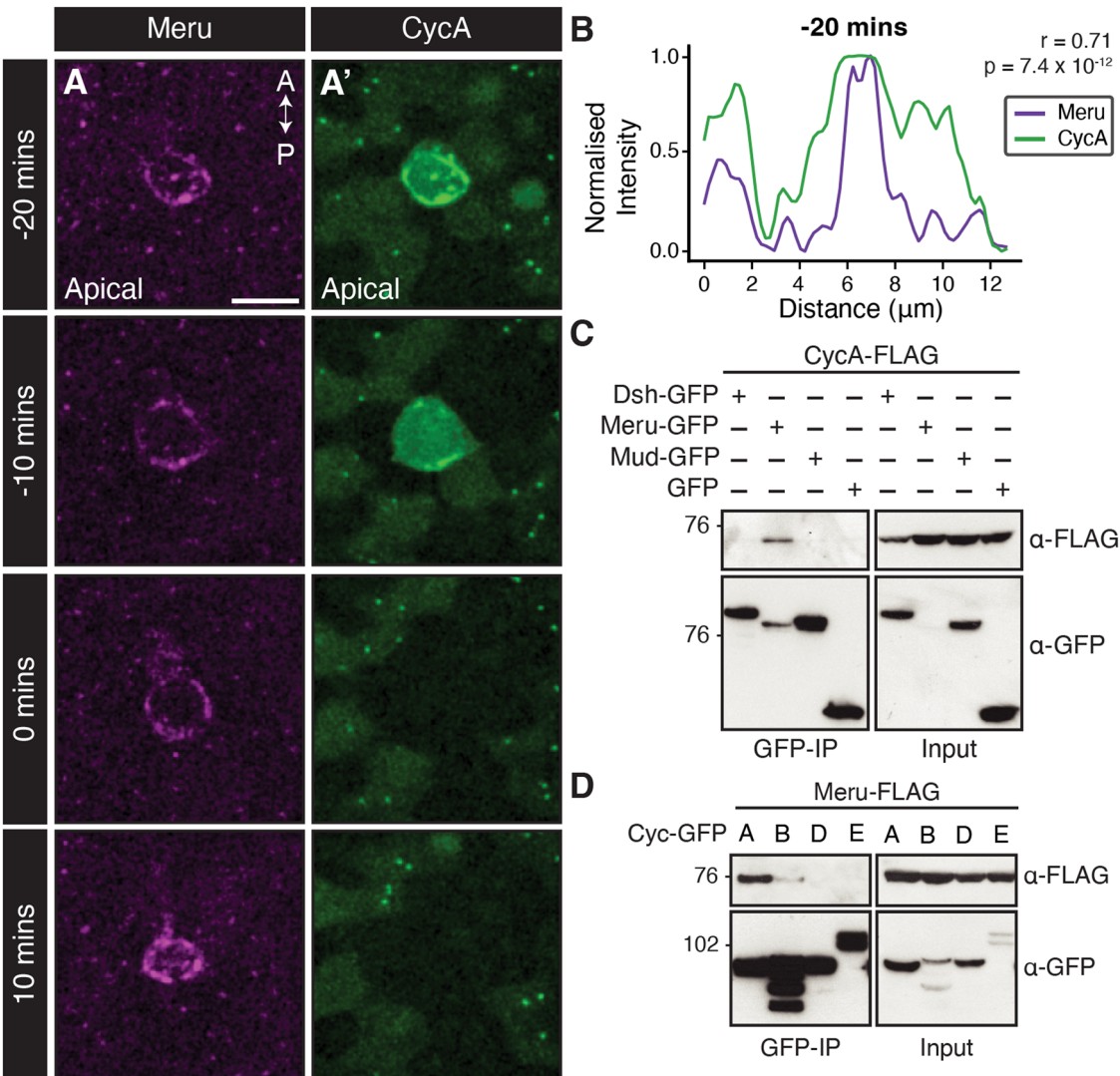

**Figure 4. Meru and CycA interact at the posterior cortex of SOPs.**

(A) Confocal live-imaging of pupal nota (apical-most three sections) expressing *neurG4 > UAS-mK2-meru* (magenta) and *CycA-GFP* (green) of an SOP during mitosis at 16 h APF and 0 min indicating the first frame in cytokinesis. (B) Single brightest slice in (A, A') at −20 min plotted as line graph by measuring the grey value across the posterior domain as defined by Meru localisation. High Pearson's coefficient (*r*) indicates a positive correlation. *P* value calculated using a two-tailed test. (C, D) Western blots of co-IP experiment using cell lysates from transfected S2 cells, immunoprecipitated using α-GFP beads and probed using α-FLAG and α-GFP antibodies. (C) Meru, but not Dsh or C-terminal Mud (1452–1961), immunoprecipitates with CycA. (D) Meru associates with CycA and weakly with CycB, but not CycD or CycE. Scale bar = 10 μm. Source data are available online for this figure.

which Meru is recruited to the plasma membrane (see "Discussion" for details).

Finally, we analysed the impact of impaired CycA binding on Meru-dependent spindle alignment. mK2-Meru^WT expression was sufficient to rescue A–P spindle alignment in the *meru[1]* mutant, however mK2-Meru^RxL expression did not rescue A–P spindle angles (Fig. 6E–G), though A–B spindle alignment was not significantly affected (Fig. 6G'). As the A–P spindle defect in the Meru^RxL-expressing animals (Fig. 6G) is less pronounced than in *meru[1]* mutants (Fig. 3F), it is possible that some residual CycA binding occurs in this mutant through the other four RxL motifs when overexpressed in vivo. Thus, CycA docking on Meru is required for normal posterior Mud recruitment and consequently SOP mitotic spindle alignment.

## Meru Cdk1 phosphorylation sites modulate binding to Mud and Dsh

The best characterised CycA function is Cdk1 regulation, thus we investigated whether Meru phosphorylation by Cdk1 could affect Meru/Mud association. Upon CycA/Cdk1 co-expression in S2 cells, we observed a marked increase in Meru/Mud binding compared to basal levels (Fig. EV4A). The minimal Cdk1 phosphorylation consensus site is S/T-P, while the optimal consensus is S/T-P-x-K/R (Enserink and Kolodner, 2010). We identified five potential phosphorylation sites within the Meru sequence (Fig. 5A). To determine whether these sites are indeed phosphorylated by Cdk1/CycA we carried out proteomic analysis by Mass Spectrometry (MS) to compare samples expressing either Meru^WT or Meru^RxL in the presence of Cdk1 and CycA. First, Meru, Dsh,

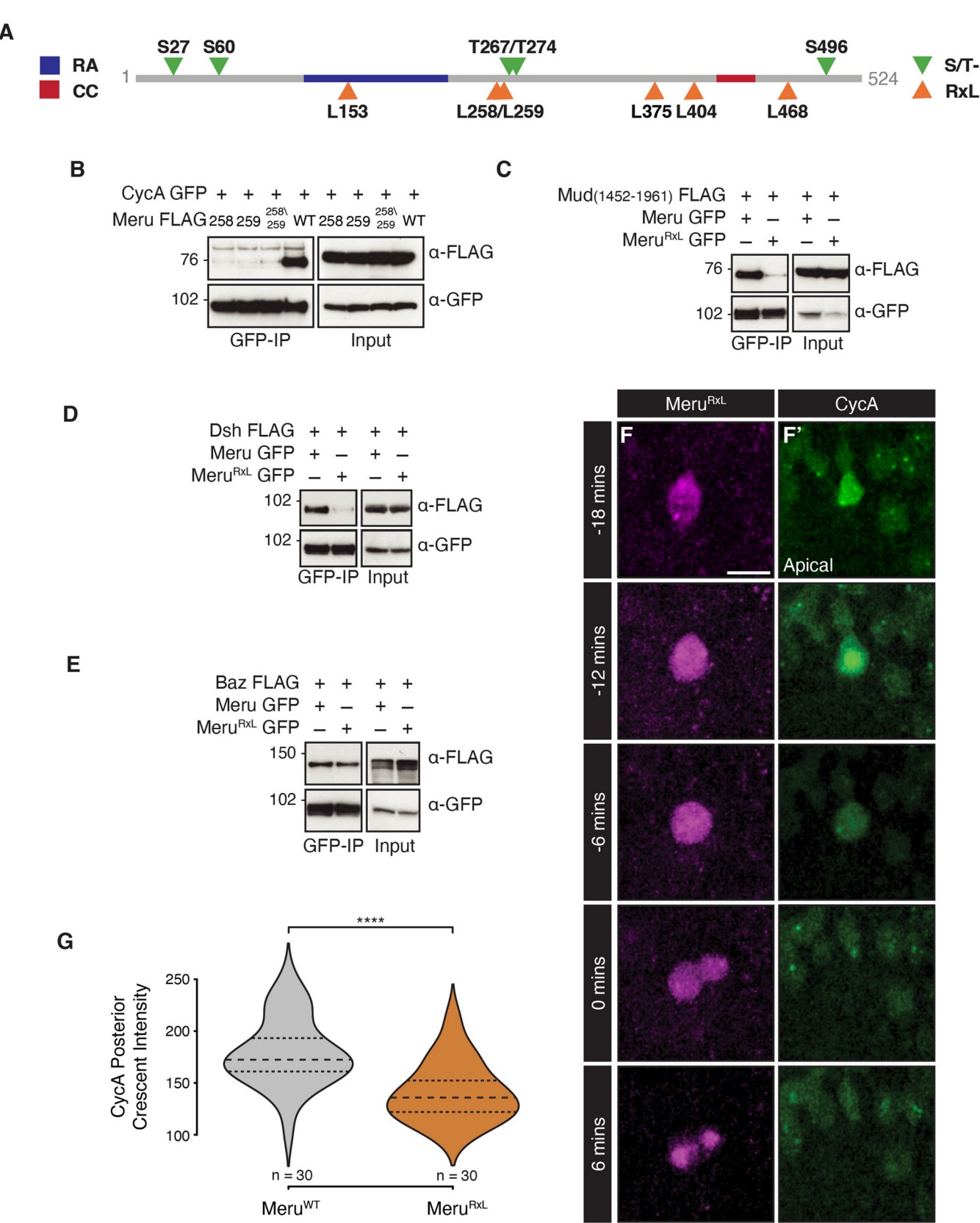

**Figure 5. CycA docking is required for Meru localisation in vivo.**

(A) Schematic of RxL motifs and Cdk1 S/T-P phosphorylation sites present in the Meru sequence. (B) Meru mutated at RxL motifs 258, 259 and 258/9 no longer immunoprecipitates with CycA. (B–E) Western blots of co-IP experiment using cell lysates from transfected S2 cells, immunoprecipitated and probed using the indicated antibodies. Mutation of Meru RxL 258 and 259 sites disrupt the interaction between Meru and CycA (B), Mud (C) and Dsh (D), but not Baz (E). (F) Maximum intensity projections from confocal live-imaging of pupal nota at 16 h APF (0 min marking first frame in cytokinesis) show expression of *UAS-mK2-meru* (magenta) driven by *neurG4* and *CycA-GFP* (green) during SOP mitosis. Scale bar = 10 μm. (G) Graphs showing the intensity of apical–posterior CycA crescent in Meru$^{WT}$ and Meru$^{RxL}$ animals at late G2. CycA cortical levels were significantly lower in Meru$^{RxL}$ relative to Meru$^{WT}$ cells ($P = 2.6 \times 10^{-5}$). Large and small dashed line represent the median and Q1/Q3, respectively. ****$P < 0.0001$ using a Mann–Whitney $U$ test. Number of SOPs imaged indicated on the panels from three nota. Source data are available online for this figure.

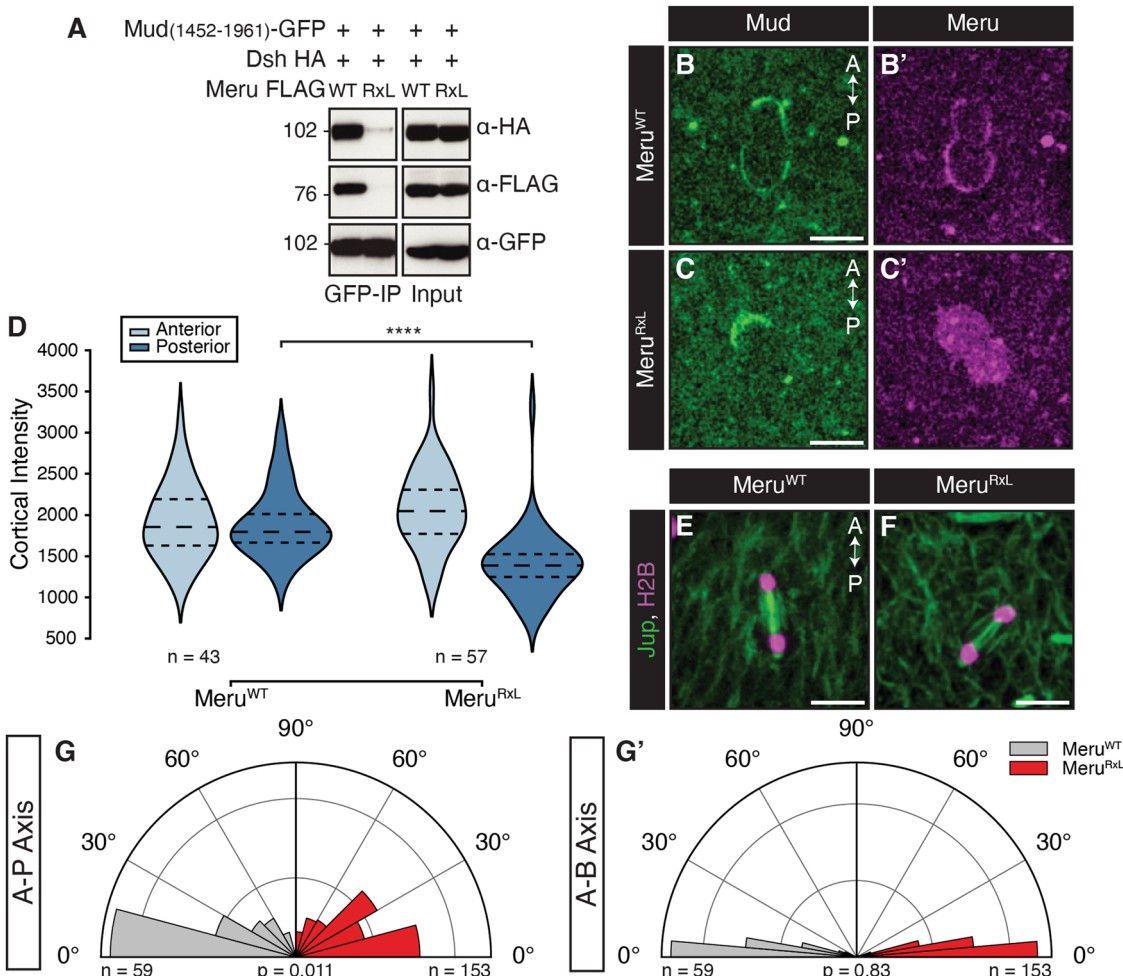

**Figure 6. Mutation of the Meru RxL motif results in decreased posterior Mud and abnormal spindle alignment.**

(A) Western blot of co-IP experiment using cell lysates from transfected S2 cells, immunoprecipitated and probed using the indicated antibodies. Dsh does not immunoprecipitate with C-terminal Mud in the presence of Meru$^{RxL}$. (B, C) Maximum intensity projections (6 slices) from confocal live-imaging of pupal nota at 16 h APF expressing *Mud-GFP* (green) and *neurG4 > UAS-mK2-meru$^{WT}$* (B) or *neurG4 > UAS-mK2-meru$^{RxL}$* (magenta) (C) in a *meru$^1$* background. (D) Graph showing the intensity of cortical Mud at the anterior and posterior crescent of each genotype. Measurements were taken at anaphase. Posterior Meru$^{RxL}$ is significantly lower than Meru$^{WT}$ ($P = 1.9 \times 10^{-8}$). ****$P < 0.0001$ using a Mann–Whitney $U$ test. Large and small dashed line represent the median and Q1/Q3, respectively. (E, F) Maximum intensity projections from confocal live-imaging of *neur-H2B-RFP* (magenta) and *Jupiter-GFP* (green) in a *neur > UAS-mK2-meru$^{WT}$* (E) and *neur > UAS-mK2-meru$^{RxL}$* background (F). (G) Graphs showing the spindle orientation of each genotype relative to the dorsal midline (G) and epithelial plane (G'). $P$ values indicated for Kolmogorov–Smirnov tests. Scale bars = 10 μm. Number of SOPs imaged from three animals indicated on the panels from three animal. Source data are available online for this figure.

CycA, Cdk1 and Mud proteins are all enriched in the Meru$^{WT}$ sample compared to the Meru$^{RxL}$ sample (Fig. EV4B), consistent with our previous data. Our analyses demonstrate the presence of phosphorylation on T496 and S60 in the Meru$^{WT}$ sample (Fig. 7B), the extracted spectrum for the T496 site is shown in Fig. 7A. In contrast, we failed to detect any phosphorylation in the Meru$^{RxL}$ sample, consistent with CycA/Cdk1 dependency of Meru phosphorylation of these sites (Fig. 7B).

In co-IPs, although several phosphosite mutants affected the Meru/ Mud association, no single mutation was sufficient to abolish the interaction (Fig. EV4C). However, when all five sites were mutated

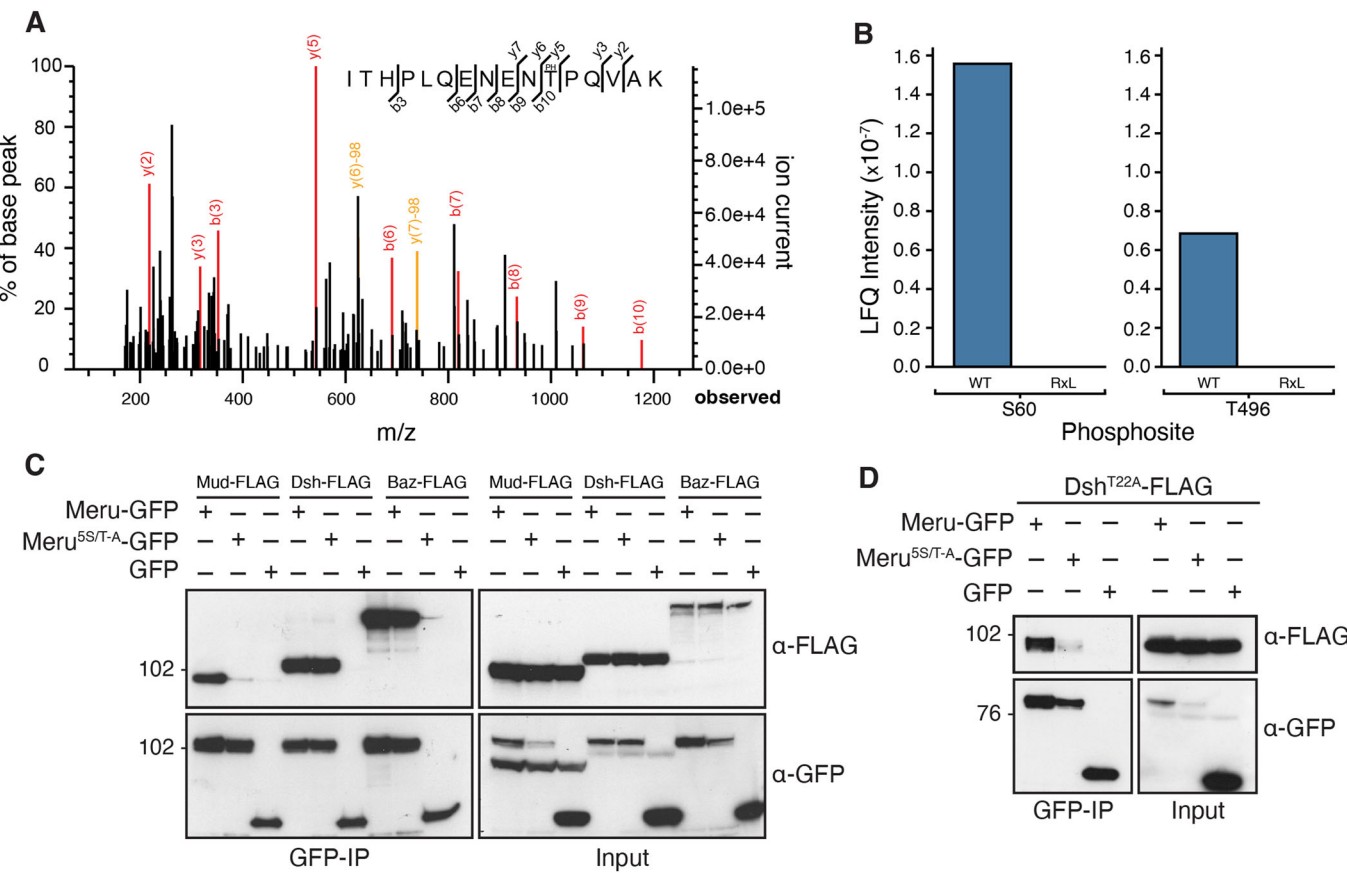

**Figure 7. Cdk1 phosphorylation sites are required for Meru binding to Mud and phospho-Dsh.**

(A) MS/MS spectrum corresponding to modification (phosphorylation) to the Meru T496 site. Fragment ions are indicated in red. Fragment ions with neutral loss are in yellow. (B) Phosphorylation of Meru at S60 and T496 was detected in Meru[WT] but not in the Meru[RxL] mutant. Label-free quantification (LFQ) intensities were plotted for each sample and phosphorylation site. (C, D) Western blots of co-IP experiment using cell lysates from transfected S2 cells, immunoprecipitated and probed using the indicated antibodies. Meru mutated to Alanine at five S/T-P phosphorylation sites disrupts co-IP with C-terminal Mud (1452–1961), but not Baz or Dsh (C), unless the Dsh T22 phosphorylation site was also mutated (D). Source data are available online for this figure.

(Meru[5S/T-A]) Meru/Mud binding was strongly reduced (Fig. 7C). In contrast, the Meru[5S/T-A] mutant still associated with both Baz and Dsh (Fig. 7C). This agrees with Meru not requiring a functional RxL site to interact with Baz (Fig. 5E). The Meru/Dsh interaction, on the other hand, is dependent on Meru's RxL motif. Thus, we explored the possibility that Dsh is also undergoing Cdk1/CycA phosphorylation. Examining the Dsh sequence, we found a single Cdk1 phosphosite at amino acid 22 (T22) within the DIX (DIshevelled and aXin) domain (Fig. 2A). Interestingly, mutating this site alone (Dsh[T22A]) had no impact on binding to wild-type Meru (Fig. 7D). However, when the Dsh[T22A] mutant was combined with the Meru[5S/T-A] mutant, the association was almost completely lost (Fig. 7D). Taken together, this suggests that Meru phosphorylation by Cdk1 is required for Meru/Mud binding, while Meru/Dsh assembly is dependent on phosphorylation of both proteins.

## Meru Cdk1 phosphorylation sites are required for proper spindle tethering

To evaluate the impact of Cdk1 phosphorylation of Meru on spindle alignment, we expressed Meru[5S/T-A] under *neurG4* control in a *meru[1]* background. First, we compared the localisation of Meru[5S/T-A] with

Meru[WT] to the posterior cortex in late G2 (Fig. EV5A–B'). Although unlike Meru[RxL] (Fig. 5F), Meru[5S/T-A] could still form a detectable posterior crescent (Fig. EV5B,B'), this was significantly reduced compared with Meru[WT] (Fig. EV5A'–C). In addition, we observed a reduction in the characteristic Meru puncta that forms at the posterior crescent at this stage (Fig. EV5A',B',D). In contrast to *meru[1]* mutants (Fig. 3B,C) and Meru[RxL] flies (Fig. 6C,D) Meru[5S/T-A]-expressing SOPs did not display a significant change in cortical Mud levels at anaphase (Fig. EV5H–J). However, there was a clear reduction in Mud at the posterior cortex in Meru[5S/T-A] animals at both prophase (Fig. EV5E–G) and metaphase (Fig. 8A–C), indicating a delay in Mud posterior recruitment. A–P spindle alignment was perturbed in the Meru[5S/T-A] mutants (Fig. 8D–F), while A–B alignment was not detectably affected (Fig. 8F'), as in Meru[RxL] animals (Fig. 6G'). Together, this suggests that Cdk1 phosphorylation of Meru is necessary to correctly orient the mitotic spindle in SOPs.

## The RRLL motif, a Cyclin-docking site shared by several spindle regulators

The cyclin-binding SLiM we identified in the Meru sequence is striking as it is comprised of two overlapping RxL motifs at amino acids 258/

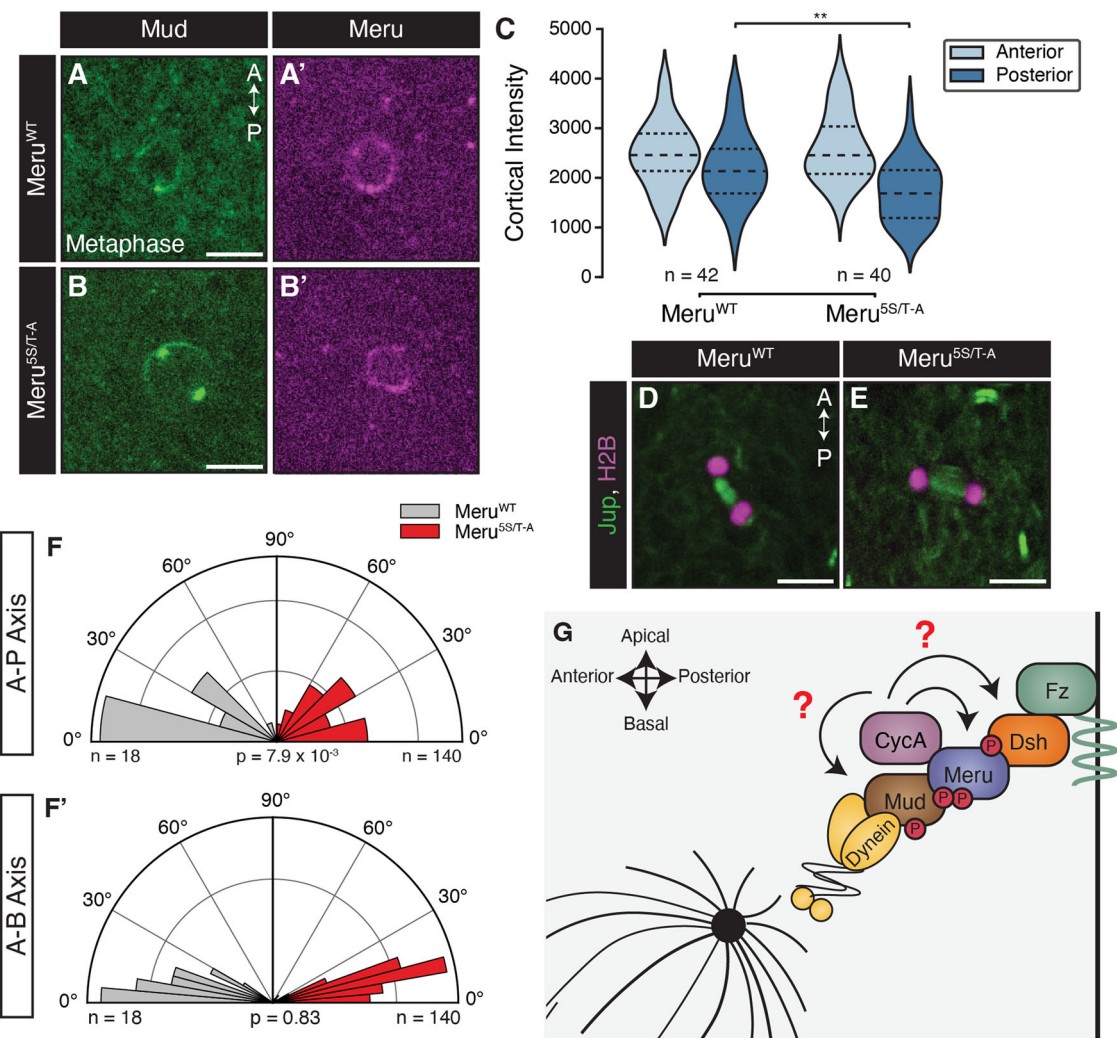

**Figure 8. Meru Cdk1 phosphorylation sites promote Mud posterior recruitment and SOP spindle orientation.**

(A, B) Maximum intensity projections (6 slices) of confocal live-imaging of pupal nota at 16 h APF expressing *Mud-GFP* (green) and *neurG4 > UAS-mK2-meru^WT* (A) or *neurG4 > UAS-mK2-meru^SS/T-A* (magenta) (B) in a *meru^1* background at metaphase (C) Graph showing the intensity of cortical Mud at the anterior and posterior crescent of each genotype. At metaphase, posterior Mud is significantly lower in Meru^SS/T-A than Meru^WT animals ($P = 2.24 \times 10^{-3}$). **$P < 0.01$ using a Mann–Whitney $U$ test. Large and small dashed line represent the median and Q1/Q3, respectively. (D, E) Confocal live-imaging of *neur-H2B-RFP* (magenta) and *Jupiter-GFP* (green) in a *neur > UAS-mK2-meru^WT* (D) and *neur > UAS-mK2-meru^SS/T-A* background (E). (F) Graphs showing the spindle orientation of each genotype relative to the dorsal midline (F) and epithelial plane (F'). *P* values indicated for Kolmogorov–Smirnov tests. (C, F, F') The number of SOPs imaged indicated on the panels from three nota. (G) Diagram showing our proposed model for Meru function in SOP spindle tethering. Meru, which is recruited to the posterior cortex of SOPs by Dsh, promotes posterior Mud recruitment. The Meru/Mud interaction is dependent on the docking of CycA to Meru at the 258/9 RxL motif and is regulated by Cdk1 phosphorylation of Meru. Mud and Dsh may also be Meru-dependent CycA/Cdk1 substrates at the SOP posterior cortex. Data information: Scale bars in (A, B, D, E) = 10 µm. Source data are available online for this figure.

259 (Fig. EV2A). In addition, this site is followed by two optimal Cdk1 consensus phosphorylation sites at amino acids 267 and 274 (Fig. 5A). We used this motif to search for proteins containing overlapping RxLs separated by 7 amino acids from a Cdk1 target site ([RK]-[RK]-L-L-x[7]-[ST]-P, which we termed RRLL motif). Remarkably, we found that Pins, which recruits Mud to the anterior cortex in SOPs and in many other contexts (Bergstralh et al, 2017; di Pietro et al, 2016; Lechler and Mapelli, 2021), possesses a matching site (Fig. EV6A). When we mutated this RRLL motif in Pins, we observed a mislocalisation to the nucleus and delayed localisation to the anterior cortex during SOP mitosis (Fig. EV6B,C). This could be due to competition between cortical localisation and nuclear import of Pins

prior to nuclear envelope breakdown. Consistent with this idea, we observed a small amount of nuclear Pins upon expression of wild-type Pins (Fig. EV6B, −24 min and −18 min time points), and loss of the anterior PCP component Vang leads to some Pins nuclear accumulation (Gomes et al, 2009). As Meru and Pins bind Mud through partially overlapping domains (Fig. 2B), this highlights strong mechanistic parallels between Meru and Pins in terms of Mud recruitment. Pins phosphorylation on S436 by Aurora A increases binding to Dlg (Johnston et al, 2009), suggesting that the Dlg/Pins/NuMA complex is also under tight regulation by the mitotic kinases. Indeed, the Dlg binding partner Guk-holder, which has been implicated in spindle tethering in neuroblasts (NBs or neural stem

cells) and S2 cells with induced polarisation (Albertson and Doe, 2003; Garcia et al, 2014; Golub et al, 2017), also has an RRLL motif (Fig. EV6A). We found RRLL motifs in other proteins involved in spindle regulation (Aurora borealis (Hutterer et al, 2006), Sister of feo (Vernì et al, 2004)) or cell cycle progression (Double parked (Whittaker et al, 2000)), suggesting it can be used as a predictor of regulation by Cdks (Fig. EV6A).

# Discussion

To build and maintain tissues of the appropriate architecture, cells must read external cues from the extracellular matrix and other cells to align their polarity along (PCP) and across (A–B polarity) the tissue plane (Buckley and St Johnston, 2022; Butler and Wallingford, 2017). A key aspect of tissue organisation is the orientation of the mitotic spindle, which is essential not only to determine the position of the daughter cells after division, but also for cell fate determination through asymmetric cell division (Bergstralh et al, 2017; di Pietro et al, 2016; Lechler and Mapelli, 2021). Although the core PCP and A–B polarity pathways are well conserved, cells adopt radically different polarised organisations and cell division orientations to give rise to the diversity of adult tissues, from the epidermis and intestine to the nervous system (Buckley and St Johnston, 2022; Butler and Wallingford, 2017). Understanding how cells use these conserved polarity complexes as landmarks to generate cell type-specific polarity and spindle orientation remains a key challenge in the field, for which few mechanisms and molecular players have been identified. One of the best-understood tissue-specific polarity adaptors is Inscuteable (Insc), which was first identified in *Drosophila* NBs (Kraut et al, 1996; Kraut and Campos-Ortega, 1996). In NBs, Insc connects the apical polarity protein Baz with the spindle tethering machinery via Pins, which promotes spindle orientation perpendicular to the tissue plane, allowing the differentiating NB progeny, the ganglion mother cells, to delaminate (Kraut et al, 1996; Schober et al, 1999; Wodarz et al, 1999; Yu et al, 2000).

We had previously shown that Meru, which is transcriptionally activated in SOPs as part of their differentiation programme (Banerjee et al, 2017; Buffin and Gho, 2010; Reeves and Posakony, 2005), links PCP with A–B polarity by promoting Baz planar polarisation (Banerjee et al, 2017). Here, we show that Meru also couples cell polarity with spindle orientation by assembling a ternary complex with the PCP protein Dsh and the spindle tethering factor Mud (Fig. 2D), promoting Mud localisation to the posterior cortex to orient the SOP spindle along the A–P axis (Fig. 3). In agreement with this model, Mud is depleted from the posterior cortex in *meru* mutants (Fig. 3A–C) and the spindle is no longer oriented along the A–P axis (Fig. 3D–F), as in *fz* and *dsh* mutants (Bellaïche et al, 2001; Gho and Schweisguth, 1998; Ségalen et al, 2010). Unexpectedly, despite the enrichment of Meru at the posterior cortex, cortical Mud levels were reduced both anteriorly and posteriorly in *meru* mutant SOPs (Fig. 3C). This could indicate that Meru is also required for Mud stability and in its absence, the overall Mud pool is depleted. This is difficult to verify as low Mud expression precludes reliable measurement of its cytosolic levels. However, in our rescue experiments, Meru$^{WT}$ overexpression in a *meru* mutant background causes Mud to be deposited equally at the anterior and posterior poles (Fig. 6B,D), in contrast to wild-type SOPs, where we detect more Mud anteriorly than posteriorly (Fig. 3C). Together with the fact that Meru$^{RxL}$ expression causes loss of Mud specifically at the posterior pole (Fig. 6C,D), this strongly argues in favour of our model that Meru promotes posterior Mud recruitment. While Meru is enriched at the

posterior SOP cortex, we detect some signal at the anterior pole, especially upon GAL4/UAS-driven overexpression (Fig. 6B'). This could be because overexpressed Meru saturates endogenous Dsh, therefore spilling over to the anterior side, or may indicate a two-step Meru cortical recruitment process where Meru is first recruited to the plasma membrane independently of Dsh, which then biases Meru cortical localisation by preferentially retaining it posteriorly.

Could RASSF proteins such as Meru be implicated in spindle orientation in other contexts? Since the Meru mammalian orthologs RASSF9 and RASSF10 associate with Dvl proteins (Hauri et al, 2013), it would be interesting to test their involvement in oriented cell divisions in contexts in which Dvl1-3 have been implicated, such as HeLa cells (Kikuchi et al, 2010; Yang et al, 2014) and zebrafish gastrulation (Ségalen et al, 2010). Furthermore, similar to *Drosophila* NBs, mouse Inscuteable (mInsc) participates in orientating the spindle perpendicular to the tissue plane to promote differentiation of embryonic epidermal progenitors (Lechler and Fuchs, 2005; Poulson and Lechler, 2010; Williams et al, 2014). As RASSF9 is highly expressed in epidermal keratinocytes and *RASSF9* mutant animals show increased proliferation and aberrant differentiation in the epidermis (Lee et al, 2011), it is exciting to speculate that RASSF9 and mInsc could both regulate spindle orientation in this tissue. In *Drosophila*, the Meru paralog RASSF8 is expressed in many epithelial tissues (Langton et al, 2009), therefore it would also be interesting to investigate its role and that of its mammalian orthologs RASSF7 and RASSF8 in symmetric cell division orientation.

Besides positioning, a second important aspect of spindle orientation is temporally linking progression through the cell cycle with spindle tethering (Bergstralh et al, 2017; di Pietro et al, 2016; Lechler and Mapelli, 2021). Several cell polarity and spindle tethering factors such as Lethal giant larvae, Dvl and NuMA, are regulated through phosphorylation by mitotic kinases, supporting the idea of an intimate relationship between spindle orientation and the cell cycle (di Pietro et al, 2016; Lechler and Mapelli, 2021; Osswald and Morais-de-Sá, 2019). We identified a mechanism whereby Mud cortical localisation is coupled to the cell cycle via Meru phosphorylation, linking cell polarity, the cell cycle and spindle orientation. Darnat et al showed that the Cdk1 partner CycA is recruited to the SOP posterior cortex in a Dsh-dependent manner (Darnat et al, 2022). We find that Meru, but not Dsh, associates with CycA in S2 cells (Fig. 4C). We uncovered in Meru a pair of overlapping Cyclin-binding RxL SLiMs (which we termed the RRLL motif) that promote Dsh/Meru and Mud/Meru binding (Fig. 5C,D) and Meru cortical recruitment (Fig. 6C'). Furthermore, both CycA docking and Cdk1 phosphorylation of Meru promote the localisation of Mud to the posterior cortex (Figs. 6C,D and 8B,C), as well as A–P spindle orientation (Fig. 6F,G). Mutating two of the Meru RxL motifs (amino acids 258/9) is sufficient to disrupt Meru function, but the spindle orientation phenotype of Meru$^{RxL}$-expressing flies (Fig. 6G,G') is slightly weaker than that of the *meru$^1$* mutant (Fig. 3F,F'), suggesting either that the remaining RxL motifs can provide some functional rescue in vivo, or that CycA binding strongly increases the affinity of Meru for Dsh and Mud, but is not absolutely required for it. We therefore propose a model in which, following its posterior recruitment via Dsh, Meru recruits CycA/Cdk1, which in turn phosphorylates Meru, promoting its stabilisation at the posterior cortex and Mud recruitment (Fig. 8G). Consistent with this model, Meru$^{RxL}$ is impaired in its ability to mediate the assembly of the ternary Dsh/Meru/Mud complex (Fig. 6A), and a form of Meru mutant for its five Cdk1 consensus sites (Meru$^{5S/T-A}$) displays much reduced Mud binding (Fig. 7C).

Does CycA/Meru association promote the phosphorylation of other posterior cortical proteins? Our data indicate that Dsh may be another target, since mutation of the only Cdk1 consensus site in Dsh causes reduced binding to Meru[5S/T-A] (Fig. 7D), suggesting phosphorylation of both partners may be required for optimal binding. This may account for the fact that Meru[RxL] displays a defect in Mud recruitment at anaphase (Fig. 6C,D) whereas the Meru[5S/T-A] mutant has delayed Mud recruitment but no detectable defect at anaphase (Figs. 8B,C and EV5E–J), as other Meru partners such as Dsh would still be phosphorylated by CycA/Cdk1 in this mutant, partially rescuing complex formation. It is interesting to note that human Dvl2 phosphorylation on T206 by Plk1 is required for spindle orientation in cultured human cells (Kikuchi et al, 2010), therefore Dsh/Dvl proteins may receive inputs from different cell cycle kinases. Finally, it is possible that Mud is also phosphorylated by Cdk1 in a Meru-dependent manner. Indeed, Cdk1 phosphorylates NuMA on four C-terminal residues to control its cortical turnover in human cells and *C. elegans*, at least in part by inhibiting a NuMA lipid-binding domain (Compton and Luo, 1995; Gehmlich et al, 2004; Hsin-Ling and Ning-Hsing, 1996; Kiyomitsu and Cheeseman, 2013; Kotak et al, 2013; Portegijs et al, 2016; Saredi et al, 1997; Seldin et al, 2013; Zheng et al, 2014).

In summary, we uncover a mechanism whereby Meru allows coupling of the spindle tethering complex both to the cell polarity machinery and cell cycle progression. We propose that similar regulatory logics underpin the diverse spindle orientation mechanisms employed by other symmetrically and asymmetrically dividing cells.

# Methods

### Reagents and tools table

| Reagent/resource | Reference or source | Identifier or catalogue number |
|---|---|---|
| **Experimental models** | | |
| w[1118]; P{w[+mW.hs] =GawB}neur[GAL4-A101] Kg[V]/TM3, Sb[1] | Bloomington *Drosophila* stock center | 6393 |
| y[1] w[*]; TI{TI}shg[GFP] | Bloomington *Drosophila* stock center | 60584 |
| FlyFos028184(pRedFlp-Hgr)(dlg1[37834]::2XTY1-SGFP-V5-preTEV-BLRP-3XFLAG)dFRT | Vienna *Drosophila* Resource Center | 318133 |
| w;; GFP::mud(62E1), GFP::mud(65B2) | Gift from Yohanns Bellaïche Ségalen et al, 2010 https://doi.org/10.1016/j.devcel.2010.10.004 | |
| w;; UAS-Pon-GFP | Gift from Yohanns Bellaïche Lu et al, 1999 https://doi.org/10.1016/s1097-2765(00)80218-x | |
| y[1] w[*]; P{w[+mC] =UAS-mCD8.mRFP.LG} 18a | Bloomington *Drosophila* stock center | 27398 |
| w;; neur-H2B-RFP | Gift from François Schweisguth Gomes et al, 2009 https://doi.org/10.1371/journal.pone.0004485 | |
| y[1] v[1]; P{y[+t7.7] v[+t1.8]=TRiP.HMJ21351} attP40 | Bloomington *Drosophila* stock center | 53968 |
| y[1] w[1118]; P{w[+mC] =PTT-GA} Jupiter[ZCL0931]/TM3, Ser[1] | Bloomington *Drosophila* stock center | 6825 |
| w;; meru[1] | Banerjee et al, 2017 https://doi.org/10.7554/eLife.25014 | |

| Reagent/resource | Reference or source | Identifier or catalogue number |
|---|---|---|
| w;; CycA-GFP | Gift from Michel Gho and Agnes Audibert Darnat et al, 2022 https://doi.org/10.1038/s41467-022-30182-1 | |
| [1] cv[1] v[1] fs(1) M29[A151] f[1] car[1]/ FM7a | Bloomington *Drosophila* stock center | 4716 |
| w; UAS-mK2-Meru | This paper | |
| w; UAS-mK2-Meru[RxL] | This paper | |
| w; UAS-mK2-Meru[SS/T-A] | This paper | |
| w; UAS-mK2-Pins | This paper | |
| w; UAS-mK2-Pins[RxL] | This paper | |
| S2 cells | *Drosophila* Genomics Resource Center | RRID:CVCL_Z232 |
| **Recombinant DNA** | | |
| *Drosophila* Gateway Vector Collection | https://carnegiescience.edu/bse/drosophila-gateway-vector-collection | |
| **Antibodies** | | |
| Mouse anti-FLAG | Sigma | F1804 |
| Rabbit anti-Flag | Sigma | |
| Mouse anti-Myc | Santa Cruz Biotechnology | sc-40 |
| Rabbit anti-HA | Cell Signalling | C29F4 |
| Rat anti-GFP | ChromoTek | 3h9 |
| Rabbit IgG HRP-Linked F(ab')2 | Sigma | NA9340 |
| Mouse IgG HRP-Linked F(ab')2 Fragment | Sigma | NA9310 |
| Rat IgG, HRP-linked whole antibody | Sigma | NA935 |
| **Oligonucleotides and other sequence-based reagents** | | |
| Dsh pCdk1mt fw | CCACATCGACGATGAGACGg CGCCGTATCTGGTGAAGATCCCC | |
| Dsh pCdk1mt rev | GGGGATCTTCACCAGATACGGCG cCGTCTCATCGTCGATGTGG | |
| Pins RxL 390/91 fw | GTGAACATCTCCGATCTAgcAgcGgc AgcCGGAATGCCCGACTCCG | |
| Pins RxL 390/91 rev | CGGAGTCGGGCATTCCGgcTgcCgcT gcTAGATCGGAGATGTTCAC | |
| Meru pCdk1mt 267 fw | GGGTAAAAAGgCGCCACCAAAACCC ACAAAAACGCCTCCAAAAGTGCCGAAC | |
| Meru pCdk1mt 267 rev | GTTCGGCACTTTTGGAGGCGTTTTTGT GGGTTTTGGTGGCGcCTTTTTACCC | |
| Meru pCdk1mt 274 fw | GGGGTAAAAAGACGCCACCAAAACCC ACAAAAgCGCCTCCAAAAGTGCCGAAC | |
| Meru pCdk1mt 274 rev | GTTCGGCACTTTTGGAGGCGcTTTTGT GGGTTTTGGTGGCGTCTTTTTACCC | |
| Meru pCdk1mt 27 fw | CGCTATTGAATCTACCCGAAgcTCCCA TTCTTCCGCCGCGAC | |
| Meru pCdk1mt 27 rev | GTCGCGGCGGAAGAATGGGAgcTTCG GGTAGATTCAATAGCG | |
| Meru pCdk1mt 60 fw | GGAGCGTTTGGTGGATGAAgCGCCATC GGTGGGGGAGCCAC | |
| Meru pCdk1mt 60 rev | GTGGCTCCCCCACCGATGGCGcTTCATC CACCAAACGCTCC | |
| Meru pCdk1mt 496 fw | CCTTACAGGAAAATGAAAATgCCCCTCA AGTTGCTAAAAATACGC | |
| Meru pCdk1mt 496 rev | GCGTATTTTTAGCAACTTGAGGGGcATTT TCATTTTCCTGTAAGG | |
| Meru Rxl 258/9 | CGATGTACAAGTGGCTCgcGgcGgcAgcGC ATCTGAAAAAGGG | |
| Meru Rxl 258 | CGATGTACAAGTGGCTCgctAAGgctTTGC ATCTGAAAAAGGG | |
| Meru Rxl 259 | CGATGTACAAGTGGCTCAAGgcGCTAgcG CATCTGAAAAAGGG | |
| Meru Rxl 154 | CGTGCGATGACATAATCgcGGCGgcGATTG ATGACGAACTGCG | |

| Reagent/resource | Reference or source | Identifier or catalogue number |
|---|---|---|
| Meru Rxl 375 | CATAAGGAGGCGAAAGGATgcGCCCgcGCG AAATAGTGTACG | |
| Meru Rxl 404 | GGAACACGCTCTAACAgcTCAGgcATCGGAA ATGTGCCGACTG | |
| Meru Rxl 468 | CGCTGATTAATAACTTAgcGCGTgcGACGTTG GAGGAATCGGAGG | |
| CycA RNAi fw | TAATACGACTCACTATAGGGCGGTGCTGGGC AGAAGGAGCTGG | |
| CycA RNAi rev | TAATACGACTCACTATAGGGCTCGCTCTCC CGGAAATATTCCAG | |
| **Chemicals, enzymes and other reagents** | | |
| S-trap (Micro) | Protifi | CO2-micro |
| Trypsin | Promega | 90057 |
| Trifluoracetic acid | Thermo Fisher Scientific | LS119-4 |
| Acetonitrile | Thermo Fisher Scientific | A955-212 |
| TEAB | Merck | 18597-100 ML |
| Effectene Transfection Reagent | Qiagen | 301425 |
| Pierce ECL Plus Western Blotting Substrate | Thermo Scientific | 32132 |
| ChromoTek GFP-Trap® Agarose | ChromoTek | gta |
| ChromoTek Myc-Trap® Agarose | ChromoTek | yta |
| Phosphatase Inhibitor Cocktail 2 | Sigma | P5726 |
| Phosphatase Inhibitor Cocktail 3 | Sigma | P0044 |
| cOmplete™, Mini, EDTA-free Protease Inhibitor Cocktail | Roche | 11836170001 |
| MEGAscript™ T7 Transcription Kit | Ambion | AMB13345 |
| Taq PCR Master Mix | Qiagen | 201445 |
| Schneider's Drosophila Medium | Gibco | 11590576 |
| Penicillin–Streptomycin | Thermo Fisher Scientific | 15070063 |
| Foetal Bovine Serum | Thermo Fisher Scientific | Batch 2503539 |
| Gateway LR Clonase II Enzyme mix | Invitrogen | 10134992 |
| pENTR™/D-TOPO™ Cloning Kit | Invitrogen | 15575730 |
| Quikchange Multi Site-Directed Mutagenesis Kit | Agilent Technologies LDA UK Limited | 200515 |
| **Software** | | |
| Fiji | https://doi.org/10.1038/nmeth.2019 | http://fiji.sc |
| Python, Numpy, Scipy | Python Software Foundation | https://www.python.org https://numpy.org https://www.scipy.org |
| Maxquant | Juergen Cox lab https://doi.org/10.1038/nprot.2016.136 | v2.6.2.0 https://maxquant.org/maxquant/ |
| Perseus | Juergen Cox lab https://doi.org/10.1007/978-1-4939-7493-1_7 | v1.6.14.0 https://maxquant.org/perseus/ |
| **Other** | | |

## Transgenes and fly stocks

*UAS-mK2-meru* transgenic lines were created by phiC31-mediated recombination using stock 9723 from Bloomington (as described in

Banerjee et al, 2017). The *meru^RxL^* and *meru^5S/T-A^* mutant constructs were generated under the same protocol but the amino acids sites 258-261 or S27, S60, T267, T274, and S496 were mutated to Alanines in the RxL or 5S/T-A motifs, respectively. The Pins wild-type and RxL mutant were generated by replacing Meru from the construct above with Pins cDNA and mutating the RxL motif at amino acids 390/1 to Alanines at sites 390–393.

The following fly stocks were used: *neurG4* (Bellaïche et al, 2001); *Ecad-GFP* (BL60584); *Dlg-GFP* (VDRC318133); *UAS-Pon-GFP* (Lu et al, 1999); *GFP::mud(62E1), GFP::mud(65B2)* gift from Yohanns Bellaïche (Ségalen et al, 2010); *UAS-cd8-mRFP* (BL27398); *neur-H2B-RFP* (Gomes et al, 2009); *UAS-pins-RNAi* (BL53968); *neurG4, UAS-RFP, jupiter::GFP* (recombined by Federica Mangione from BL6825); *meru^1^* (Banerjee et al, 2017); *CycA-GFP*, gift from Michel Gho and Agnes Audibert (Darnat et al, 2022); *UAS-nls-GFP* (BL4716).

## Genotypes

Figure 1B–C' *w; UAS-mK2-meru/+; neurG4/+* (D) *w; Ecad-GFP/ UAS-mK2-meru; neurG4/+* (E–E') *w; UAS-mK2-meru/+; neurG4, UAS-Pon-GFP/+* (F–F") *w; UAS-mK2-meru(WT)/+; GFP::mud(62E1), GFP::mud(65B2), neurG4/+* Fig. 3A *w; UAS-cd8-mRFP/+ GFP::mud(62E1), GFP::mud(65B2), neurG4/+* (B) *w; UAS-cd8-mRFP/+ GFP::mud(62E1), GFP::mud(65B2), neurG4, meru^1^/meru^1^* (D) *w;; neur-H2B-RFP, jupiter::GFP* (E) *w;; neur-H2B-RFP, jupiter::GFP, meru^1^/meru^1^* Fig. 4A *w; UAS-mK2-meru/+; CycA::GFP, neruG4/+* Fig. 5F *w; UAS-mK2-meru^RxL^/+; CycA::GFP, neurG4, meru^1^/meru^1^* Fig. 6B,B' *w; UAS-mK2-meru^WT^/+; GFP::mud(62E1), GFP::mud(65B2), neurG4, meru^1^/meru^1^* (C–C') *w; UAS-mK2-meru^RxL^/+; GFP::mud(62E1), GFP::mud(65B2), neurG4, meru^1^/meru^1^* (E) *w; UAS-mK2-meru^WT^/+; neurG4, UAS-RFP, jup::GFP, meru^1^/meru^1^* (F) *w; UAS-mK2-meru^RxL^/+; neurG4, UAS-RFP, jup::GFP meru^1^/meru^1^* Fig. 8A,A' *w; UAS-mK2-meru^WT^/ +; GFP::mud(62E1), GFP::mud(65B2), neurG4, meru^1^/meru^1^* (B–B') *w; UAS-mK2-meru^5S/T-A^/ +; GFP::mud(62E1), GFP::mud(65B2), neurG4, meru^1^/meru^1^* (D) *w; UAS-mK2-meru^WT^/+; neurG4, UAS-RFP, jup::GFP, meru^1^/meru^1^* (E) *w; UAS-mK2-meru^5S/T-A^/+; neurG4, UAS-RFP, jup::GFP meru^1^/meru^1^* Fig. EV1A,A' *w; Ecad-GFP/UAS-mK2-meru; neurG4/+* (B–B') *w; UAS-mK2-meru(WT)/ +; GFP::mud(62E1), GFP::mud(65B2), neurG4/+* (D–D') *w; UAS-mK2-meru/+; neurG4/+* Fig. EV3A,A' *w; UAS-nls-GFP/UAS-mK2-meru^RxL^; neurG4/+* (B–B') *w; Ecad-GFP/UAS-mK2-meru^RxL^; neurG4/+* (C–C') *w; UAS-mK2-meru^RxL^/+; neurG4, Dlg-GFP/+* (D–D') *w; UAS-mK2-meru^WT^/+; CycA::GFP, neurG4, meru^1^/meru^1^* Fig. EV5A,A',E,E',H,H' *w; UAS-mK2-meru^WT^/+; GFP::mud(62E1), GFP::mud(65B2), neurG4, meru^1^/meru^1^* (B–B', F–F', I–I') *w; UAS-mK2-meru^5S/T-A^/ +; GFP::mud(62E1), GFP::mud(65B2), neurG4, meru^1^/meru^1^* Fig. EV6B *w; UAS-mK2-Pins^WT^/+; neurG4/+* (C) *w; UAS-mK2-Pins^RxL^/+; neurG4/+* Movie EV1 *w;; neur-H2B-RFP, jupiter::GFP* Movie EV2 *w;; neur-H2B-RFP, jupiter::GFP, meru^1^/meru^1^*.

## Confocal live-imaging

*Drosophila* pupae were attached to a Superfrost microscope slide (Thermo Scientific) using double-sided tape so that the A–P axis was positioned along the width of the slide. Once adhered, the pupal case from the head to the start of the abdomen was peeled off. Two stacks of four 22 × 22 mm No. 1.5 Cover Glass (VWR) were glued together and

placed on either side of the pupae. A lightly oiled $24 \times 50$ mm No. 1.5 Cover Glass (VWR) was placed atop the two cover slip stacks so that the oil side rested gently on the exposed notum. The sample was then inverted to image on an inverted Nikon Spinning Disk (×60 or ×100 oil objectives) confocal microscope.

## Adult notum and wing imaging

Female adults, raised at 25 °C, were collected based on genotype and were stored at −20 °C or in 70% EtOH at 4 °C for notum and wing imaging, respectively. Adult nota were mounted in 0.8% low melting agarose and imaged on an MZ16 stereomicroscope (Leica) with a QImaging MicroPublisher 6 Colour Camera. Adult wings were mounted onto Superfrost microscope slides (Thermo Scientific) in Euparal covered in $22 \times 22$ mm No. 1.5 Cover Glass (VWR) for imaging. Wings were imaged with an Axio Scan.Z1 (Zeiss) slide scanner with a ×2.5 and ×10 objective.

## Colocalisation analysis, puncta quantification, cortical Mud/Meru/CycA levels and spindle orientation in time-lapse movies

All analyses were carried out using ImageJ. The degree of correlation between Meru and Mud/CycA localisation at the posterior cortex was measured by drawing a 2 μm thick line over the Meru cortical crescent. The line was applied to the single brightest slice (0.6 μm thick) in the stack used in the corresponding figure to the Meru and Mud/CycA channels, and the normalised grey value was plotted versus the distance along the line (in μm). The correlation between the Meru and Mud/CycA grey value levels was calculated by the Pearson's correlation coefficient (r) in which 1 represents perfect correlation, −1 represents anti-correlation and 0 depicts no correlation. Cortical levels of CycA/Meru were collected by the same means and measured by plotting mean intensity across cells. The number of Mud puncta was quantified by the observation of clustering and manually counting puncta at the cortex.

Levels of cortical Mud were quantified by measuring the sum of the integrated density of a cortex-specific ROI corrected for background fluorescence and ROI area. A mask of the membrane marker (CD8-mRFP or mK2-Meru) was used to outline the cell membrane and placed over the GFP channel to isolate the signal to the dividing cell. As the mitotic spindle in SOPs has a z-tilt (David et al, 2005), the anterior and posterior crescents were analysed separately. The three brightest slices (0.6 μM z-slices) for GFP::Mud signal were max-projected and isolated by drawing around the ROI, ensuring that the signal coming from the centrosome was excluded. The cytoplasmic signal (background fluorescence) was measured using a $10 \times 10$ pixel square for each cortical max projection. The sum of the anterior and posterior cortical Mud signals, respectively, were corrected for background fluorescence and area of the ROI by the following formula:

$$Corrected\ cortical\ value = Cortical\ integrated\ density\ sum \\ -(Cytosolic\ mean \times ROI\ area)$$

To measure spindle angles, movies obtained by confocal microscopy were imported into the Imaris Microscopy Image Analysis Software (Oxford Instruments) to quantify the x- and z axis of every centrosome pair in dividing cells of interest. All measurements were made by manually positioning a point at the

center of either centrosome, creating a line through the spindle. The x-, y- and z-coordinates of each point were used to quantify the angle of division relative to a reference position. The first measurement pair was used to define the dorsal midline, identified by visual cues such as anisotropic cell shape, followed by measurement pairs at all dividing cells of interest. The spindle angles in x axis (A–P axis) were calculated as degrees away from the midline. The reference slope (the midline angle) was calculated by the following equation:

$$ref_m = \left| \frac{\Delta Position\ Y}{\Delta Position\ X} \right|$$

To quantify the x axis angle of each centrosome pair, the slope was found with the same equation above (called $centrosome_m$) and calculated as follows:

$$\theta_{A-P} = \tan^{-1} \frac{|ref_m - centrosome_m|}{1 + (ref_m \times centrosome_m)}$$

The z-axis angle (A–B axis) was calculated as degrees away from the plane. The reference plane angle was set to 0. and the absolute z-axis angle of each centrosome pair was calculated with the following equation:

$$\theta_{A-P} = \left| \frac{\Delta Position\ Z}{\sqrt{\Delta Position\ X^2 + \Delta Position\ Y^2 + \Delta Position\ Z^2}} \right|$$

## Statistical analysis

Mann–Whitney $U$ tests were performed to test if mean values of cortical Mud/CycA levels were significantly different. The Kolmogorov–Smirnov (K-S) test was used to test if measurements from SOP spindle alignment sample distributions were significantly different. To test the colocalisation of two fluorescently tagged proteins in a time-lapse movie, the Pearson's correlation test was performed. No blinding was done.

## *Drosophila* cell culture and expression constructs

*Drosophila* S2 cells were transfected with Effectene transfection reagent (Qiagen) and grown in *Drosophila* Schneider's medium (Life Technologies) containing 10% FBS (Sigma), 50 μg/ml penicillin and 50 μg/ml streptomycin. Expression plasmids were generated using Gateway® technology (Life Technologies). Open reading frames (ORF) were PCR-amplified from existing plasmids or cDNA (DGRC, https://dgrc.bio.indiana.edu/vectors/Catalog) and cloned into entry vectors (pENTR™/D-TOPO). Vectors from the *Drosophila* Gateway Vector Collection (https://carnegiescience.edu/bse/drosophila-gateway-vector-collection) were used as Destination vectors. Meru, Mud and Baz were tagged N-terminally and Dsh, Cdk1 and all the Cyclins were tagged C-terminally. Point mutations were generated using the Quikchange Multi Site-Directed Mutagenesis Kit (Agilent). All expression plasmids were sequence-verified.

## dsRNA production and treatment

dsRNAs were synthesised using the Megascript T7 kit (Thermo Fisher Scientific). DNA templates for dsRNA synthesis were

PCR-amplified from genomic DNA using primers that contained the 5' T7 RNA polymerase-binding site sequence. dsRNAi primers were designed using the DKFZ RNAi design tool (https://e-rnai.dkfz.de/signaling/e-rnai3/). RNAi experiments were carried out in six-well plates using S2 cells. In total, $1$–$2 \times 10^6$ cells were plated per well and cells left to settle. After 3 h, the medium was removed and replaced with 1 mL of serum-free Schneider's medium containing dsRNAs (20 µg). Cells were soaked for 30 min, and then 2 mL of full Schneider's medium was added. Transfections were carried out 1 h after dsRNA treatment.

### Immunoprecipitation and immunoblot analysis

For immunoprecipitation assays, cells were lysed in lysis buffer (150 mM NaCl, 50 mM HEPES pH 7.5, 0.5% (v/v) Triton X-100) supplemented with phosphatase inhibitor cocktails 2 and 3 (Sigma) and protease inhibitor cocktail (Roche). Cell extracts were cleared at 13,200 rpm for 20 min at 4 °C. GFP and Myc-tagged proteins were purified using agarose beads (Chromotek) according to the manufacturer's protocol. For MS analysis, proteins were eluted using 5% SDS 40 mM in Hepes pH 7.0. Detection of purified proteins and associated complexes was performed by immunoblot analysis using chemiluminescence (Pierce). Western Blots were probed with mouse anti-FLAG (M2, Sigma), rabbit anti-FLAG (Sigma), mouse anti-Myc (9E10, Santa Cruz Biotechnology), rabbit anti-HA (C29F4, NEB), and rat anti-GFP (Chromotek).

### Mass spectrometry analysis

Following immunoprecipitation, samples were digested using the S-Trap micro protocol (ProtiFi). Eluted peptides were resuspended in 0.1% trifluoroacetic acid (TFA) and analysed on an Orbitrap Eclipse (Thermo Fisher Scientific) coupled to an UltiMate3000 LC system. In brief, 2 mg of dried peptides were injected onto a trap column (Acclaim PepMap 100; Thermo Fisher Scientific), and peptides were resolved using an Easy-Spray 50 cm column (PepMap RSLC C18) with the following gradient: 0 min—2% solvent B (75% acetonitrile in 0.1% formic acid), 120 min—35% solvent B, 137 min—95% solvent B, 143 min—95% solvent B, 148 min—2% solvent B). The Orbitrap Eclipse was operated in data-dependent acquisition mode using a fixed cycle time of 3 s. The Orbitrap resolution was set to 120,000, the scan range to 300–1500 $m/z$, standard AGC and auto-injection were enabled, and the fragmentation method was set to HCD at 30. MS2 scans were performed in the Orbitrap with a resolution of 30,000, standard AGC and scan parameters and dynamic injection times. Raw data from duplicate samples was searched in MaxQuant with phosphorylation (STY) and oxidation set as variable modifications, while carbamidomethylation was set as a fixed modification. Label-free quantification (LFQ) intensities corresponding to modified peptides of interest were plotted on the graph.

## Data availability

Source data (imaging and biochemistry) are associated with the manuscript. The Mass Spectrometry proteomics data have been deposited to the ProteomeXchange Consortium via the PRIDE (Perez-Riverol et al, 2025) partner repository (https://www.ebi.ac.uk/pride/) with the dataset identifier PXD060813.

The source data of this paper are collected in the following database record: biostudies:S-SCDT-10_1038-S44318-025-00420-5.

## Peer review information

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

## Acknowledgements

The authors thank Y Bellaïche, M Gho, F Schweisguth, F Mangione and the Bloomington *Drosophila* Stock Center for fly stocks and Y Bellaïche for plasmids. The authors are grateful to M Renshaw and members of the Crick Advanced Light Microscopy Facility for advice on confocal imaging and image analysis; J Kurth, S Murray, H Shaw, S Ruiz-Herrera from the Crick Fly Facility for embryo injections and fly stock maintenance; and Tapon lab members for advice. We thank A Moraiti, F Mangione, N Goehring and A Audibert for critical reading of the manuscript. This work was supported by the Francis Crick Institute, which receives its core funding from Cancer Research UK (CC2138, CC1063), the UK Medical Research Council (CC2138, CC1063) and the Wellcome Trust (CC2138, CC1063), and a Wellcome Trust Investigator award (107885/Z/15/Z) to NT. For the purpose of Open Access, the authors have applied a CC BY public copyright licence to any Author Accepted Manuscript version arising from this submission.

## Author contributions

**Melissa M McLellan**: Conceptualisation; Investigation; Writing—original draft; Writing—review and editing. **Birgit L Aerne**: Conceptualisation; Supervision; Investigation; Writing—original draft; Writing—review and editing. **Jennifer J Banerjee Dhoul**: Conceptualisation; Investigation; Writing—review and editing. **Maxine V Holder**: Investigation; Writing—review and editing. **Tania Auchynnikava**: Investigation; Writing—review and editing. **Nicolas Tapon**: Conceptualisation; Supervision; Funding acquisition; Writing—original draft; Writing—review and editing.

Source data underlying figure panels in this paper may have individual authorship assigned. Where available, figure panel/source data authorship is listed in the following database record: biostudies:S-SCDT-10_1038-S44318-025-00420-5.

## Funding

## Disclosure and competing interests statement

The authors declare no competing interests.

# Expanded View Figures

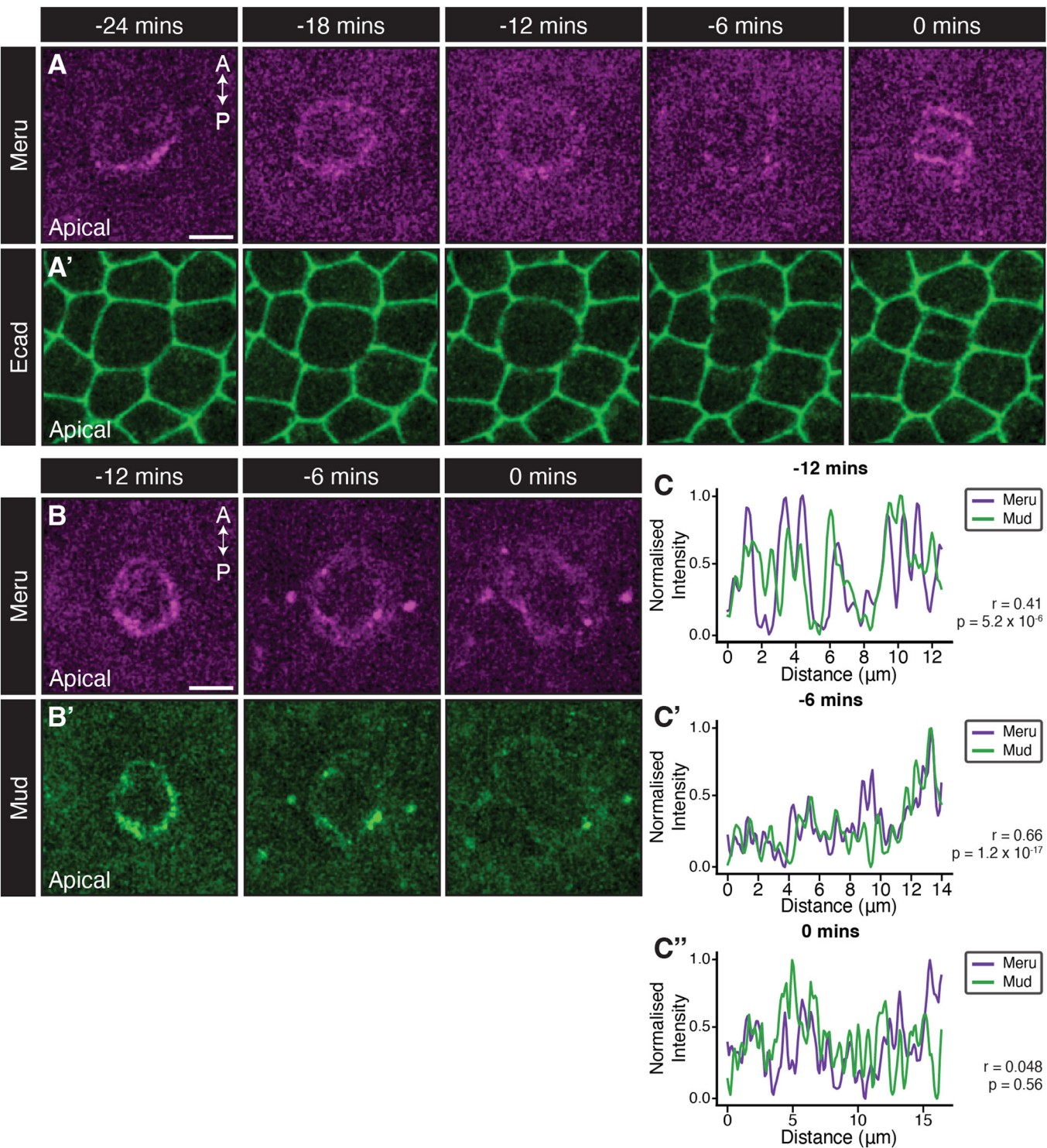

**Figure EV1. Posterior–apical localisation of Meru occurs prior to SOP mitosis and persists throughout division.**

(A–B') Maximum intensity projections of apical-most three sections of pupal nota live-imaged on a confocal microscope at 16 h APF, 0 min marking the first frame indicating cytokinesis, of *neurG4 > UAS-mK2-meru* (magenta; **A**, **B**, see composite in Fig. 1D), co-expressed with Ecad-GFP and Mud-GFP (green; **A'**, **B'**, respectively). (**C–C''**) Single brightest slice in (**B**, **B'**) plotted as line graphs by measuring the grey value across the posterior domain as defined by Meru localisation normalised to highest value in each channel. High Pearson's coefficient (*r*) indicates positive correlation. *P* values calculated using a two-tailed test. Scale bars = 5 µm. Source data are available online for this figure.

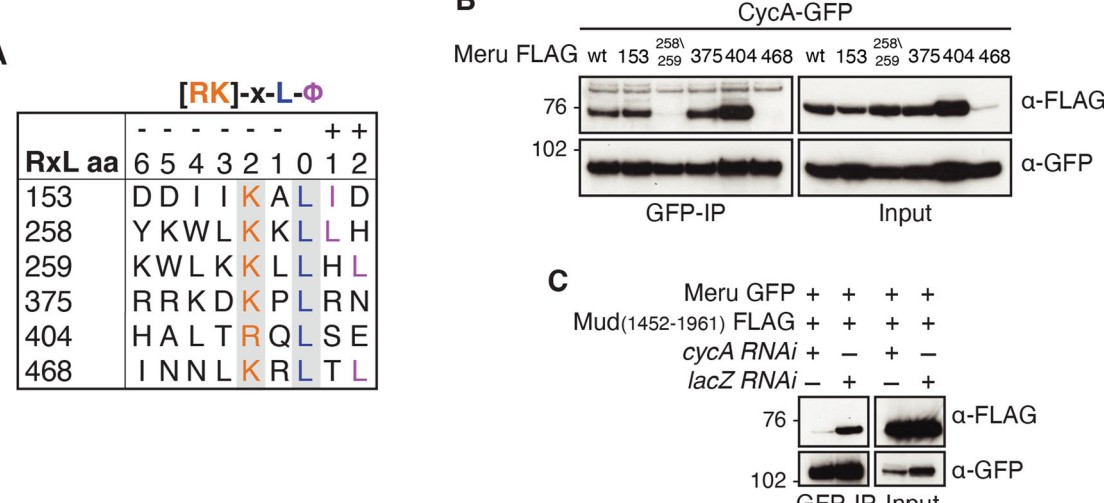

**Figure EV2. The RxL motifs of Meru.**

(A) The amino acid (aa) positions of the six RxL motifs identified in the Meru sequence are indicated on the left. On the right is the sequence alignment of all RxL motifs. Orange indicates the R/K and the L is labelled in blue at position 0. The hydrophobic aa is marked in purple and the grey boxes indicate the two aa that were mutated to Alanine in RxL mutants. (B, C) Western blots of co-IP experiment using cell lysates from transfected S2 cells, immunoprecipitated and probed using the indicated antibodies. (B) CycA immunoprecipitates with each Meru RxL motif mutant, except when Meru 258/9 is mutated to Alanines. (C) CycA depletion by RNAi reduced Meru/Mud association. Source data are available online for this figure.

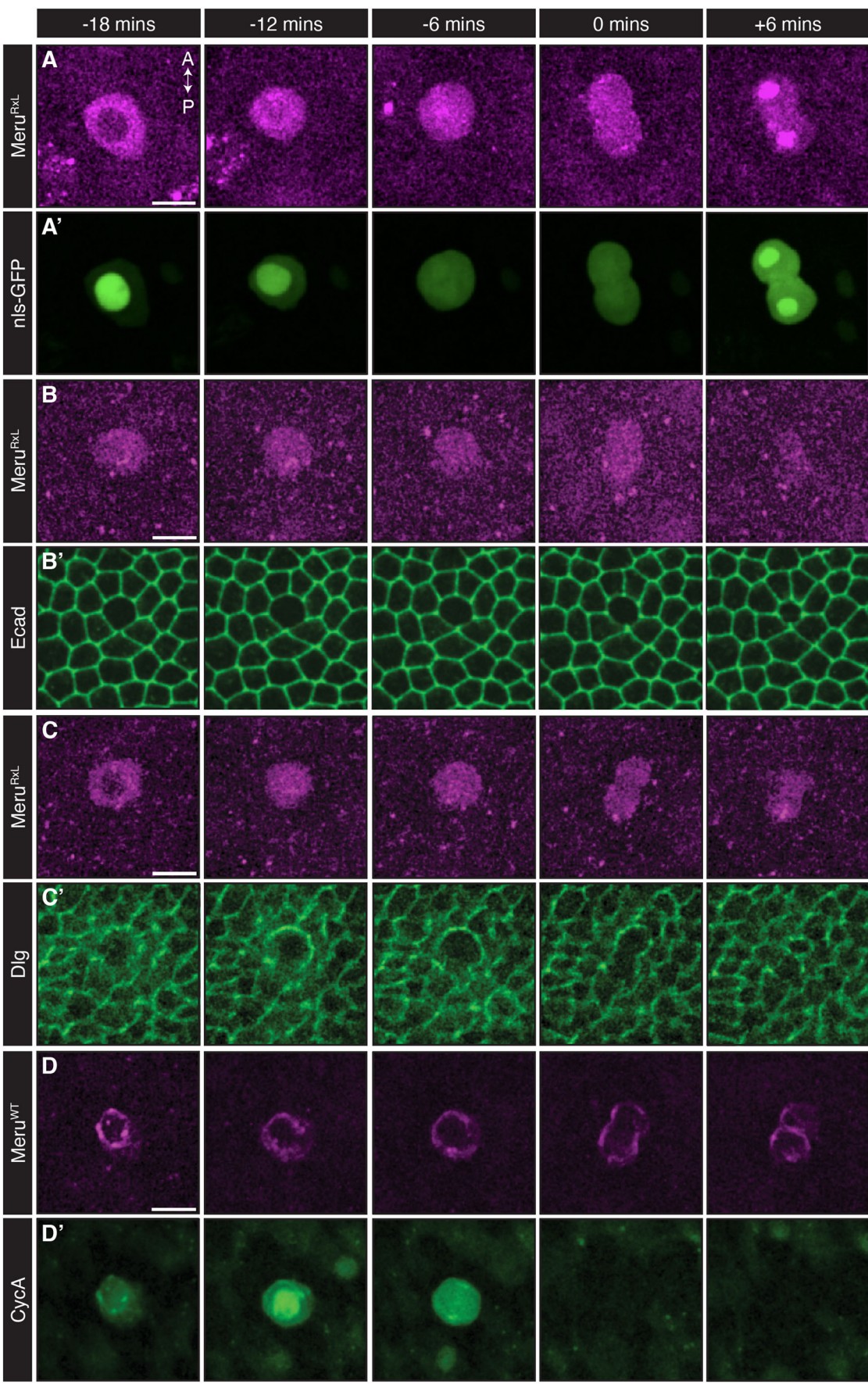

◀ **Figure EV3. 258/9 RxL motif mutant Meru no longer binds CycA and is mislocalised in vivo.**

(**A–D**) Maximum intensity projections of pupal nota live-imaging at 16 h APF, 0 min marking the first frame indicating cytokinesis, of *neurG4 > UAS-mK2-meru*^RxL^ and *neurG4 > UAS-mK2-meru*^WT^ (magenta; **A–C** and **D**, respectively), co-expressed with nuclear nls-GFP, apical Ecad-GFP, basolateral Dlg-GFP and CycA-GFP (green; panels **A′**, **B′**, **C′** and **D′**, respectively). Scale bars = 10 μm. Source data are available online for this figure.

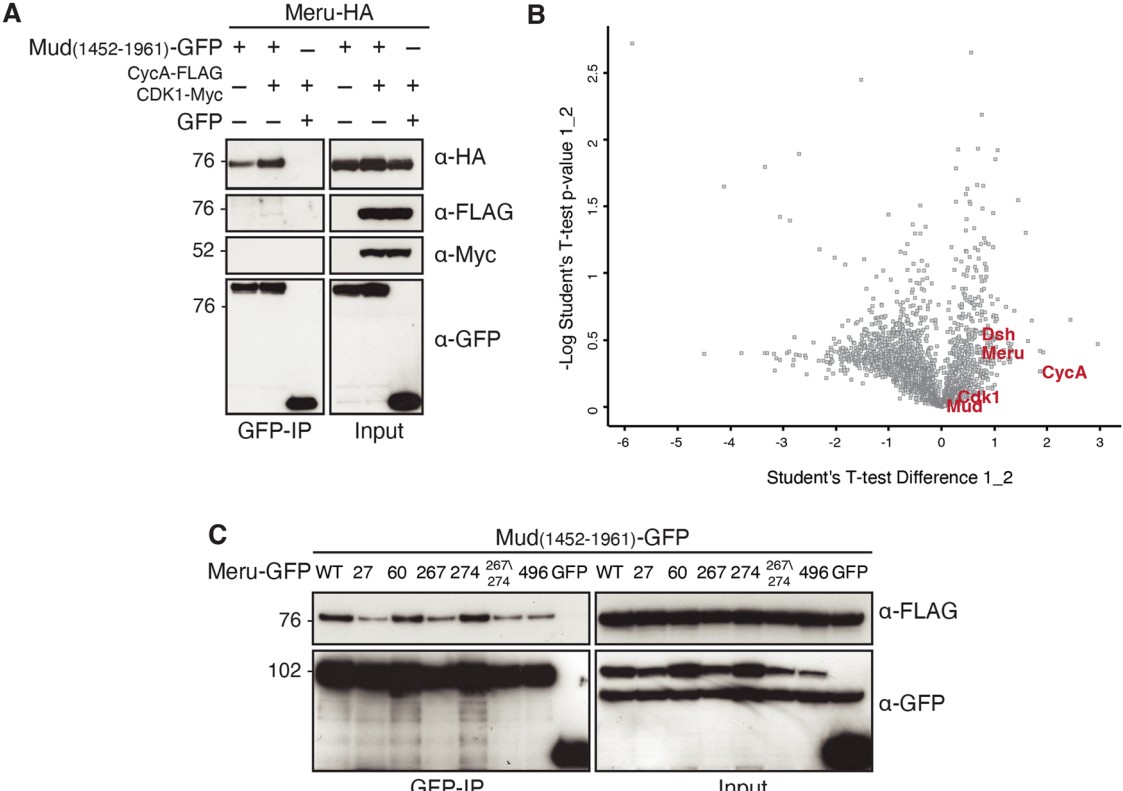

**Figure EV4. Overexpression of Cdk1 and CycA promotes the Meru/Mud interaction.**

(A, C) Western blots of co-IP experiment using cell lysates from transfected S2 cells, immunoprecipitated and probed using the indicated antibodies. (A) Overexpression of Cdk1/CycA increases the amount of Meru binding to Mud. (B) MeruRxL has decreased interaction with CycA, Cdk1, Mud and Dsh relative to MeruWT in a peptide identification analysis by MS (N = 2). (C) No single Meru S/T-P phosphosite is essential to co-IP with C-terminal Mud. Source data are available online for this figure.

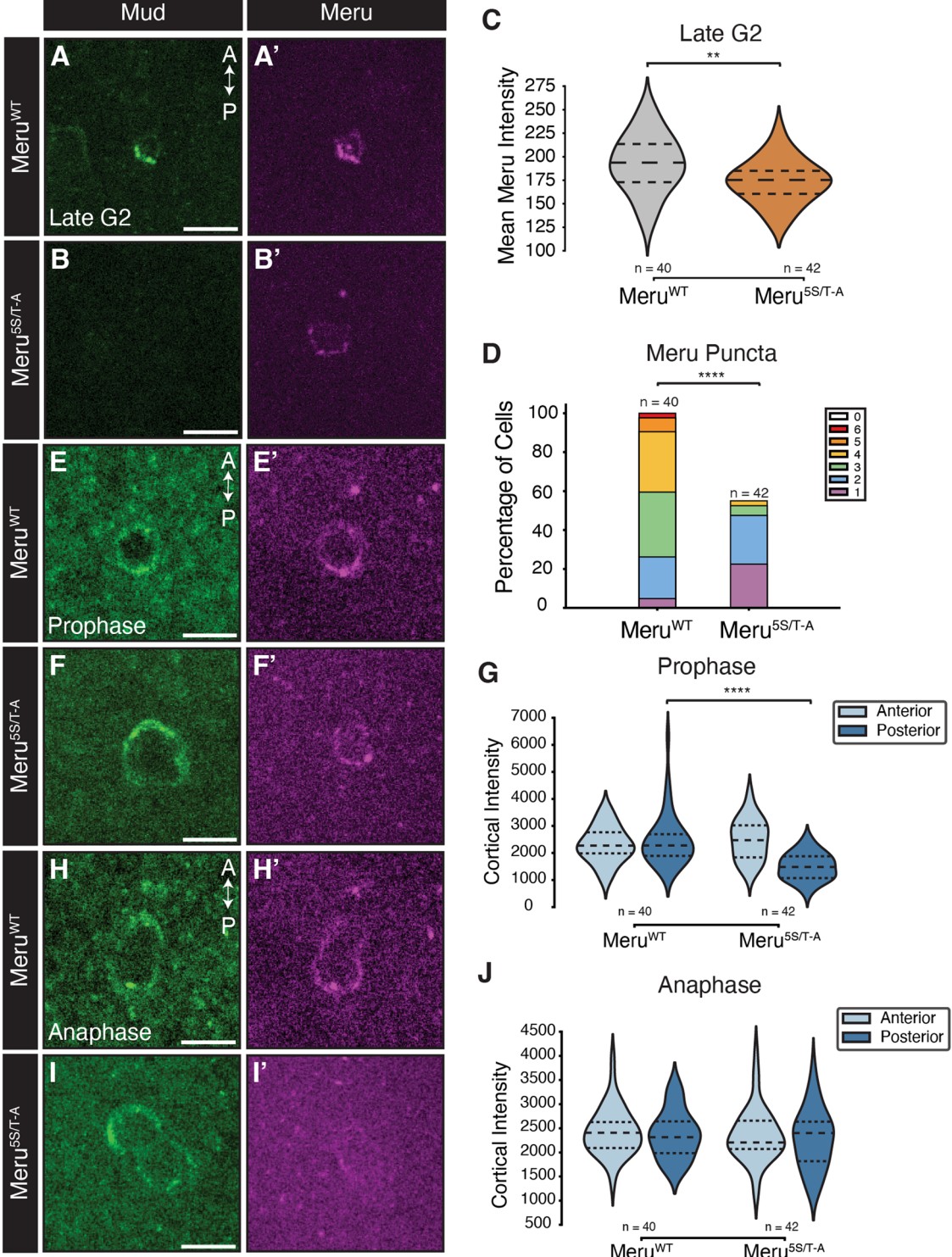

**Figure EV5. Posterior cortical Mud levels return by anaphase in Meru$^{SS/T-A}$ mutants.**

(**A–B'**) Single slice from confocal live-imaging of pupal nota at 16 h APF expressing *Mud-GFP* (green; **A**, **B**) and *neurG4 > UAS-mK2-meru$^{WT}$* (magenta; **A'**) or *neurG4 > UAS-mK2-meru$^{SS/T-A}$* (magenta; **B'**) at late G2. (**C**) Graph showing the mean intensity of posterior cortical Meru during late G2 in Meru$^{WT}$ and Meru$^{SS/T-A}$ ($P = 2.7 \times 10^{-3}$). (**D**) Graph displaying the average number of Meru puncta at the posterior cortex in late G2. Puncta of the Meru$^{SS/T-A}$ mutant are significantly decreased relative to Meru$^{WT}$ ($P = 9.10 \times 10^{-15}$). (**E–F'**) Maximum intensity projections (6 slices; **E**, **F**) from confocal live-imaging of pupal nota at 16 h APF expressing *Mud-GFP* (green; **E**, **F**) and *neurG4 > UAS-mK2-meru$^{WT}$* (magenta; **E'**) or *neurG4 > UAS-mK2-meru$^{SS/T-A}$* (magenta; **F'**) in a *meru$^1$* background at prophase. (**G**) Graph showing the intensity of cortical Mud at the anterior and posterior crescent of each genotype at prophase. Posterior Meru$^{SS/T-A}$ is significantly lower than Meru$^{WT}$ ($P = 1.61 \times 10^{-7}$). (**H–I'**) Maximum intensity projections (6 slices; **H**, **I**) from confocal live-imaging of pupal nota at 16 h APF expressing *Mud-GFP* (green; **H**, **I**) and *neurG4 > UAS-mK2-meru$^{WT}$* (magenta; **H'**) or *neurG4 > UAS-mK2-meru$^{SS/T-A}$* (magenta; **I'**) in a *meru$^1$* background at anaphase. (**J**) Graphs showing the intensity of cortical Mud at the anterior and posterior crescent of each genotype at anaphase, where there is no measurable difference at anaphase ($P = 0.47$). **$P < 0.01$ and ****$P < 0.0001$ using a Mann–Whitney *U* test. Large and small dashed lines in violin plots represent the median and Q1/Q3, respectively. Scale bars = 10 μm. (**C, D, G, J**) Number of SOPs imaged indicated on the panels from 3 nota. Source data are available online for this figure.

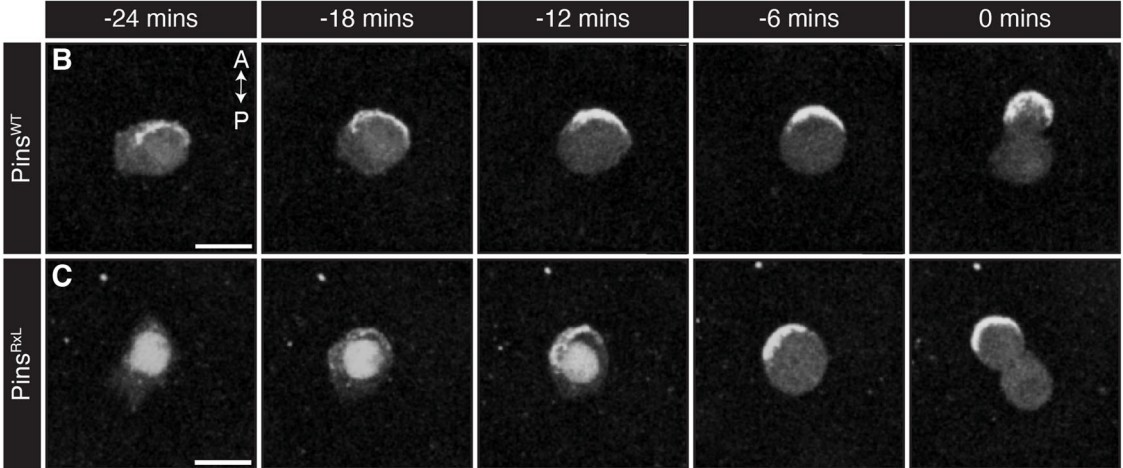

**Figure EV6. Loss of the Pins RxL motif results in delayed cortical localisation.**

(**A**) BLAST search results for novel [R/K]-[R/K]-L-L-x[7]-[S/T]-P (RRLL) motifs resulted in nine proteins of interest, including Meru, Pins, Gukh and AurA, all of which are involved in spindle orientation. (**B**, **C**) Maximum intensity projections from confocal live-imaging of pupal nota at 16 h APF, 0 min marking first frame in cytokinesis, shows expression of *UAS-mK2-Pins*$^{WT}$ (**B**) and *UAS-mK2-Pins*$^{RxL}$ (**C**) driven by *neurG4* during SOP mitosis. Scale bars = 10 μm. Source data are available online for this figure.

