## [Peer Review File · The EMBO Journal]

Meru co-ordinates spindle orientation with cell polarity and cell cycle progression

Melissa McLellan, Birgit Aerne, Jennifer Banerjee Dhoul, Maxine Holder, Tatsiana Auchynnikava, and Nicolas Tapon

Corresponding author(s): Nicolas Tapon (Nic.Tapon@crick.ac.uk)

Review Timeline:

Submission Date:	26th Nov 23
Editorial Decision:	24th Feb 24
Revision Received:	14th Feb 25
Editorial Decision:	3rd Mar 25
Revision Received:	13th Mar 25
Accepted:	17th Mar 25

Editor: Hartmut Vodermaier

Transaction Report:

Dear Nic,

Thank you again for submitting your manuscript on Meru and spindle orientation to The EMBO Journal. I apologize for the delay in getting back to you with a decision, but all suitable reviewers had been very busy at the end of the year and those available needed additional time for their reports. We have now finally received a full set of comments from three expert, which I am copying below for your attention. As you will see, all referees appreciate the subject of the study and your approach for tackling it, as well as the overall presentation. However, while referees 1 and 2 are generally supportive of publication, referee 3 remains unconvinced that the main conclusions of the study are at this stage sufficiently supported by the presented biochemical and genetic data. In this situation, I would be interested in hearing your thoughts on these criticisms, and how you might be able to address/clarify them if given an opportunity to revise the paper for The EMBO Journal. Therefore, please carefully consider the reports together with your coworkers, and get back to me with a tentative point-by-point response, answering in particular to the main issues raised by referee 3, as well as the key points of the other referees. Based on this proposal (and possible follow-up discussion), I could then determine whether it would be justified (and promising) to invite a major revision for The EMBO Journal. It would be great if you could get back to me with such a revision plan over the course of next week.

Looking forward to hearing from you,

Best regards,

Hartmut

Referee #1 (Report for Author)

Mechanisms that orient the mitotic spindle is important in epithelial morphogenesis and for cell fate decisions in the developing nervous system. A key model system in this regard is the fruit fly *Drosophila*. In the present study, the spindle orientation mechanism of

sensory organ precursors is investigated. In a particular, the study aims at understanding how the first division of the sensory organs is occurring in the correct orientation of the epithelium they reside in. This is developmentally important to orient the small hairs on the back of the fly. Traditionally this is orchestrated by the planar cell polarity machinery that has been intensively studied. What is much less clear is how this machinery once polarised actually grabs the mitotic spindle at the correct time. The lab previously identified the protein Meru to couple planar cell polarity and apico basal polarity. In this follow up study the lab provides more mechanistic insight. In fact they propose now that Meru is a bridging factor that recruits Mud to the apical posterior domain where Dishevelled is localised. This is mediated by Meru's ability to bind Cyclin A enabling the formation of a ternary complex between Meru, Dish and Mud to position the spindle at the correct time. This in turn orients the P1 division such that it is aligned with the dorsal midline in the notum but mostly does not affect the characteristic z-tilt in apico basal direction.

This was revealed using solid and cleanly presented data involving genetics in combination with live imaging (overexpression of Meru and variants in which the most likely Cyclin A docking site sequences had been mutated (MeruRxL)) as well as biochemical approaches to study protein complex formation (S2 cell assay, overexpression of various relevant combinations of proteins followed by IP and western blotting to monitor protein-protein interaction and complex formation).

This is a neat study providing sound mechanistic insights. Cell-type specific mechanism of Mud regulation and function exist and this study provides a further example which is of interest. I suggest minor revision before publication.

Comments

Meru and Mud localisation look convincing. Regarding Mud and spindle orientation in SOPs:

Fig 1 A suggest Mud to be basal at the anterior and apical at the posterior. Yet in Fig 3A and 7B Mud seems rather planar polarised at anaphase. The amounts of the relevant proteins localised looks very little and the asymmetries there but not black and white in this system. Why is Mud not orienting the spindle at the anterior? Does this mean that Mud is only competent to grab the spindle when complexed with Dsh and Meru, as it is at the apical posterior?

(minor: signal that was quantified in Fig EV1 B' 0min looks different to that of Fig 3A and 7B, is this normal variability? There is something wrong in Fig EV1 annotation I think)

In Fig 2B Mud (1825-2457) is not at all detected in the input but becomes just detectable in the IP yet it pulls down almost similar amounts of Meru-FLAG as Mud (1452-1961) which is expressed significantly more. This made me wonder whether it is purely the PBD that mediates affinity or something else in the Cter behind the PBD of Mud that helps bind Meru? Also the proteins produced in insect cells are likely to be hyperphosphorylated, does this have any implication for the interpretation of the results?

CylinA localisation is lost in anaphase, yet Mud and Meru stay positioned, so something needs to propagate that positional information in mitosis to prevent spindles from rotating which is likely to be the phosphorylation of Meru by CDK1. The data for Meru- CyclinA interaction and function is strong, the part invoking CDK1 itself less so. I agree it is highly likely that CDK1 activity is involved, yet there is no direct experiment to show that CDK1 is the kinase, or that Meru is even phosphorylated on the site (the point mutations may change structure and thereby affect the function). It may be interesting and not too time consuming to IP Meru-GFP from S2 cells and perform phosphoproteomics to check if the suspected sites are phosphorylated (co-express CycA/CDK1 if needed) and whether that is lost in the RxL mutants and perhaps use this assay in some other form to probe that CDK1 can directly phosphorylate Meru on the suspected sites? It will be very difficult to demonstrate that in the tissue.

Checking what happens to Cyclin A in meru1 mutants alone and those expressing wt Meru and MeruRxL, if Cyclin A localisation lost could further strengthen the model.

The paragraph line 418 onwards should be in the results as it introduced hitherto not presented data and could go with figure 8 detailing the model and hinting at broader implications of aspects of the findings.

Referee #2 (Report for Author)

McLelland et al. demonstrated that the N-terminal RASSF protein, Meru, bridges the Frizzled/Dishevelled planner cell-polarity complex with Mud spindle-tethering machinery using *Drosophila* SOPs and S2 cells. In addition, they showed that the RxL motif on Meru is required for its interaction with Cyclin A, and that mutations of putative CDK phosphorylation sites on Meru and Dsh disrupt the ternary complex formation of Mud/Meru/Dsh. Overall, the data are of high quality and the results are well documented.

This study reveals a missing link that couples cell polarity with spindle orientation in a cell-cycle-dependent manner. This study will contribute to the asymmetric-cell-division field; thus, it warrants publication in EMBO, but the following matters need to be addressed.

Major comments

1. The authors propose that cyclin A/Cdk1 promotes formation of the Mud/Meru/Dsh complex by phosphorylation of Meru and Dsh. Although the authors performed IP experiments nicely using Meru5S/T-A mutants (Figure 6 and EV4), there is no direct evidence of Meru phosphorylation at these Cdk consensus sites by cyclin A/Cdk1. It is unclear whether Meru-HA shows band shifts in a Cyclin A/Cdk1-dependent manner (Figure EV4-A). Using mass spectrometry or phos-tag gels in combination with a Meru5S/T-A mutant would be useful to demonstrate phosphorylation on Meru at these sites. If direct evidence is not provided, the authors should reword lines 35, 392, 956 and others accordingly to avoid overstatements.
2. Localization and functions of the Meru5S/T-A mutant have not been analyzed. Although Meru5S/T-A-GFP still interacts with Dsh in S2 cells (Figure 6), the Meru5S/T-A mutant may cause mis-localization and functional abnormalities in SOP cells. As shown for the MeruRXL mutant (Figure 7), it would be worth analyzing localization and functions of the Meru5S/T-A mutant in SOP cells in order to support the model shown in Figure 8.

Minor comments

3. In the legend to Figure 2A, the authors need to explain what RA denotes. Is this the conserved Ras-association domain?
4. In the diagram of Figure 5A, it would be helpful if the authors indicate where the RA and coiled-coil are located. Are the important 258/259 RXL motifs located in the RA domain?
5. On page 8 line 157, the authors mentioned that GFP-Meru localizes to the posterior cortex at interphase, whereas they mentioned that both meru and cyclin A co-localize in late G2 (page 11, lines 243-244). Does Meru localize to the posterior cortex only in late G2? If Meru localizes in G1 or S, how does it localize?
6. Fig EV5A-C are considered in the Discussion, but they are not mentioned in the Results. The authors need to confirm whether it is permissible in EMBO to discuss figures without first describing them in the Results.
7. On page 17, line 360 and others, the authors use the meru1 mutant. It would be helpful to briefly describe characteristics of this mutant for the benefit of those who do not work in this field.
8. On page 31, line 663, the journal name of Darnat et al., is missing.

Referee #3 (Report for Author)

An evolutionarily-conserved biological machine comprised of Mud (NuMA/LIN-5) and the microtubule motor dynein localizes to the mitotic cell cortex, where it is thought to reel astral microtubules, and therefore spindles, into alignment. The question of how Mud achieves its specific cortical location has received a lot of attention and appears to have a number of different cell-specific answers. Spindle orientation is particularly important in asymmetrically-dividing cells like the *Drosophila* SOP. In this study the authors tackle a cool question raised by previous work. How is Mud recruited to the apical/posterior cortex in SOPs? Two models have been proposed; one relies on interaction between Mud and Dsh, whereas the other on Cyclin A. The authors undertake to settle the discrepancy and describe the scaffolding protein Meru as a molecular "bridge" that links Mud, Dsh, and CycA.

There's a lot to like about this paper. It's neatly written, thorough in its scholarship, and the combination of biochemistry, genetics, and live imaging approaches is commendable. Unfortunately, I didn't find the story convincing. My impression is that this study is trying to shoe-horn some nice biochemical data into a model that the genetics don't support.

Major concerns:

1. I have a fundamental concern with the presentation of cortical Mud levels. It is clear from the methods that the authors put an impressive amount of thought into how this should be done, and I am therefore confident that the experimentalists were very careful. However, I very strongly disagree with a comparison of absolute pixel intensity across samples. Pixel intensity of confocal micrographs is impacted by some variables that are hard to control for, for example distance from the lens (even at the level of microns).

Related to this concern, I cannot find the number of nota that were imaged for each condition, only the number of pl cells.

The methods section described an "A-P ratio" and given the concern about variability in intensity between sample preparations this seems to be the most reliable way of presenting the data. If this was the case, the differences in Figure 3C would probably disappear. Comparison of the data in Figure 3C with 7D might also be facilitated (see below).

2. What kind of allele is meru[1]? (This needs to be described). Ideally more than one

allele/chromosome should be used to account for the possibility of more than one relevant mutation on the meru[1] chromosome. Is meru[1] the only way to disrupt Meru function genetically? Are there deletions available?

3. I had trouble with the interpretation of Figure 3. If meru[1] is a null, then Meru is clearly not required for the cortical localization of Mud (as claimed in the text). At best it contributes, but that interpretation is based on a comparison of pixel intensity across samples, which is problematic as described above. More importantly, the authors point out that loss of Meru function does not impact asymmetry. These data therefore do not agree with the authors' model, which is that posterior Meru links Dsh with Mud.

Related to this, the authors report that spindle orientation in meru mutants is less planar (more tilted) than the wt. That is the *opposite* phenotype to frizzled and dsh mutants, which are more planar than wt (Ségalan). The authors need to address this contradiction.

Another consideration is that the cortical localization of Mud being shown in Figure 3A is at anaphase. Is this the point being measured in C? This might impact interpretation, since Mud-dependent spindle orientation has already been achieved by anaphase.

4. "In the rescued conditions, both the anterior and posterior cortices had nearly identical levels of Mud when mK2-MeruWT was expressed (Figure 7B, B' and D). It is interesting to note that, in the absence of Meru overexpression, Mud is more enriched at the anterior than the posterior cortex (Fig 3C). The equal cortical distribution of Mud upon mK2-MeruWT overexpression is therefore consistent with Meru promoting Mud posterior recruitment."

One concern raised with this interpretation is that the authors do not present evidence that Meru[WT] is being overexpressed, only that it is being expressed. How strong is normal expression vs. the GAL4-mediated expression?

Figure 7D shows that expression of Meru[WT] evens out the Mud asymmetry observed in wild type SOPs. I'm confused by the interpretation of these data, which is that Meru[WT] promotes Mud posterior recruitment. I don't think there's enough information to make that claim. Firstly, Meru[WT] is observed at both the anterior and posterior. Secondly, the authors made a convincing case in Figure 1G that Meru and Mud colocalise at confocal resolution. This does not appear to be strictly true in the Meru[WT] condition. They overlap somewhat but intensities don't match. Thirdly, there is no evidence to show where Dsh is in this condition. Finally, we have been told previously (Figure 3C) to compare absolute pixel

intensity between conditions. If we do that here, comparing 7D to 3C yields the straightforward conclusion that Mera[WT] interferes with the anterior localization of Mud, which is reduced in comparison to wt cells, whereas posterior localization is unaffected.

Spindle angles in Figure 7 should be juxtaposed with those in Figure 3 and the appropriate statistical comparisons made. The authors state that Meru[WT] rescues A-P spindle orientation, which appears to be the case, but I can't tell whether Meru[RxL] is defective to the same extent as meru[1]. Most spindle angles in the Meru[RxL] condition are below 45 degrees, with the largest population below 15 degrees (as in the control). That looks quite a bit different from meru[1], arguing that Meru[RxL] rescues at least partially. Related to this, there is no A-B spindle orientation defect in the Meru[RxL] condition.

Minor comments:

Mud and Dsh interact in HEK293T cells. The authors propose that this is explained by the expression of Meru orthologs. It's a very compelling idea that really should be pushed further. Can it be tested? The authors could at least mention how similar these orthologs are to Meru.

It's not clear to me what the live images are. Z-stacks? The authors point out in the Methods that they measured posterior and anterior localizations separately because they are not at the same depth. A specific concern is that the images shown in, for example, Figure 7C, are not representative of the quantifications in 7D.

Additional suggestions/questions:

The cartoon in Figure 1A illustrates an approximately 45 degree spindle tilt in the A-B axis. The measured tilt is 6.8 degrees. I do not dispute a "characteristic z-tilt" (Line 231) in SOP cells, but 6.8 is close enough to zero that a reasonable explanation for that number in these measurements is simple noise (whether biological or technical) in the data. This is especially true given that all data points are positive. (You'd have to measure only zeros to get a zero average). Can the authors comment on this?

The model these authors present (summarized in Figure 8) is that Meru links Mud to Dsh. While I don't consider it a requirement, it would be nice to see evidence that Meru and Dsh colocalise at confocal resolution, as in Figure 1G.

I found the juxtaposed rose plots in figures 3 and 7 a little puzzling (at first) and ultimately

hard to interpret. I suggest that the authors could find a different way to plot the data (e.g. violin plots). This would also facilitate comparison between figures.

What is that big ball of Pins[RxL] in the middle of the cell!?

McLellan *et al.* EMBOJ-2023-116230**Response to reviewers and revision plan**

We thank the reviewer for their insightful comments. Our responses and revision plans are in blue.

Referee #1

Mechanisms that orient the mitotic spindle is important in epithelial morphogenesis and for cell fate decisions in the developing nervous system. A key model system in this regard is the fruit fly *Drosophila*. In the present study, the spindle orientation mechanism of sensory organ precursors is investigated. In a particular, the study aims at understanding how the first division of the sensory organs is occurring in the correct orientation of the epithelium they reside in. This is developmentally important to orient the small hairs on the back of the fly. Traditionally this is orchestrated by the planar cell polarity machinery that has been intensively studied. What is much less clear is how this machinery once polarised actually grabs the mitotic spindle at the correct time. The lab previously identified the protein Meru to couple planar cell polarity and apico basal polarity. In this follow up study the lab provides more mechanistic insight. In fact they propose now that Meru is a bridging factor that recruits Mud to the apical posterior domain where Dishevelled is localised. This is mediated by Meru's ability to bind Cyclin A enabling the formation of a ternary complex between Meru, Dish and Mud to position the spindle at the correct time. This in turn orients the P1 division such that it is aligned with the dorsal midline in the notum but mostly does not affect the characteristic z-tilt in apico basal direction. This was revealed using solid and cleanly presented data involving genetics in combination with live imaging (overexpression of Meru and variants in which the most likely Cyclin A docking site sequences had been mutated (MeruRxL)) as well as biochemical approaches to study protein complex formation (S2 cell assay, overexpression of various relevant combinations of proteins followed by IP and western blotting to monitor protein-protein interaction and complex formation.

This is a neat study providing sound mechanistic insights. Cell-type specific mechanism of Mud regulation and function exist and this study provides a further example which is of interest. I suggest minor revision before publication.

We thank the reviewer for his/her positive assessment of our manuscript.

Comments

Point 1

Meru and Mud localisation look convincing. Regarding Mud and spindle orientation in SOPs:

Fig 1 A suggest Mud to be basal at the anterior and apical at the posterior. Yet in Fig 3A and 7B Mud seems rather planar polarised at anaphase.

In the Figure 1A diagram, we illustrate the characteristic tilt of the spindle to make it obvious to the reader that Mud is planar polarised along the A-P axis, and also that its recruitment is more basal (via Gai and Pins) on the anterior side and more apical (via Dsh and Meru) on the posterior side. In reality, the planar tilt in Mud positioning is less pronounced than in the illustration, and the maximum intensity projections we use in Figure 3A and 7B allow us to capture both anterior and posterior crescents in the same image. Using a maximum intensity projection makes it easier for the reader to visualise the lack of recruitment of Mud to the posterior in the Meru^{RXL} rescue experiment in Figure 7C. We can clarify this in the figure legend.

The amounts of the relevant proteins localised looks very little and the asymmetries there but not black and white in this system.

Although the levels of cortical Mud are rather low, as pointed out by the reviewer, given the very strong SOP spindle orientation defects in *mud* mutants (PMID: 21074723), it is clear that Mud is crucial for spindle orientation in this system.

Why is Mud not orienting the spindle at the anterior? Does this mean that Mud is only competent to grab the spindle when complexed with Dsh and Meru, as it is at the apical posterior?

Both the anterior and the posterior Mud crescents are competent to capture the spindle. Indeed, the Schweisguth lab has shown that the anterior and the posterior PCP complexes act redundantly to promote normal spindle orientation (PMID: 19214234). The likely reason for the fact that disruption of the posterior complex elicits a strong A-P spindle orientation defect (whereas *pins* loss doesn't, PMID: 21074723) is that the posterior complex, which includes Baz and aPKC, is required to restrict Pins localisation posteriorly. Therefore, upon loss of the posterior complex, Pins crescent extension is thought to allow it to capture both centrosomes, increasing the likelihood spindle rotation defects (discussed in PMID: 19214234).

(minor: signal that was quantified in Fig EV1 B' 0min looks different to that of Fig 3A and 7B, is this normal variability? There is something wrong in Fig EV1 annotation I think)

Figure EV1 B' is a projection of the three apical-most slices of the cells, whereas Figure 3A and 7B are maximum intensity projections all the way through the cells, hence the difference in appearance. We will clarify this in the Figure legends.

We apologise for the mistake in annotation, we will correct it.

Point 2

In Fig 2B Mud (1825-2457) is not at all detected in the input but becomes just detectable in the IP yet it pulls down almost similar amounts of Meru-FLAG as Mud (1452-1961) which is expressed significantly more. This made me wonder whether it is purely the PBD that mediates affinity or something else in the Cter behind the PBD of Mud that helps bind Meru? Also the proteins produced in insect cells are likely to be hyperphosphorylated, does this have any implication for the interpretation of the results?

It can be difficult to conclude on the relative affinities of truncated constructs in co-IP experiments, but it does seem like the Meru pulldown is more efficient with Mud (1825-2457) than Mud (1452-1961) as pointed out by the reviewer. We propose to perform a co-IP with Mud (2089-end) to clarify if the domain C-terminal to the PBD has some affinity for Meru.

As our S2 cell cultures have a mixture of cells at different cell cycle stages, we expect that the proteins we express will be at least partially phosphorylated. According to our model, if Meru was not phosphorylated in S2 cell extracts, we would not observe a robust Mud/Meru co-IP. In agreement with this, *cycA* depletion, which is expected to reduce Cdk1 activity, reduces Mud/Meru binding (Figure EV3B). As *CycA/Cdk1* overexpression can still increase Mud/Meru interaction (EV4A), it seems not all Meru is phosphorylated in our extracts, consistent with their mixed cell cycle status. Thus, the phosphorylation status of our S2-expressed proteins is in agreement with our model.

Point 3

CylinA localisation is lost in anaphase, yet Mud and Meru stay positioned, so something needs to propagate that positional information in mitosis to prevent spindles from rotating which is likely to be the phosphorylation of Meru by CDK1. The data for Meru- CyclinA interaction and function is strong, the part invoking CDK1 itself less so. I agree it is highly likely that CDK1 activity is involved, yet there is no direct experiment to show that CDK1 is the kinase, or that Meru is even phosphorylated on the site (the point mutations may change structure and thereby affect the function). It may be interesting and not too time consuming to IP Meru-GFP from S2 cells and perform phosphoproteomics to check if the suspected sites are phosphorylated (co-express *CycA/CDK1* if needed) and whether that is lost in the RxL mutants and perhaps use this assay in some other form to probe that CDK1 can directly phosphorylate Meru on the suspected sites? It will be very difficult to demonstrate that in the tissue.

Although Cyclin A cortical localisation disappears at anaphase, the ability of Meru to be retained at the cortex after that point will be dependent on how fast it becomes

dephosphorylated. Unless the dephosphorylation is very fast, it is therefore not surprising that Meru can perdure at the cortex beyond anaphase. We do however agree with the reviewer we could strengthen the evidence for the involvement of Cdk1 by performing Mass Spectrometry on Meru in S2 cells, comparing wild type with the RXL mutant. We will attempt this approach during the revision.

Point 4

Checking what happens to Cyclin A in meru1 mutants alone and those expressing wt Meru and MeruRXL, if Cyclin A localisation lost could further strengthen the model.

We agree this would be an interesting experiment and plan to address this in the revised version.

The paragraph line 418 onwards should be in the results as it introduced hitherto not presented data and could go with figure 8 detailing the model and hinting at broader implications of aspects of the findings.

We will move these data to the result section.

Referee #2

McLelland et al. demonstrated that the N-terminal RASSF protein, Meru, bridges the Frizzled/Dishevelled planner cell-polarity complex with Mud spindle-tethering machinery using Drosophila SOPs and S2 cells. In addition, they showed that the RxL motif on Meru is required for its interaction with Cyclin A, and that mutations of putative CDK phosphorylation sites on Meru and Dsh disrupt the ternary complex formation of Mud/Meru/Dsh. Overall, the data are of high quality and the results are well documented. This study reveals a missing link that couples cell polarity with spindle orientation in a cell-cycle-dependent manner. This study will contribute to the asymmetric-cell-division field; thus, it warrants publication in EMBO, but the following matters need to be addressed.

Major comments

1. The authors propose that cyclin A/Cdk1 promotes formation of the Mud/Meru/Dsh complex by phosphorylation of Meru and Dsh. Although the authors performed IP experiments nicely using Meru5S/T-A mutants (Figure 6 and EV4), there is no direct evidence of Meru phosphorylation at these Cdk consensus sites by cyclin A/Cdk1. It is unclear whether Meru-HA shows band shifts in a Cyclin A/Cdk1-dependent manner (Figure EV4-A). Using mass spectrometry or phos-tag gels in combination with a Meru5S/T-A mutant would be useful to demonstrate phosphorylation on Meru at these sites. If direct evidence is not provided, the authors should reword lines 35, 392, 956 and others accordingly to avoid overstatements.

As suggested by this reviewer, as well as reviewer 1, we will perform Mass Spectrometry experiments in S2 cells to look at Meru phosphorylation in S2 cells.

2. Localization and functions of the Meru5S/T-A mutant have not been analyzed. Although Meru5S/T-A-GFP still interacts with Dsh in S2 cells (Figure 6), the Meru5S/T-A mutant may cause mis-localization and functional abnormalities in SOP cells. As shown for the MeruRxL mutant (Figure 7), it would be worth analyzing localization and functions of the Meru5S/T-A mutant in SOP cells in order to support the model shown in Figure 8.

We will analyse Mud localisation and spindle orientation upon Meru5S/T-A expression in the *meru* mutant background as requested by the reviewer.

Minor comments

3. In the legend to Figure 2A, the authors need to explain what RA denotes. Is this the conserved Ras-association domain?

It is the Ras-association domain, we will state this in the legend.

4. In the diagram of Figure 5A, it would be helpful if the authors indicate where the RA and coiled-coil are located. Are the important 258/259 RxL motifs located in the RA domain?

We will do this.

5. On page 8 line 157, the authors mentioned that GFP-Meru localizes to the posterior cortex at interphase, whereas they mentioned that both *meru* and cyclin A co-localize in late G2 (page 11, lines 243-244). Does Meru localize to the posterior cortex only in late G2? If Meru localizes in G1 or S, how does it localize?

We have not examined the localisation of Meru together with markers of G1 and S-phase, therefore we will amend the text to read late G2 rather than interphase.

6. Fig EV5A-C are considered in the Discussion, but they are not mentioned in the Results. The authors need to confirm whether it is permissible in EMBO to discuss figures without first describing them in the Results.

We will move these data to the results section.

7. On page 17, line 360 and others, the authors use the *meru1* mutant. It would be helpful to briefly describe characteristics of this mutant for the benefit of those who do not work in this field.

We will describe the nature of the *meru*¹ mutant in the methods.

8. On page 31, line 663, the journal name of Darnat et al., is missing.

We will fix this.

Referee #3

An evolutionarily-conserved biological machine comprised of Mud (NuMA/LIN-5) and the microtubule motor dynein localizes to the mitotic cell cortex, where it is thought to reel astral microtubules, and therefore spindles, into alignment. The question of how Mud achieves its specific cortical location has received a lot of attention and appears to have a number of different cell-specific answers. Spindle orientation is particularly important in asymmetrically-dividing cells like the *Drosophila* SOP. In this study the authors tackle a cool question raised by previous work. How is Mud recruited to the apical/posterior cortex in SOPs? Two models have been proposed; one relies on interaction between Mud and Dsh, whereas the other on Cyclin A. The authors undertake to settle the discrepancy and describe the scaffolding protein Meru as a molecular "bridge" that links Mud, Dsh, and CycA.

There's a lot to like about this paper. It's neatly written, thorough in its scholarship, and the combination of biochemistry, genetics, and live imaging approaches is commendable. Unfortunately, I didn't find the story convincing. My impression is that this study is trying to shoe-horn some nice biochemical data into a model that the genetics don't support.

We thank the reviewer for their positive comments on the approaches and scholarship. We believe that there are no discrepancies between the genetics and biochemistry and have hopefully addressed their concerns in this respect in our response below.

Major concerns:

1. I have a fundamental concern with the presentation of cortical Mud levels. It is clear from the methods that the authors put an impressive amount of thought into how this should be done, and I am therefore confident that the experimentalists were very careful. However, I very strongly disagree with a comparison of absolute pixel intensity across samples. Pixel intensity of confocal micrographs is impacted by some variables that are hard to control for, for example distance from the coverslip (even at the level of microns).

Related to this concern, I cannot find the number of nota that were imaged for each condition, only the number of pl cells.

The methods section described an "A-P ratio" and given the concern about variability

in intensity between sample preparations this seems to be the most reliable way of presenting the data. If this was the case, the differences in Figure 3C would probably disappear. Comparison of the data in Figure 3C with 7D might also be facilitated (see below).

As the reviewer states, we have been very careful in our approach to Mud quantifications. Firstly, the method for mounting pupae for imaging is very well established by many labs and yields consistent results. In particular, because the notal epidermis is apposed to the pupal cuticle on which the cuticle rests, we can ensure consistent distance between the objective and the cells. Secondly, control and experimental animals were imaged sequentially with identical settings, minimising the risk of hardware-associated variation. Finally, to account for potential variability, we always image from multiple nota (four control and four mutants for Figure 3A-C and three control and four mutants in Figure 7B-D). This was indicated as “animals” rather than nota in the figure legends, but we can change this.

We can indeed use the ratio between the anterior and posterior domains and will add these plots to the manuscript as suggested. However, as noted by the reviewer, this would obscure the fact that the intensity of both the anterior and posterior crescents is decreased in *meru*¹ mutant, therefore the ratio would be similar to control animals (Figure 3A-C). As we propose in the discussion, this is likely because Meru is limiting to maintain Mud levels. However, our model for Meru promoting posterior Mud recruitment is clearly supported by the rescue experiment where we express Meru^{WT} and Meru^{RXL} in the *meru*¹ mutant, in which case the A-P ratio is clearly increased in the mutant (Figure 7B-D, we will provide the ratios in the revised manuscript for this Figure as well).

2. What kind of allele is *meru*[1]? (This needs to be described). Ideally more than one allele/chromosome should be used to account for the possibility of more than one relevant mutation on the *meru*[1] chromosome. Is *meru*[1] the only way to disrupt Meru function genetically? Are there deletions available?

We apologise for not describing *meru*¹ in the text, which we will remedy in the revision. We characterized *meru*¹ extensively in Banerjee *et al* 2017 (PMID: 28665270). It is a strong hypomorph resulting from a deletion of exon 1. In Banerjee *et al* we showed that we could rescue the bristle defects in *meru*¹ flies with a *meru* rescue construct. Furthermore, in the present manuscript we show that spindle orientation in *meru*¹ mutants is rescued by Meru^{WT} expression (Figure 7E, G). Therefore, we believe the defects we are observing are due to *meru* loss. There are larger deletions that also take out other genes in the region and would therefore not be appropriate to study *meru*.

3. I had trouble with the interpretation of Figure 3. If *meru*[1] is a null, then Meru is

clearly not required for the cortical localization of Mud (as claimed in the text). At best it contributes, but that interpretation is based on a comparison of pixel intensity across samples, which is problematic as described above. More importantly, the authors point out that loss of Meru function does not impact asymmetry. These data therefore do not agree with the authors' model, which is that posterior Meru links Dsh with Mud.

As mentioned above, *meru*¹ is a strong hypomorph. In the discussion, we state that Meru promotes posterior cortical Mud recruitment (rather than is strictly required for it), but we acknowledge the fact that other Mud cortical recruitment mechanisms exist, for instance via Pins. We can change the text where relevant to “Meru promotes Mud posterior enrichment”, which is consistent with the data. The fact that Meru loss impacts both anterior and posterior Mud recruitment is something we address in the discussion. As stated above, we believe this can be explained by an effect of Meru loss on Mud levels. The clear-cut effect of the RXL mutation on posterior Mud recruitment in Figure 7B-D also argues in favour of our model.

Related to this, the authors report that spindle orientation in *meru* mutants is less planar (more tilted) than the wt. That is the *opposite* phenotype to *frizzled* and *dsh* mutants, which are more planar than wt (Ségalan). The authors need to address this contradiction.

This is an interesting point that we can add in the revised discussion. Darnat *et al* have previously shown that, consistent with the *meru* mutants, *cyclin A* depletion also leads to divisions that are more tilted than controls (Figure 4H in PMID: 35581185). Why is this different from the *fz* and *dsh* mutant, where the divisions are more planar? The published work on PCP genes in SOPs takes advantage of the fact that the PCP-specific *dsh*¹ allele and *fz* mutants are homozygous viable, therefore all the analyses on SOP division were performed on whole mutant animals. Because the anterior and posterior PCP complexes are known to stabilise each other across cell junctions as well as antagonise each other intracellularly, global loss of *dsh* or *fz* perturbs the organisation of both the anterior and posterior cortices in SOPs which no longer receive PCP input from their neighbours on the anterior side. In these conditions, the cell divisions become planar since spindle tilt is lost (PMID: 21074723). As depletion of *meru* and *cyclin A* do not affect PCP in neighbouring cells, the basal pulling force from Pins/Mud remains and is no longer counterbalanced by the apical posterior side and spindle tilt increases (PMID: 35581185).

Another consideration is that the cortical localization of Mud being shown in Figure 3A is at anaphase. Is this the point being measured in C? This might impact interpretation, since Mud-dependent spindle orientation has already been achieved by anaphase.

We chose to perform the Mud measurements at anaphase because in SOPs the centrosomes are pulled towards the cortical domains at anaphase, suggesting this is the relevant stage for Mud function (PMID: 25619594; PMID: 19214234), and spindle angle can still change prior to that stage.

4. "In the rescued conditions, both the anterior and posterior cortices had nearly identical levels of Mud when mK2-MeruWT was expressed (Figure 7B, B' and D). It is interesting to note that, in the absence of Meru overexpression, Mud is more enriched at the anterior than the posterior cortex (Fig 3C). The equal cortical distribution of Mud upon mK2-MeruWT overexpression is therefore consistent with Meru promoting Mud posterior recruitment."

One concern raised with this interpretation is that the authors do not present evidence that Meru[WT] is being overexpressed, only that it is being expressed. How strong is normal expression vs. the GAL4-mediated expression?

The *neur^{P72}-GAL4* driver we use has been widely characterised and generally gives robust expression in SOPs, hence our assumption that the *meru* constructs are overexpressed. This issue would be difficult to address by Western blotting from fly extracts as Meru is expressed only in SOPs. Another reason to believe the UAS-mKate::Meru is overexpressed is that we can clearly detect it, whereas the mKate::Meru knockin line we have made is barely detectable.

We also have made a GFP::Meru knockin line (PMID: 28665270), which is readily detectable, however, we cannot compare this with the mKate::Meru transgenics as these are tagged with a different fluorophore. We do have a GFP::Meru UAS line inserted at the same locus as the UAS-mKate::Meru transgenes. If the reviewer and editor believe it would add value to the manuscript, we could compare the GFP knockin animals with UAS-GFP::Meru driven by *neur^{P72}-GAL4*.

Figure 7D shows that expression of Meru[WT] evens out the Mud asymmetry observed in wild type SOPs. I'm confused by the interpretation of these data, which is that Meru[WT] promotes Mud posterior recruitment. I don't think there's enough information to make that claim. Firstly, Meru[WT] is observed at both the anterior and posterior. Secondly, the authors made a convincing case in Figure 1G that Meru and Mud colocalise at confocal resolution. This does not appear to be strictly true in the Meru[WT] condition. They overlap somewhat but intensities don't match. Thirdly, there is no evidence to show where Dsh is in this condition. Finally, we have been told previously (Figure 3C) to compare absolute pixel intensity between conditions. If we do that here, comparing 7D to 3C yields the straightforward conclusion that Meru[WT] interferes with the anterior localization of Mud, which is reduced in comparison to wt cells, whereas posterior localization is unaffected.

“Firstly, Meru[WT] is observed at both the anterior and posterior.”

While it is true that there is some Meru on the anterior side (perhaps as a result of overexpression), Meru remains strongly posteriorly biased in this experiment.

“Secondly, the authors made a convincing case in Figure 1G that Meru and Mud colocalise at confocal resolution. This does not appear to be strictly true in the Meru[WT] condition. They overlap somewhat but intensities don't match.”

Figure 1G is quantified from a single confocal slice, whereas Figure 7B (Meru[WT]) is a maximum intensity projection so the reader can see both anterior and posterior crescent in the same image. Therefore, it is not surprising that the colocalization is less perfect.

“Thirdly, there is no evidence to show where Dsh is in this condition.”

While this is an interesting question we do not make any claims about Dsh localisation and it would be technically difficult to reliably follow Dsh localisation during mitosis with available reagents.

“Finally, we have been told previously (Figure 3C) to compare absolute pixel intensity between conditions. If we do that here, comparing 7D to 3C yields the straightforward conclusion that Meru[WT] interferes with the anterior localization of Mud, which is reduced in comparison to wt cells, whereas posterior localization is unaffected.”

Our experiments to measure Mud localisation are carefully set up in order to compare each experimental with its matching control (e.g. *meru* mutant versus wild type in Figure 3C and Meru^{RXL} versus Meru^{WT} in Figure 7D). However, it is not possible to compare the fluorescence values in Figure 3C with 7D because these were done at different times, and the gain settings were different to account for the elevated levels of Mud at the cortex in Meru overexpression conditions, likely the result of Mud stabilisation by the presence of overexpressed Meru.

Spindle angles in Figure 7 should be juxtaposed with those in Figure 3 and the appropriate statistical comparisons made. The authors state that Meru[WT] rescues A-P spindle orientation, which appears to be the case, but I can't tell whether Meru[RxL] is defective to the same extent as *meru*[1]. Most spindle angles in the Meru[RxL] condition are below 45 degrees, with the largest population below 15 degrees (as in the control). That looks quite a bit different from *meru*[1], arguing that Meru[RxL] rescues at least partially. Related to this, there is no A-B spindle orientation defect in the Meru[RxL] condition.

It is true that in the Meru^{RXL} conditions there is a slight improvement of the A-P spindle orientation compared to the *meru* mutant, but it is not statistically significant ($p = 0.056$). Meru has several RXL motifs besides the ones we are mutating and

therefore there is likely residual Cyclin A binding, which we see in several co-IP experiments. This might explain why Meru^{RXL} retains weak activity. We can state this in the revised manuscript. Nevertheless, Meru^{RXL} has a robust phenotype compared with Meru^{WT}, supporting our conclusions that Cyclin A/Meru binding play a role in spindle orientation. For the A-B tilt, as outlined by the reviewer below, this is more difficult to measure and therefore the effects can be masked by noise.

Minor comments:

Mud and Dsh interact in HEK293T cells. The authors propose that this is explained by the expression of Meru orthologs. It's a very compelling idea that really should be pushed further. Can it be tested? The authors could at least mention how similar these orthologs are to Meru.

It would be interesting to study the function of the Meru orthologs, but we feel that this is out of the scope of the present manuscript. Our data demonstrating that Meru is required for Mud/Dsh association in *Drosophila* S2 cells is thoroughly controlled and convincingly backs our model without the need for validation in an exogenous system like HEK293 cells.

It's not clear to me what the live images are. Z-stacks? The authors point out in the Methods that they measured posterior and anterior localizations separately because they are not at the same depth. A specific concern is that the images shown in, for example, Figure 7C, are not representative of the quantifications in 7D.

As stated in the materials and methods, the measurements in 3C and 7D are taken from three apical and basolateral slices, whereas the images in 3A-B and 7B-C are maximum intensity projections to allow the reader to see both the anterior and posterior crescent in the same picture. We can clarify this in the figure legends.

Additional suggestions/questions:

The cartoon in Figure 1A illustrates an approximately 45 degree spindle tilt in the A-B axis. The measured tilt is 6.8 degrees. I do not dispute a "characteristic z-tilt" (Line 231) in SOP cells, but 6.8 is close enough to zero that a reasonable explanation for that number in these measurements is simple noise (whether biological or technical) in the data. This is especially true given that all data points are positive. (You'd have to measure only zeros to get a zero average). Can the authors comment on this?

We agree that the Figure 1A diagram shows an exaggerated view of the tilt for the purpose of illustration and the tilt is in reality more subtle. We have measured the z-tilt in order to be thorough in our analysis, as this has been done in previous work (e.g. PMID: 16228010; PMID: 21074723; PMID: 35581185). However, unlike spindle orientation in the A-P direction, which impacts cell fate determinant segregation and therefore the differentiation of the SOP lineage, the biological relevance of the z-tilt to

sensory organ differentiation is unknown. Thus, we feel the A-P phenotype, which is far easier to measure accurately, is of more functional importance. We agree with the reviewer that the z-tilt is subject to noise, although we have taken care to perform the measurement as accurately as possible.

The model these authors present (summarized in Figure 8) is that Meru links Mud to Dsh. While I don't consider it a requirement, it would be nice to see evidence that Meru and Dsh colocalise at confocal resolution, as in Figure 1G.

We have previously shown (PMID: 28665270) that Meru and Dsh colocalise, and that *dsh* and *fz* are required for Meru cortical recruitment.

I found the juxtaposed rose plots in figures 3 and 7 a little puzzling (at first) and ultimately hard to interpret. I suggest that the authors could find a different way to plot the data (e.g. violin plots). This would also facilitate comparison between figures.

Rose plots are widely used to visualise angular data. If the reviewer wishes we can prepare different ways of representing it in the revised version, but we feel the rose plots are easier to navigate.

What is that big ball of Pins[RxL] in the middle of the cell!?

It is the nucleus.

Dr. Nicolas Tapon
The Francis Crick Institute
1 Midland Road
London NW1 1AT
United Kingdom

24th Feb 2024

Re: EMBOJ-2023-116230
Meru co-ordinates spindle orientation with cell polarity and cell cycle progression

Dear Nic,

Thank you for sending your detailed tentative responses to the referee reports on your recent EMBO Journal submission. With some delay due to conference travel, I have now had a chance to carefully go through all points, and concluded that we shall be happy to formally invite a revision modified along the lines proposed in your letter. Regarding experimental revisions, I appreciate the adding the mass-spectrometry phosphorylation studies will be key to answering a shared concern of referees 1 and 2, and that the co-IP experiment in response to referee 1 and the localization/spindle orientation studies in response to referee 2 should also be very helpful. Related to referee 3's criticisms, I realize that they can to large parts be addressed through clarifications and altered presentation as proposed, and would therefore recommend to proceed accordingly; studies with a GFP::Meru construct as alternative evidence for Meru overexpression would in my view not be adding substantially to the study.

Please keep in mind that it is our policy to allow only a single round of (major) revision, and I would therefore encourage you to update me should there be any unexpected problems with the revisions, or should you require an extension beyond the default 3-months deadline. As always, competing manuscript published during the course of this revision will not affect our final decision on your study. Finally, please note the detailed information and guidelines on how to prepare a revision below (and in our online Guide to Authors) - closely adhering to them shall greatly facilitate the editorial process at the time of resubmission.

Thank you again for the opportunity to consider this work, and I look forward to receiving your revision in due time.

With kind regards,

Hartmut

4) Each main and each Expanded View (EV) figure should be uploaded as individual production-quality files (preferably in .eps,

.tif, .jpg formats). For suggestions on figure preparation/layout, please refer to our Figure Preparation Guidelines: <http://bit.ly/EMBOPressFigurePreparationGuideline>

9) Digital image enhancement is acceptable practice, as long as it accurately represents the original data and conforms to community standards. If a figure has been subjected to significant electronic manipulation, this must be clearly noted in the figure legend and/or the 'Materials and Methods' section. The editors reserve the right to request original versions of figures and the original images that were used to assemble the figure. Finally, we generally encourage uploading of numerical as well as gel/blot image source data; for details see: embopress.org/page/journal/14602075/authorguide#sourcedata

At EMBO Press, we ask authors to provide source data for the main manuscript figures. Our source data coordinator will contact you to discuss which figure panels we would need source data for and will also provide you with helpful tips on how to upload and organize the files.

In the interest of ensuring the conceptual advance provided by the work, we recommend submitting a revision within 3 months (24th May 2024). Please discuss the revision progress ahead of this time with the editor if you require more time to complete the revisions. Use the link below to submit your revision:

Link Not Available

The Francis Crick Institute Laboratory 1 Midland Road London NW1 1AT
 +44 (0)203 796 0000 info@crick.ac.uk www.crick.ac.uk

Dr Nicolas Tapon
Principal Group Leader
The Francis Crick Institute
 1 Midland Road
 London NW1 1AT
 United Kingdom
 T +44 (0)203 796 2050 (off.)
 F +44 (0)203 796 1253 (lab)
 E nic.tapon@crick.ac.uk
<http://www.crick.ac.uk/nic-tapon>

London, 13 February 25

McLellan et al. EMBOJ-2023-116230

Dear Hartmut,

I am delighted to submit our revised manuscript entitled “**Meru co-ordinates spindle orientation with cell polarity and cell cycle progression**” (EMBOJ-2023-116230). We are very grateful to the referees whose comments have helped us substantially improve the manuscript. Below is a summary of the new data we have added, followed by a point-by-point response to the reviewers.

I look forward to hearing from you.

Best regards,
 Nic Tapon

New data summary

- To provide evidence that Meru is a Cdk1 target, we used Mass Spectrometry to show that CycA/Cdk1-induced Meru phosphorylation is dependent on the Cyclin docking site (RxL motif – Fig. 7A, B and EV4B).
- We analysed the localisation of CycA in cells expressing the Meru^{RxL} mutant and show that the RxL Cyclin docking site on Meru is required for CycA posterior localisation *in vivo* (Fig. 5F-G' and EV3D-D').
- We showed that cells expressing the Meru mutant lacking the Cdk1 phosphorylation sites (Meru^{5S/T-A}) have delayed Mud posterior recruitment (Fig. 8A-C, EV5E-J) and defective A-P spindle orientation (Fig. 8D-F'). The posterior cortical recruitment of this mutant Meru is also compromised compared with wild type Meru (Fig. EV5 A-D).
- We analysed two new Mud deletions for Meru binding by co-IP (Fig. 2B) and showed that Meru/Mud association is mediated both by the Mud PBD (Pins-binding domain) and C-terminus (aa1951-2456).

Response to referees

Referee #1

Point 1

Meru and Mud localisation look convincing. Regarding Mud and spindle orientation in SOPs: Fig 1 A suggest Mud to be basal at the anterior and apical at the posterior. Yet in Fig 3A and 7B Mud seems rather planar polarised at anaphase.

In the Figure 1A diagram, we illustrate the characteristic tilt of the spindle to make it obvious to the reader that Mud is planar polarised along the A-P axis, and also that its recruitment is more basal (via Gai and Pins) on the anterior side and more apical (via Dsh and Meru) on the posterior side. In reality, the planar tilt in Mud positioning is less pronounced than in the illustration, and the maximum intensity projections we use in Figure 3A and 7B allow us to capture both anterior and posterior crescents in the same image. Using a maximum intensity projection makes it easier for the reader to visualise the lack of Mud recruitment to the posterior in the Meru^{RXL} rescue experiment in Figure 7C. We have clarified in the legend to Figure 1A: "Note that the extent of spindle tilt in the diagram is amplified compared with reality for illustrative purposes."

The amounts of the relevant proteins localised looks very little and the asymmetries there but not black and white in this system.

Although the levels of cortical Mud are rather low, as pointed out by the reviewer, given the very strong SOP spindle orientation defects in *mud* mutants (PMID: 21074723), it is clear that Mud is crucial for spindle orientation in this system, as it is in many other cell types.

Why is Mud not orienting the spindle at the anterior? Does this mean that Mud is only competent to grab the spindle when complexed with Dsh and Meru, as it is at the apical posterior?

Both the anterior and the posterior Mud crescents are competent to capture the spindle. Indeed, the Schweisguth lab has shown that the anterior and the posterior PCP complexes act redundantly to promote normal spindle orientation (PMID: 19214234). The likely reason for the fact that disruption of the posterior complex elicits a strong A-P spindle orientation defect (whereas *pins* loss doesn't, PMID: 21074723) is that the posterior complex, which includes Baz and aPKC, is required to restrict Pins localisation posteriorly. Therefore, upon loss of the posterior complex, Pins crescent extension is thought to allow it to capture both centrosomes, increasing the likelihood spindle rotation defects (discussed in PMID: 19214234).

(minor: signal that was quantified in Fig EV1 B' 0min looks different to that of Fig 3A and 7B, is this normal variability? There is something wrong in Fig EV1 annotation I think)

Figure EV1 B' is a projection of the three apical-most slices of the cells, whereas Figure 3A and 7B (6B in the revised manuscript) are maximum intensity projections all the way through the cells, hence the difference in appearance. We have clarified this in the Figure legend to Fig. EV1B' and in the rest of the figure legends

We have checked the annotation for Figure EV1.

Point 2

In Fig 2B Mud (1825-2457) is not at all detected in the input but becomes just detectable in the IP yet it pulls down almost similar amounts of Meru-FLAG as Mud (1452-1961) which is expressed significantly more. This made me wonder whether it is purely the PBD that mediates affinity or something else in the Cter behind the PBD of Mud that helps bind Meru? Also the proteins produced in insect cells are likely to be hyperphosphorylated, does this have any implication for the interpretation of the results?

As suggested by the reviewer, we have analysed further truncations to investigate whether sequences C-terminal to the PBD also participate in the Meru/Mud interaction (new Fig. 2B, which replaces the previous version). Two new truncations lacking the PBD, Mud (1951-2456) and Mud (2089-2499) can both co-immunoprecipitate Meru, though not as efficiently as Mud (1825-2456), which has both the PBD and C-terminus and, as pointed out by the reviewer, seemed to associate with Meru better than Mud (1452-1961), which contains the PBD and some of the N-terminal coiled-coils. This suggests that Meru/Mud association is mediated both by the Mud PBD and C-terminus (aa1951-2456). We have amended the results accordingly.

In terms of phosphorylation status, as our S2 cell cultures have a mixture of cells at different cell cycle stages, we expect that the proteins we express will be at least partially phosphorylated. According to our model, if Meru was not phosphorylated in S2 cell extracts, we would not observe a robust Mud/Meru co-IP. In agreement with this, *cycA* depletion, which is expected to reduce endogenous Cdk1 activity, reduces Mud/Meru binding (Figure EV2B). As *CycA/Cdk1* overexpression can still increase Mud/Meru interaction (Figure EV4A), it seems not all Meru is phosphorylated in our extracts, consistent with their mixed cell cycle status. Thus, the phosphorylation status of our S2-expressed proteins is in agreement with our model.

Point 3

CylinA localisation is lost in anaphase, yet Mud and Meru stay positioned, so something needs to propagate that positional information in mitosis to prevent spindles from rotating which is likely to be the phosphorylation of Meru by CDK1. The data for Meru- CyclinA interaction and function is strong, the part invoking CDK1 itself less so. I agree it is highly likely that CDK1 activity is involved, yet there is no direct experiment to show that CDK1 is the kinase, or that Meru is even phosphorylated on the site (the point mutations may change structure and thereby affect the function). It may be interesting and not too time

consuming to IP Meru-GFP from S2 cells and perform phosphoproteomics to check if the suspected sites are phosphorylated (co-express CycA/CDK1 if needed) and whether that is lost in the RxL mutants and perhaps use this assay in some other form to probe that CDK1 can directly phosphorylate Meru on the suspected sites? It will be very difficult to demonstrate that in the tissue.

Although Cyclin A cortical localisation disappears at anaphase, the ability of Meru to be retained at the cortex after that point will be dependent on how fast it becomes dephosphorylated. Unless the dephosphorylation is very fast, it is therefore not surprising that Meru can perdure at the cortex beyond anaphase.

As suggested by the reviewer, we used Mass Spectrometry phosphoproteomics to compare phosphorylation of Meru and Meru^{RXL} in S2 cells expressing Cyclin A/Cdk1 (Fig. 7A, B and EV4B). Using trypsin digest, we were able to perform label-free quantification on two of the Cdk1 sites (S60 and T496), which were both detectably phosphorylated in wild type Meru, but not Meru^{RXL}, consistent with our model (Fig. 7A, B).

Point 4

Checking what happens to Cyclin A in meru1 mutants alone and those expressing wt Meru and MeruRxL, if Cyclin A localisation lost could further strengthen the model.

As suggested, we looked at Cyclin A localisation in meru1 mutants expressing either wild type Meru or Meru^{RXL} (Fig. 5F-G, EV3D-D'). Consistent with a role for Meru in recruiting Cyclin A to the posterior cortex, Meru^{RXL} expression severely reduced the Cyclin A posterior crescent compared to Meru^{WT} (Fig. 5F-G, EV3D-D').

The paragraph line 418 onwards should be in the results as it introduced hitherto not presented data and could go with figure 8 detailing the model and hinting at broader implications of aspects of the findings.

We have moved these data to the results section as requested.

Referee #2

Major comments

1. The authors propose that cyclin A/Cdk1 promotes formation of the Mud/Meru/Dsh complex by phosphorylation of Meru and Dsh. Although the authors performed IP experiments nicely using Meru5S/T-A mutants (Figure 6 and EV4), there is no direct evidence of Meru phosphorylation at these Cdk consensus sites by cyclin A/Cdk1. It is unclear whether Meru-HA shows band shifts in a Cyclin A/Cdk1-dependent manner (Figure EV4-A). Using mass spectrometry or phos-tag gels in combination with a Meru5S/T-A mutant would be useful to demonstrate phosphorylation on Meru at these sites. If

direct evidence is not provided, the authors should reword lines 35, 392, 956 and others accordingly to avoid overstatements.

As suggested by the reviewer, we used Mass Spectrometry phosphoproteomics to compare phosphorylation of Meru and Meru^{RxL} in S2 cells expressing Cyclin A/Cdk1 (Fig. 7A, B and EV4B). Using trypsin digest, we were able to perform label-free quantification on two of the Cdk1 sites (S60 and T496), which were both detectably phosphorylated in wild type Meru, but not Meru^{RxL}, consistent with our model (Fig. 7A, B).

2. Localization and functions of the Meru5S/T-A mutant have not been analyzed. Although Meru5S/T-A-GFP still interacts with Dsh in S2 cells (Figure 6), the Meru5S/T-A mutant may cause mis-localization and functional abnormalities in SOP cells. As shown for the MeruRxL mutant (Figure 7), it would be worth analyzing localization and functions of the Meru5S/T-A mutant in SOP cells in order to support the model shown in Figure 8.

As suggested by the reviewer, we have analysed the localisation and function of the Meru^{5S/T-A} *in vivo* by expressing it in the *meru*¹ mutant background. Although unlike Meru^{RxL} (Fig. 5F), Meru^{5S/T-A} could still form a detectable posterior crescent in early G2 (Fig. EV5B-B'), this was significantly reduced compared with Meru^{WT} (Fig EV5A'-C). In addition, we observed a reduction in the characteristic Meru puncta that form at the posterior crescent at this stage (EV5D). We then examined Meru^{5S/T-A} function by looking at Mud localisation (Fig. 8A-C, EV5E-J) and spindle orientation (Fig. 8D-F'). Interestingly, while Mud posterior localisation is reduced in Meru^{5S/T-A} at prophase (Fig. EV5E-G) and metaphase (Fig. 8A-C), by anaphase (Fig. EV5H-J), we cannot detect a significant change in Mud posterior localisation. Accordingly, Meru^{5S/T-A}-expressing animals display a significant A-P spindle rotation defect (Fig. 8D-F') that was nevertheless weaker than *meru* mutants (Fig. 3D-F'). This suggests that Meru^{5S/T-A} is partially defective and delayed in its ability to recruit Mud, consistent with reduced Meru^{5S/T-A}/Mud binding in S2 cells (Fig. 7C). A partial loss-of-function is expected, since Meru^{5S/T-A} is still able to bind Dsh and Baz in S2 cells (Fig. 7C) and should still be able to recruit Cyclin A. Thus, we expect that other Cdk1 targets in the posterior complex such as Dsh and possibly Mud can still be phosphorylated in Meru^{5S/T-A} animals, likely explaining the partial effect.

Minor comments

3. In the legend to Figure 2A, the authors need to explain what RA denotes. Is this the conserved Ras-association domain?

It is the Ras-association domain, we have stated this in the Fig. 2A legend.

4. In the diagram of Figure 5A, it would be helpful if the authors indicate where the RA and coiled-coil are located. Are the important 258/259 RxL motifs located in the RA domain?

We have indicated the domains in the Figure as requested.

5. On page 8 line 157, the authors mentioned that GFP-Meru localizes to the posterior cortex at interphase, whereas they mentioned that both meru and cyclin A co-localize in late G2 (page 11, lines 243-244). Does Meru localize to the posterior cortex only in late G2? If Meru localizes in G1 or S, how does it localize?

We have not examined the localisation of Meru together with markers of G1 and S-phase, therefore we have amended the text to read late G2 rather than interphase.

6. Fig EV5A-C are considered in the Discussion, but they are not mentioned in the Results. The authors need to confirm whether it is permissible in EMBO to discuss figures without first describing them in the Results.

We have moved these data to the end of the results section.

7. On page 17, line 360 and others, the authors use the meru1 mutant. It would be helpful to briefly describe characteristics of this mutant for the benefit of those who do not work in this field.

We have described the nature of the *meru*¹ mutant in the results (lines 214-5). It is a 1.6 kb deletion that removes half the coding region including the RA domain (PMID: 28665270).

8. On page 31, line 663, the journal name of Darnat et al., is missing.

We have fixed this.

Referee #3

An evolutionarily-conserved biological machine comprised of Mud (NuMA/LIN-5) and the microtubule motor dynein localizes to the mitotic cell cortex, where it is thought to reel astral microtubules, and therefore spindles, into alignment. The question of how Mud achieves its specific cortical location has received a lot of attention and appears to have a number of different cell-specific answers. Spindle orientation is particularly important in asymmetrically-dividing cells like the *Drosophila* SOP. In this study the authors tackle a cool question raised by previous work. How is Mud recruited to the apical/posterior cortex in SOPs? Two models have been proposed; one relies on interaction between Mud and Dsh, whereas the other on Cyclin A. The authors undertake to settle the discrepancy and describe the scaffolding protein Meru as a molecular "bridge" that links Mud, Dsh, and CycA.

There's a lot to like about this paper. It's neatly written, thorough in its scholarship, and the combination

of biochemistry, genetics, and live imaging approaches is commendable. Unfortunately, I didn't find the story convincing. My impression is that this study is trying to shoe-horn some nice biochemical data into a model that the genetics don't support.

We thank the reviewer for their positive comments on the approaches and scholarship. We believe that there are no discrepancies between the genetics and biochemistry and have hopefully addressed their concerns in this respect in our response below.

Major concerns:

1. I have a fundamental concern with the presentation of cortical Mud levels. It is clear from the methods that the authors put an impressive amount of thought into how this should be done, and I am therefore confident that the experimentalists were very careful. However, I very strongly disagree with a comparison of absolute pixel intensity across samples. Pixel intensity of confocal micrographs is impacted by some variables that are hard to control for, for example distance from the coverslip (even at the level of microns).

Related to this concern, I cannot find the number of nota that were imaged for each condition, only the number of pl cells.

The methods section described an "A-P ratio" and given the concern about variability in intensity between sample preparations this seems to be the most reliable way of presenting the data. If this was the case, the differences in Figure 3C would probably disappear. Comparison of the data in Figure 3C with 7D might also be facilitated (see below).

As the reviewer states, we have been very careful in our approach to Mud quantifications. Firstly, the method for mounting pupae for imaging is very well established by many labs and yields consistent results. In particular, because the notal epidermis is apposed to the pupal cuticle on which the cuticle rests, we can ensure consistent distance between the objective and the cells. Secondly, control and experimental animals were imaged sequentially with identical settings, minimising the risk of hardware-associated variation. Finally, to account for potential variability, we always image from multiple nota (four control and four mutants for Figure 3A-C and three control and four mutants in Figure 7B-D). This was indicated as "animals" rather than nota in the figure legends, but we have changed this to clarify.

It is indeed possible to use the ratio between the anterior and posterior domains. However, as noted by the reviewer, this would obscure the fact that the intensity of both the anterior and posterior crescents is decreased in *meru*¹ mutant, therefore the ratio would be similar to control animals (Figure 3A-C). As we propose in the discussion, this is likely because Meru is limiting to maintain Mud levels. However, our model for Meru promoting posterior Mud recruitment is clearly supported by the rescue experiment

where we express Meru^{WT} and Meru^{RXL} in the *meru*¹ mutant, in which case the A-P ratio would be increased in the mutant (Figure 6B-D).

2. What kind of allele is *meru*[1]? (This needs to be described). Ideally more than one allele/chromosome should be used to account for the possibility of more than one relevant mutation on the *meru*[1] chromosome. Is *meru*[1] the only way to disrupt Meru function genetically? Are there deletions available?

We apologise for not describing *meru*¹ in the text, which we have remedied in the revision. We characterized *meru*¹ extensively in Banerjee *et al* 2017 (PMID: 28665270). It is a strong loss-of-function allele resulting from a 1.6 kb deletion that removes half the coding region including the RA domain. In Banerjee *et al* we showed that we could rescue the bristle defects in *meru*¹ flies with a *meru* rescue construct. Furthermore, in the present manuscript we show that spindle orientation in *meru*¹ mutants is rescued by Meru^{WT} expression (Figure 6E, G-G'). Therefore, we believe the defects we are observing are due to *meru* loss. There are larger deletions that also take out other genes in the region and would therefore not be appropriate to study *meru*.

3. I had trouble with the interpretation of Figure 3. If *meru*[1] is a null, then Meru is clearly not required for the cortical localization of Mud (as claimed in the text). At best it contributes, but that interpretation is based on a comparison of pixel intensity across samples, which is problematic as described above. More importantly, the authors point out that loss of Meru function does not impact asymmetry. These data therefore do not agree with the authors' model, which is that posterior Meru links Dsh with Mud.

As mentioned above, *meru*¹ is a strong loss-of-function allele. We acknowledge the fact that Meru-independent Mud cortical recruitment mechanisms exist, for instance via Pins. We have changed to text to consistently state that Meru promotes Mud posterior recruitment rather than being required for it, which is consistent with the data. The fact that Meru loss impacts both anterior and posterior Mud recruitment is something we address in the discussion. As stated above, we believe this can be explained by an effect of Meru loss on Mud levels. The clear-cut effect of the R_xL mutation on posterior Mud recruitment in Figure 6B-D also argues in favour of our model.

Related to this, the authors report that spindle orientation in *meru* mutants is less planar (more tilted) than the wt. That is the *opposite* phenotype to *frizzled* and *dsh* mutants, which are more planar than wt (Ségalan). The authors need to address this contradiction.

This is an interesting point that we now address in the results (lines 236-248). Darnat *et al* have previously shown that, consistent with the *meru* mutants, *cyclin A* depletion also leads to divisions that are more tilted than controls (Figure 4H in PMID: 35581185). Why is this different from the *fz* and *dsh* mutant, where the divisions are more planar? The published work on PCP genes in SOPs takes advantage of the fact that the PCP-specific *dsh*¹ allele and *fz* mutants are homozygous viable, therefore

all the analyses on SOP division were performed on whole mutant animals. Because the anterior and posterior PCP complexes are known to stabilise each other across cell junctions as well as antagonise each other intracellularly, global loss of *dsh* or *fz* perturbs the organisation of both the anterior and posterior cortices in SOPs which no longer receive PCP input from their neighbours on the anterior side. In these conditions, the cell divisions become planar since spindle tilt is lost (PMID: 21074723). As depletion of *meru* and *cyclin A* do not affect PCP in neighbouring cells, the basal pulling force from Pins/Mud remains and is no longer counterbalanced by the apical posterior side and spindle tilt increases (PMID: 35581185).

Another consideration is that the cortical localization of Mud being shown in Figure 3A is at anaphase. Is this the point being measured in C? This might impact interpretation, since Mud-dependent spindle orientation has already been achieved by anaphase.

We chose to perform Mud measurements at anaphase because in SOPs the centrosomes are pulled towards the cortical domains at anaphase and spindle angle can still change until metaphase (PMID: 25619594; PMID: 19214234).

4. "In the rescued conditions, both the anterior and posterior cortices had nearly identical levels of Mud when mK2-MeruWT was expressed (Figure 7B, B' and D). It is interesting to note that, in the absence of Meru overexpression, Mud is more enriched at the anterior than the posterior cortex (Fig 3C). The equal cortical distribution of Mud upon mK2-MeruWT overexpression is therefore consistent with Meru promoting Mud posterior recruitment."

One concern raised with this interpretation is that the authors do not present evidence that Meru[WT] is being overexpressed, only that it is being expressed. How strong is normal expression vs. the GAL4-mediated expression?

The *neur^{P72}-GAL4* driver we use has been widely characterised and generally gives robust expression in SOPs, hence our assumption that the *meru* constructs are overexpressed. Another reason to believe the UAS-mKate::Meru is overexpressed is that we can easily image it, whereas the mKate::Meru knockin line we have made is barely detectable by spinning disc microscopy.

Figure 7D shows that expression of Meru[WT] evens out the Mud asymmetry observed in wild type SOPs. I'm confused by the interpretation of these data, which is that Meru[WT] promotes Mud posterior recruitment. I don't think there's enough information to make that claim. Firstly, Meru[WT] is observed at both the anterior and posterior. Secondly, the authors made a convincing case in Figure 1G that Meru and Mud colocalise at confocal resolution. This does not appear to be strictly true in the Meru[WT] condition. They overlap somewhat but intensities don't match. Thirdly, there is no evidence to show where Dsh is in this condition. Finally, we have been told previously (Figure 3C) to compare absolute

pixel intensity between conditions. If we do that here, comparing 7D to 3C yields the straightforward conclusion that Mera[WT] interferes with the anterior localization of Mud, which is reduced in comparison to wt cells, whereas posterior localization is unaffected.

“Firstly, Meru[WT] is observed at both the anterior and posterior.”

While it is true that there is some Meru on the anterior side (perhaps as a result of overexpression), Meru remains strongly posteriorly biased in this experiment.

“Secondly, the authors made a convincing case in Figure 1G that Meru and Mud colocalise at confocal resolution. This does not appear to be strictly true in the Meru[WT] condition. They overlap somewhat but intensities don't match.”

Figure 1G is quantified from a single confocal slice, whereas Figure 7B (which is now Figure 6B in the revised manuscript) (Meru[WT]) is a maximum intensity projection so the reader can see both anterior and posterior crescent in the same image. Therefore, it is not surprising that the colocalization is less perfect. We have clarified this in the figure legends.

“Thirdly, there is no evidence to show where Dsh is in this condition.”

While this is an interesting question, we do not make any claims about Dsh localisation and it would be technically difficult to reliably follow Dsh localisation during mitosis with available reagents.

“Finally, we have been told previously (Figure 3C) to compare absolute pixel intensity between conditions. If we do that here, comparing 7D to 3C yields the straightforward conclusion that Mera[WT] interferes with the anterior localization of Mud, which is reduced in comparison to wt cells, whereas posterior localization is unaffected.”

Our experiments to measure Mud localisation are carefully set up in order to compare each experimental with its matching control (e.g. *meru* mutant versus wild type in Figure 3C and Meru^{RXL} versus Meru^{WT} in Figure 7D – now Figure 6D). However, it is not possible to compare the fluorescence values in Figure 3C with 6D because these were done at different times, and the gain settings were different to account for the elevated levels of Mud at the cortex in Meru overexpression conditions, likely the result of Mud stabilisation by the presence of overexpressed Meru.

Spindle angles in Figure 7 should be juxtaposed with those in Figure 3 and the appropriate statistical comparisons made. The authors state that Meru[WT] rescues A-P spindle orientation, which appears to be the case, but I can't tell whether Meru[RxL] is defective to the same extent as *meru*[1]. Most spindle angles in the Meru[RxL] condition are below 45 degrees, with the largest population below 15 degrees (as in the control). That looks quite a bit different from *meru*[1], arguing that Meru[RxL] rescues at least partially. Related to this, there is no A-B spindle orientation defect in the Meru[RxL] condition.

It is true that in the Meru^{RXL} conditions there is a slight improvement of the A-P spindle orientation compared to the *meru* mutant, but it is not statistically significant ($p = 0.056$). Meru has four RxL motifs besides the ones we are mutating and therefore there is likely residual Cyclin A binding, which we see in several co-IP experiments. This might explain why Meru^{RXL} retains weak activity. We have now indicated this in the results (lines 320-323) and discussion (lines 462-467). Nevertheless, Meru^{RXL} has a robust phenotype compared with Meru^{WT}, supporting our conclusions that Cyclin A/Meru binding play a role in spindle orientation. For the A-B tilt, as outlined by the reviewer below, this is more difficult to measure and therefore the effects can be masked by noise.

Minor comments:

Mud and Dsh interact in HEK293T cells. The authors propose that this is explained by the expression of Meru orthologs. It's a very compelling idea that really should be pushed further. Can it be tested? The authors could at least mention how similar these orthologs are to Meru.

It would be interesting to study the function of the Meru orthologs, but we feel that this is out of the scope of the present manuscript. Our data demonstrating that Meru is required for Mud/Dsh association in *Drosophila* S2 cells is thoroughly controlled and convincingly backs our model without the need for validation in an exogenous system like HEK293 cells.

It's not clear to me what the live images are. Z-stacks? The authors point out in the Methods that they measured posterior and anterior localizations separately because they are not at the same depth. A specific concern is that the images shown in, for example, Figure 7C, are not representative of the quantifications in 7D.

As stated in the materials and methods, the measurements in 3C and 7D (now 6D) are taken from three apical and basolateral slices, whereas the images in 3A-B and 7B-C are maximum intensity projections to allow the reader to see both the anterior and posterior crescent in the same picture. We have clarified this in the figure legends.

Additional suggestions/questions:

The cartoon in Figure 1A illustrates an approximately 45 degree spindle tilt in the A-B axis. The measured tilt is 6.8 degrees. I do not dispute a "characteristic z-tilt" (Line 231) in SOP cells, but 6.8 is close enough to zero that a reasonable explanation for that number in these measurements is simple noise (whether biological or technical) in the data. This is especially true given that all data points are positive. (You'd have to measure only zeros to get a zero average). Can the authors comment on this?

We agree that the Figure 1A diagram shows an exaggerated view of the tilt for the purpose of illustration and the tilt is in reality more subtle. We have measured the z-tilt in order to be thorough in our analysis, as this has been done in previous work (e.g. PMID: 16228010; PMID: 21074723; PMID: 35581185).

However, unlike spindle orientation in the A-P direction, which impacts cell fate determinant segregation and therefore the differentiation of the SOP lineage, the biological relevance of the z-tilt to sensory organ differentiation is unknown. Thus, we feel the A-P phenotype, which is far easier to measure accurately, is of more functional importance. We agree with the reviewer that the z-tilt is subject to noise, although we have taken care to perform the measurement as accurately as possible.

The model these authors present (summarized in Figure 8) is that Meru links Mud to Dsh. While I don't consider it a requirement, it would be nice to see evidence that Meru and Dsh colocalise at confocal resolution, as in Figure 1G.

We have previously shown (PMID: 28665270) that Meru and Dsh colocalise, and that *dsh* and *fz* are required for Meru cortical recruitment.

I found the juxtaposed rose plots in figures 3 and 7 a little puzzling (at first) and ultimately hard to interpret. I suggest that the authors could find a different way to plot the data (e.g. violin plots). This would also facilitate comparison between figures.

Rose plots are widely used to visualise angular data, and we feel the rose plots are easier to navigate.

What is that big ball of Pins[RxL] in the middle of the cell!?

It is the nucleus.

Dr. Nicolas Tapon
The Francis Crick Institute
1 Midland Road
London NW1 1AT
United Kingdom

3rd Mar 2025

Re: EMBOJ-2023-116230R
Meru co-ordinates spindle orientation with cell polarity and cell cycle progression

Dear Nic,

Thank you for submitting your revised manuscript, which has now been re-reviewed by the original referees 2 and 3. I am happy to say that both consider the study significantly improved, and that we shall therefore be able to proceed with formal acceptance and publication, as soon as a few remaining specific comments by referee 3 (see below) will have been incorporated in a final minor revision round.

At this stage, please also take care of the following outstanding editorial issues:

- Please upload all main Figures and all Expanded View figures as individual files (with sufficient resolution/quality for production), separate from the editable text file of the manuscript (and the legends).
- Please adjust the order of the manuscript sections: Title page with complete author information, Abstract, Keywords, Introduction, Results, Discussion, Methods, Data Availability, Acknowledgements, Disclosure and Competing Interests Statement, References, Main Figure Legends, Tables, Expanded Figure Legends. Please rename the Conflict of Interest section into "Disclosure and Competing Interests Statement", in accordance with our updated Guide to Authors (<https://www.embopress.org/competing-interests>). And as we are switching from a free-text author contribution statement towards a more formal statement based on Contributor Role Taxonomy (CRediT) terms, please remove the present Author Contribution section and instead specify each author's contribution(s) directly in the Author Information page of our submission system during upload of the final manuscript. See <https://casrai.org/credit/> for more information.
- Please upload the source data according to a scheme that separates main and EV figure source data. There should be one archive per each of the main figures; and one single archive combining the source data for all EV figure. Also, it seems that source data for Figure 5B is missing, please check/amend.
- In the data availability section, please include a URL for the PRIDE repository where PXD060813 has been deposited. Also, please remove the reviewer access information and ensure that the data become publicly accessible at this stage.
- Please rename the supplementary movies as Expanded View movies (in-text callouts again "Movie EV1/2/..."). Their legends should be moved out of the text into individual text files, each of which should be combined with the respective movie file into a separate ZIP file and uploaded as such.
- During routine pre-acceptance image checks, we noted that Fig. 1D appears to constitute a composite of data panels EV1A/A' - please explicitly state this in the respective figure legends of both figures.
- During their pre-acceptance checks, our data editors have raised the following queries regarding figures, data, and legends, which I would ask you to address (ideally using the Track Changes option):
 1. Please note that the figure 8F does not contain a micrograph, kindly rectify the scale-bar-related information in the figure legend appropriately.
 2. Please note that the legends for figures EV5 is not provided in the sequential manner (legend for figure EV5 G is provided before legend of figure EV5 E, F). This needs to be rectified.
 3. Please note that the exact p values are not provided in the legends of figures 3C, 5G, 6D, 8C; EV5 C, D, G
 4. Please note that information related to N is missing in the legends of figures 8C, EV4 B.
 5. Please note that the scale bar needs to be defined for figures 8A, B, D, E.
- Finally, please provide suggestions for a short 'blurb' text prefacing and summing up the study in two sentences (max. 250 characters), followed by 3-5 one-sentence 'bullet points' with brief factual statements about key results of the paper; they will form the basis of an editor-written 'Synopsis' accompanying the online version of the article (see new articles on our journal website for some recent examples). You may also upload a simple synopsis image, sort-of a "visual title" for the synopsis section of your paper (maybe we could simply use Fig 8G for this purpose?). The image should be in PNG or JPG format with

the modest dimensions of 550 x 300-600 pixels (width x height).

I am returning the manuscript to you for a final round of revision, solely to allow you to make these modifications and upload the revised files. Once we will have received them, we should be able to proceed with formal acceptance and production of the manuscript.

With kind regards,

Hartmut

9) To facilitate reproducibility and cross-laboratory adoption of methodologies, please structure the Materials & Methods section as outlined in our guide to authors, including a completed Reagents and Tools Table that can be downloaded from our author guidelines as well (<https://www.embopress.org/page/journal/14602075/authorguide#structuredmethods>).

10) Digital image enhancement is acceptable practice, as long as it accurately represents the original data and conforms to community standards. If a figure has been subjected to significant electronic manipulation, this must be clearly noted in the figure legend and/or the 'Materials and Methods' section. The editors reserve the right to request original versions of figures and the original images that were used to assemble the figure. Finally, we generally encourage uploading of numerical as well as gel/blot image source data; for details see: embopress.org/page/journal/14602075/authorguide#sourcedata

At EMBO Press, we ask authors to provide source data for the main manuscript figures. Our source data coordinator will contact you to discuss which figure panels we would need source data for and will also provide you with helpful tips on how to upload and organize the files.

In the interest of ensuring the conceptual advance provided by the work, we recommend submitting a revision within 3 months (1st Jun 2025). Please discuss the revision progress ahead of this time with the editor if you require more time to complete the revisions. Use the link below to submit your revision:

Link Not Available

Referee #2:

The authors have thoroughly addressed my comments and have added a substantial amount of new data and clarifications to their manuscript. In my opinion the manuscript is much improved and warrants publication in The EMBO journal with no further revisions needed.

Referee #3:

In my initial review I commented on the impressive scholarship and careful experimental work in this paper. I'm happy to report that they kept it up. This a thorough, thoughtful revision. I'm afraid that I still have just a few concerns. These are described below in the form of a response to the response, but only to those few issues that remain after revision.

Major:

With respect to Figure 6D (formerly 7D), I noted previously that:

ORIGINAL: "Firstly, Meru[WT] is observed at both the anterior and posterior."

AUTHORS: While it is true that there is some Meru on the anterior side (perhaps as a result of overexpression), Meru remains strongly posteriorly biased in this experiment.

RESPONSE: The concern that remains is that the paper/model doesn't mention the anterior Meru signal nor provide an explanation for why Meru would be there, even if overexpressed.

Related to this, elsewhere in the response letter the authors note that a "reason to believe the UAS-mKate::Meru is overexpressed is that we can easily image it, whereas the mKate::Meru knockin line we have made is barely detectable by spinning disc microscopy," and that raises the possibility that a weaker anterior population of Meru was simply missed in the wild type condition because the imaging tools aren't strong enough detect it.

ORIGINAL: "Secondly, the authors made a convincing case in Figure 1G that Meru and Mud colocalise at confocal resolution. This does not appear to be strictly true in the Meru[WT] condition. They overlap somewhat but intensities don't match."

AUTHORS: Figure 1G is quantified from a single confocal slice, whereas Figure 7B (which is now Figure 6B in the revised manuscript) (Meru[WT]) is a maximum intensity projection so the reader can see both anterior and posterior crescent in the same image. Therefore, it is not surprising that the colocalization is less perfect. We have clarified this in the figure legends.

RESPONSE: Ok. I agree that the max intensity projection is going to be noisy and that explains why colocalization is less than perfect. A regular max projection (not max intensity) might be less noisy, but that is just a thought and not a concern for revision.

ORIGINAL: "Thirdly, there is no evidence to show where Dsh is in this condition."

AUTHORS: While this is an interesting question, we do not make any claims about Dsh localisation and it would be technically difficult to reliably follow Dsh localisation during mitosis with available reagents.

RESPONSE: I don't agree that the authors "do not make any claims about Dsh localisation" in the paper. The model presented is "that the scaffold protein Meru, which is recruited to the posterior cortex by the Frizzled/Dishevelled planar cell polarity complex, in turn recruits Mud" and that this is through direct interaction (a "Mud/Meru/Dsh complex"). *This is a claim that Dsh localizes at least to the same place as Mud and Meru, and that place is the posterior cortex.* In Figure 6D they clearly show that a population of Meru is at the opposite side of the cortex to where Dsh is expected.

In sum, the paper should acknowledge that there is anterior Meru in this image, suggest an explanation (this could require a

different mechanism for cortical localization of Meru than the one proposed), and incorporate it into the discussion.

Minor:

ORIGINAL: Another consideration is that the cortical localization of Mud being shown in Figure 3A is at anaphase. Is this the point being measured in C? This might impact interpretation, since Mud-dependent spindle orientation has already been achieved by anaphase.

AUTHORS: We chose to perform Mud measurements at anaphase because in SOPs the centrosomes are pulled towards the cortical domains at anaphase and spindle angle can still change until metaphase (PMID:25619594; PMID: 19214234).

RESPONSE: If the measurements are taken at anaphase (after Mud-dependent spindle orientation has occurred) then the reader should know that. The figure legend reads "(C) Graph showing the intensity of cortical Mud at the anterior and posterior crescent of each genotype." Please just add "Measurements were taken at anaphase."

ORIGINAL: What is that big ball of Pins[RxL] in the middle of the cell!?

AUTHORS: It is the nucleus.

RESPONSE: Ok, great. Why is there Pins in the nucleus? Please comment for the reader.

Typos

Lines 308; 319; 434; and 469: Error! Reference source not found.

The Francis Crick Institute Laboratory 1 Midland Road London NW1 1AT
+44 (0)203 796 0000 info@crick.ac.uk www.crick.ac.uk

Dr Nicolas Tapon
Principal Group Leader
The Francis Crick Institute
1 Midland Road
London NW1 1AT
United Kingdom
T +44 (0)203 796 2050 (off.)
F +44 (0)203 796 1253 (lab)
E nic.tapon@crick.ac.uk
<http://www.crick.ac.uk/nic-tapon>

London, 13 March 25

Dear Hartmut,

I am happy to submit our revised manuscript entitled "**Meru co-ordinates spindle orientation with cell polarity and cell cycle progression**" (EMBOJ-2023-116230). As requested, we have replied to the last few requests of reviewer 3 (see detailed below) and have made the formatting changes you requested in your decision letter. For the requested changes to the Figure legends, we left change tracker on. For one of these changes ("Please note that the exact p values are not provided in the legends of figures 3C, 5G, 6D, 8C; EV5 C, D, G") we saw that the exact p values were actually provided so we were not sure what to change.

We also uploaded the blurbs and bullet points you requested, as well as a separate file with Fig. 8G to be used as a synopsis image.

Referee #3:

In my initial review I commented on the impressive scholarship and careful experimental work in this paper. I'm happy to report that they kept it up. This a thorough, thoughtful revision. I'm afraid that I still have just a few concerns.

These are described below in the form of a response to the response, but only to those few issues that remain after revision.

Major: With respect to Figure 6D (formerly 7D), I noted previously that:
ORIGINAL: "Firstly, Meru[WT] is observed at both the anterior and posterior."

AUTHORS: While it is true that there is some Meru on the anterior side (perhaps as a result of overexpression), Meru remains strongly posteriorly biased in this experiment.

RESPONSE: The concern that remains is that the paper/model doesn't mention the anterior Meru signal nor provide an explanation for why Meru would be there, even if overexpressed.

Related to this, elsewhere in the response letter the authors note that a "reason to believe the UAS-mKate::Meru is overexpressed is that we can easily image it, whereas the mKate::Meru knockin line we have made is barely detectable by spinning disc microscopy," and that raises the possibility that a weaker anterior population of Meru was simply missed in the wild type condition because the imaging tools aren't strong enough to detect it.

ORIGINAL: "Secondly, the authors made a convincing case in Figure 1G that Meru and Mud colocalise at confocal resolution. This does not appear to be strictly true in the Meru[WT] condition. They overlap somewhat but intensities don't match."

AUTHORS: Figure 1G is quantified from a single confocal slice, whereas Figure 7B (which is now Figure 6B in the revised manuscript) (Meru[WT]) is a maximum intensity projection so the reader can see both anterior and posterior crescent in the same image. Therefore, it is not surprising that the colocalization is less perfect. We have clarified this in the figure legends.

RESPONSE: Ok. I agree that the max intensity projection is going to be noisy and that explains why colocalization is less than perfect. A regular max projection (not max intensity) might be less noisy, but that is just a thought and not a concern for revision.

ORIGINAL: "Thirdly, there is no evidence to show where Dsh is in this condition."

AUTHORS: While this is an interesting question, we do not make any claims about Dsh localisation and it would be technically difficult to reliably follow Dsh localisation during mitosis with available reagents.

RESPONSE: I don't agree that the authors "do not make any claims about Dsh localisation" in the paper. The model presented is "that the scaffold protein Meru, which is recruited to the posterior cortex by the Frizzled/Dishevelled planar cell polarity complex, in turn recruits Mud" and that this is through direct interaction (a "Mud/Meru/Dsh complex"). *This is a claim that Dsh localizes at least to the same place as Mud and Meru, and that place is the posterior cortex.* In Figure 6D they clearly show that a population of Meru is at the opposite side of the cortex to where Dsh is expected.

In sum, the paper should acknowledge that there is anterior Meru in this image, suggest an explanation (this could require a different mechanism for cortical localization of Meru than the one proposed), and incorporate it into the discussion.

As requested by the reviewer, we have clarified the relationship between Dsh and Meru as follows:

- Throughout the text, instead of writing that Meru is **recruited** posteriorly by Dsh, we now write that Meru is **enriched** at the posterior cortex by Dsh.

- When discussing Figure 6B' (we assume this is the panel the reviewer means rather than Fig 6D which is the quantification of Mud localisation), we now mention the Meru anterior localisation:

"We also note that, while Meru^{WT} is enriched at the posterior cortex (Fig 6B'), we consistently detect some Meru at the anterior cortex (see Fig 4A for another example). This could be a consequence of overexpression or of the way in which Meru is recruited to the plasma membrane (see discussion for details)." (lines 317-320)

- In the discussion, we further elaborate on this point:

"While Meru is enriched at the posterior SOP cortex, we detect some signal at the anterior pole, especially upon GAL/UAS-driven overexpression (Fig 6B'). This could be because overexpressed Meru saturates endogenous Dsh, therefore spilling over to the anterior side, or may indicate a two-step Meru cortical recruitment process where Meru is first recruited to the plasma membrane independently of Dsh, which then biases Meru cortical localisation by preferentially retaining it posteriorly." (lines 437-443).

Minor:

ORIGINAL: Another consideration is that the cortical localization of Mud being shown in Figure 3A is at anaphase. Is this the point being measured in C? This might impact interpretation, since Mud-dependent spindle orientation has already been achieved by anaphase.

AUTHORS: We chose to perform Mud measurements at anaphase because in SOPs the centrosomes are pulled towards the cortical domains at anaphase and spindle angle can still change until metaphase (PMID:25619594; PMID: 19214234).

RESPONSE: If the measurements are taken at anaphase (after Mud-dependent spindle orientation has occurred) then the reader should know that. The figure legend reads "(C) Graph showing the intensity of cortical Mud at the anterior and posterior crescent of each genotype." Please just add "Measurements were taken at anaphase."

We have done this as requested.

ORIGINAL: What is that big ball of Pins[RxL] in the middle of the cell!?

AUTHORS: It is the nucleus.

RESPONSE: Ok, great. Why is there Pins in the nucleus? Please comment for the reader.

We have added the following sentence (underlined) to the results section:

“When we mutated this RRL motif in Pins, we observed a mislocalisation to the nucleus and delayed localisation to the anterior cortex during SOP mitosis (Fig EV6B and C). This could be due to competition between cortical localisation and nuclear import of Pins prior to nuclear envelope breakdown. Consistent with this idea, we observe a small amount of nuclear Pins upon expression of wild type Pins (Fig EV6B, -24min and -18 min time points), and loss of the anterior PCP component Vang leads to some Pins nuclear accumulation (Gomes et al., 2009).”
lines 386-390.

Typos

Lines 308; 319; 434; and 469: Error! Reference source not found.

This error occurred during conversion to PDF of the original Word file. We have fixed it.

I look forward to hearing from you.

Best regards,

Nic Tapon

Dr. Nicolas Tapon
The Francis Crick Institute
1 Midland Road
London NW1 1AT
United Kingdom

17th Mar 2025

Re: EMBOJ-2023-116230R1
Meru co-ordinates spindle orientation with cell polarity and cell cycle progression

Dear Nic,

Thank you for submitting your final revised manuscript for our consideration. I am pleased to inform you that we have now accepted it for publication in The EMBO Journal.

With kind regards,

Hartmut
